# Exploiting Low-Dimensional Manifold of Features for Few-Shot Whole Slide Image Classification

**Conghao Xiong**[1,2]   **Zhengrui Guo**[3]   **Zhe Xu**[1]   **Yifei Zhang**[4]*   **Raymond Kai-yu Tong**[1]
**Si Yong Yeo**[2,5,6]   **Hao Chen**[3]   **Joseph J. Y. Sung**[2,5]   **Irwin King**[1]
[1]The Chinese University of Hong Kong   [2]Centre of AI in Medicine, Singapore
[3]The Hong Kong University of Science and Technology   [4]Nanyang Technological University
[5]Lee Kong Chian School of Medicine, Nanyang Technological University
[6]MedVisAI Lab, Singapore
Email: {chxiong21,king}@cse.cuhk.edu.hk, yifeiacc@gmail.com

## Abstract

Few-shot Whole Slide Image (WSI) classification is severely hampered by over-fitting. We argue that this is not merely a data-scarcity issue but a fundamentally geometric problem. Grounded in the manifold hypothesis, our analysis shows that features from pathology foundation models exhibit a low-dimensional manifold geometry that is easily perturbed by downstream models. This insight reveals a key potential issue in downstream multiple instance learning models: linear layers are geometry-agnostic and, as we show empirically, can distort the manifold geometry of the features. To address this, we propose the Manifold Residual (MR) block, a plug-and-play module that is explicitly geometry-aware. The MR block reframes the linear layer as residual learning and decouples it into two pathways: (1) a fixed, random matrix serving as a geometric anchor that approximately preserves topology while also acting as a spectral shaper to sharpen the feature spectrum; and (2) a trainable, low-rank residual pathway that acts as a residual learner for task-specific adaptation, with its structural bottleneck explicitly mirroring the low effective rank of the features. This decoupling imposes a structured inductive bias and reduces learning to a simpler residual fitting task. Through extensive experiments, we demonstrate that our approach achieves state-of-the-art results with significantly fewer parameters, offering a new paradigm for few-shot WSI classification. Code is available in https://github.com/BearCleverProud/MR-Block.

## 1 Introduction

Histopathology is the gold standard for disease diagnosis (Chen et al., 2024; Guo et al., 2024; Xiong et al., 2024a;b), and its computational analysis via Whole Slide Images (WSIs) operates under a uniquely challenging paradigm defined by two constraints. First, the gigapixel nature of WSIs makes Multiple Instance Learning (MIL) the de facto setting, where each slide is represented as a bag of patch features (Ilse et al., 2018; Li et al., 2021; Zhang et al., 2022). Second, expert-intensive annotation limits supervision, yielding datasets that are both small (few-shot) and weakly labeled, typically with only slide-level labels (Lu et al., 2021). This few-shot, weakly supervised setting forces models to identify discriminative patterns from thousands of unlabeled instances using only a handful of slide-level labels (Xiong et al., 2023; Guo et al., 2025b), often leading to severe overfitting.

To understand the root cause of overfitting, we *look beyond the learning algorithm and into the intrinsic structure of the features themselves*. Grounded in the manifold hypothesis (Tenenbaum et al., 2000), we investigate the intrinsic geometry of Camelyon16 (Litjens et al., 2018) under different feature extractors, including CONCH (Lu et al., 2024), UNI (Chen et al., 2024), and ResNet-50 (He et al., 2016) (UNI and ResNet-50 results are reported in Sec. B), and present multi-faceted evidence that these representations lie on a low-dimensional, nonlinear manifold. First, spectral

---
*Corresponding author.

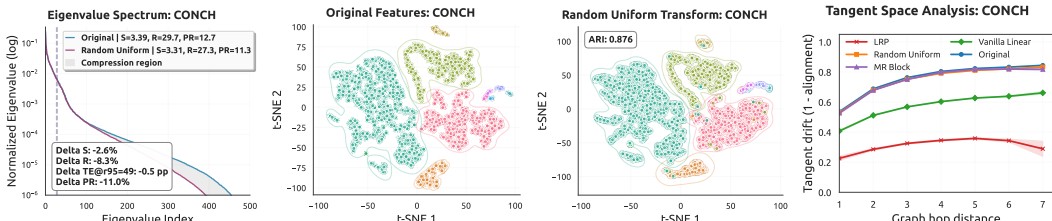

Figure 1: Feature geometric analysis reveals a low-dimensional manifold, which linear layers distort and our method preserves. (a) Spectral analysis confirms low-dimensionality. (b, c) t-SNE shows a cluster topology, and a random transformation preserves it. (d) Tangent space analysis visualizes this preservation: our MR block maintains the manifold curvature, while a linear layer causes distortion.

analysis (Skean et al., 2025) reveals a low effective rank of 29.7 (*vs.* the 512-dimensional CONCH features), confirming low dimensionality (Fig. 1(a)). Second, t-SNE visualizations reveal a distinct cluster topology (Fig. 1(b)). Finally, and most critically, tangent space analysis shows a non-flat, distance-dependent growth of geometric drift (Fig. 1(d), Original), providing quantitative evidence of intrinsic manifold curvature, thus excluding the pure linear subspace hypothesis.

Therefore, we posit that one important contributing factor to overfitting is geometric: *while features from pathology foundation models exhibit a low-dimensional manifold geometry, existing MIL models often fail to preserve this fragile structure*. This is not incidental but a limitation rooted in their ubiquitous building block: the linear layer. Across MIL models, *e.g.* ABMIL (Ilse et al., 2018; Lu et al., 2021; Li et al., 2021), Transformers (Shao et al., 2021; Xiong et al., 2023; Tang et al., 2024), graph-based models (Chen et al., 2021; Li et al., 2024b), information-bottleneck (Zhang et al., 2024; Huang et al., 2024) and density-based (Zhu et al., 2024) approaches, the linear layer serves as the indispensable component. However, the linear layer is inherently geometry-agnostic, lacking an architectural bias to respect the manifold structure. Our tangent space analysis provides direct, empirical evidence of this destructive tendency: as shown in Fig. 1(d), a trained linear layer (Vanilla Linear in the figure) severely distorts the intrinsic geometry of the manifold compared to the original (blue line). Consequently, especially under the few-shot setting, these linear layers are prone to learning an overly complex mapping that does not respect the low-rank nature of features and distorts their structure, thereby discarding the geometric priors that the pathology foundation models learned during large-scale pretraining. This points to a fundamental research gap: a lack of mechanisms designed to actively preserve and leverage this crucial, yet fragile, manifold structure.

To bridge this research gap, we propose a plug-and-play Manifold Residual (MR) block. The MR block aims to better preserve the geometric structure of the pretrained features from the Pathology Foundation Models (PFMs). It reframes the feature mapping as residual learning and decouples it into two parallel paths: a frozen base path for geometric preservation and a trainable Low-rank Residual Path (LRP) for task-specific adaptation. The base path employs a fixed random matrix with Kaiming uniform initialization that serves as both a geometric anchor and a spectral sharpener. Theoretically, random projection theory shows that such a transformation provides a geometry-preserving embedding that, with high probability, approximately preserves pairwise Euclidean distances (Johnson et al., 1984; Dasgupta & Gupta, 2003) and local geodesic structure (Baraniuk & Wakin, 2009), and acts as a near-isometry on tangent spaces under standard smoothness assumptions (Papaspiliopoulos, 2020). Empirically, Fig. 1 shows that the geometric anchor preserves the cluster topology (c), retains the original distance-drift profile in tangent space (d), and sharpens the eigenvalue spectrum (a). The LRP employs a bottleneck structure to model the inherently low-rank task-specific residuals, explicitly mirroring the low-rank features. Although LRP alone substantially distorts the geometry (Fig. 1(d), LRP), this behavior is expected because it is designed to learn task-specific residuals. The geometric anchor counterbalances this effect (Fig. 1(d), Random Uniform). Together, the full MR block (Fig. 1(d), MR Block) preserves the manifold geometry while capturing task-specific signals. The final representation is the sum of the two paths, balancing structural stability with adaptivity.

The contribution of our paper can be summarized as follows:

1. We provide quantitative evidence that foundation-model features lie on a low-dimensional manifold and identify a common failure mode: linear layers can disrupt this fragile structure.

2. We propose the theoretically grounded plug-and-play Manifold Residual block that mitigates manifold degradation by combining a geometric anchor with a low-rank residual.

3. We conduct extensive experiments, including comparative experiments, ablation studies, and sensitivity analyses, to validate both the efficacy and efficiency of our method.

## 2 RELATED WORK

### 2.1 MULTIPLE INSTANCE LEARNING

In MIL, WSIs are tessellated into patches, which are then input into a pretrained neural network to obtain features. In terms of how slide-level predictions are computed from patch features, MIL can be categorized into two streams: *instance-level* and *embedding-level* approaches. In the instance-level paradigm, logits are first computed for each patch and then aggregated to yield a slide-level prediction (Campanella et al., 2019; Chikontwe et al., 2020; Hou et al., 2016; Kanavati et al., 2020; Pan et al., 2025). Conversely, embedding-level approaches first aggregate patch features via pooling or aggregator networks to form a slide-level embedding, which is subsequently used to generate the final prediction (Guo et al., 2025a; Huang et al., 2024; Ilse et al., 2018; Lin et al., 2023; Li et al., 2024b; Lu et al., 2021; Shao et al., 2021; Xiong et al., 2023; Zhang et al., 2022; Yu et al., 2025).

### 2.2 PATHOLOGY FOUNDATION MODELS AND FEW-SHOT WSI CLASSIFICATION

Previously, feature extraction relied on ImageNet-pretrained ResNet (He et al., 2016). However, the domain discrepancy between natural images and WSIs often limits their efficacy in the pathology domain. To address this, both vision and vision-language pathology foundation models have been developed (Xiong et al., 2025). For example, Virchow (Vorontsov et al., 2024), UNI (Chen et al., 2024), and GPFM (Ma et al., 2024) are pretrained on pathology image datasets, and VLMs, including PLIP (Huang et al., 2023), CONCH (Lu et al., 2024), mSTAR (Xu et al., 2024) and MUSK (Xiang et al., 2025), leverage image-text pairs to enhance multimodal tasks. These pathology foundation models enable few-shot weakly-supervised learning (FSWSL) in WSI classification, which is especially appealing due to the substantial costs and labor incurred by WSI annotations. FSWSL was introduced in TOP (Qu et al., 2023), and subsequent studies (Fu et al., 2024; Li et al., 2024a; Qu et al., 2024; Shi et al., 2024; Guo et al., 2025b) have further expanded on this paradigm.

Beyond pathology-specific FSWSL, generic few-shot and metric-based paradigms (Song et al., 2023) can also operate on bag-level embeddings. Prototypical Networks (Snell et al., 2017), for example, replace the final linear classifier with a distance-based prototype head. In this work, we keep the standard linear readout used in prior WSI MIL studies and do not apply MR to this small classifier head. Instead, MR is inserted into internal linear layers of the MIL backbone, where our tangent space analysis reveals substantial distortion of the low-dimensional manifold of pretrained pathology features. Prototype-based heads endow the embedding space with a simple Euclidean metric but do not explicitly control this internal tangent drift. MR is in principle orthogonal to either linear or prototype-based readouts and can be used together with these two techniques.

### 2.3 GEOMETRIC DEEP LEARNING AND REPRESENTATION ANALYSIS

The manifold hypothesis, central to geometric deep learning, posits that high-dimensional data often lie on a low-dimensional manifold (Tenenbaum et al., 2000; Fefferman et al., 2016; Brahma et al., 2016). The geometric properties of this learned manifold are shaped by the pretraining objectives, for instance, contrastive methods like CLIP (Radford et al., 2021) encourage a topology of well-separated semantic clusters (Tian et al., 2021); self-distillation approaches like DINO (Oquab et al., 2024) preserve fine-grained local structures; while supervised learning often prioritizes linear separability. These manifolds can be quantitatively characterized using spectral analysis of the Gram matrix (Skean et al., 2025), which yields metrics like the effective rank (Roy & Vetterli, 2007; Vershynin, 2009). To further probe for nonlinearity, tangent space analysis directly measures the curvature of the manifold, with significant non-zero curvature being the signature of a nonlinear structure (Papaspiliopoulos, 2020). Our work leverages these geometric tools to conduct a rigorous geometric analysis of the pathology features, revealing the properties our MR block is designed to preserve.

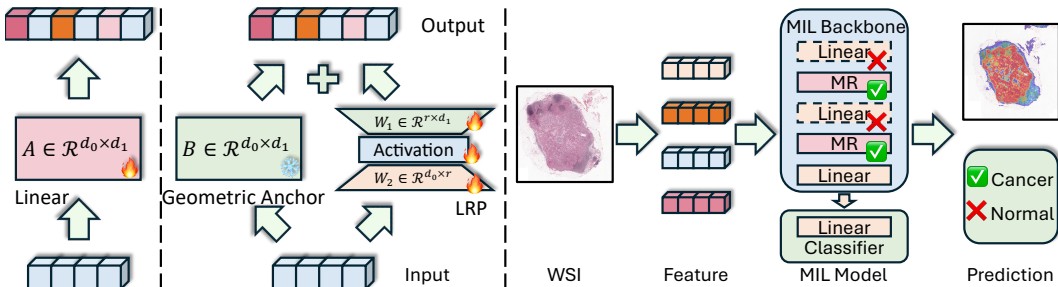

Figure 2: Illustration of the linear transformation (a), our proposed MR block (b), and the overall pipeline of MIL for WSI classification task and where our MR block takes effect (c). Specifically, the MR block is applied only to some of the internal linear layers within the MIL backbone, while the final linear classification head (the classifier) is left unchanged.

## 3 METHODOLOGY

We first formalize the bag-level MIL setting and introduce the geometric tools used to probe the feature manifold of the pathology foundation models, and then detail the Manifold Residual block, mitigating manifold degradation while enabling parameter-efficient task-specific adaptation. Finally, we present empirical evidence and theoretical support for our MR block.

### 3.1 PRELIMINARIES

#### 3.1.1 BAG-LEVEL MULTIPLE INSTANCE LEARNING FORMULATION

In this work, we focus on bag-level MIL. In MIL, we first partition the WSI into $N$ non-overlapping patches. A pretrained feature extractor then encodes each patch into a feature vector $\boldsymbol{f}_i \in \mathbb{R}^{d_p}$, where $d_p$ is the output dimension. The set of these patch features, known as the bag $\boldsymbol{F} = \{\boldsymbol{f}_i\}_{i=1}^N$, is aggregated by an MIL model to produce a single bag feature $\boldsymbol{f} = \mathrm{MIL}(\boldsymbol{F}) \in \mathbb{R}^{d_f}$, where $d_f$ is the dimension of the bag feature. Finally, this bag feature is fed into a classifier to obtain the slide-level logits $\boldsymbol{p} = \mathrm{CLS}(\boldsymbol{f}) \in \mathbb{R}^C$ with $C$ being the number of classes. Notably, most modern MIL models employ attention pooling to compute $\boldsymbol{f}$, which simultaneously yields both the bag feature and interpretable importance scores for each patch. This framework is illustrated in Fig. 2 (right).

#### 3.1.2 SPECTRAL ANALYSIS OF FEATURE REPRESENTATIONS

Spectral analysis is a classic technique used to estimate the intrinsic dimensionality of the feature representations. The process begins by computing the Gram matrix $\boldsymbol{K} = \boldsymbol{F}_n \boldsymbol{F}_n^\top$ from the normalized feature matrix $\boldsymbol{F}_n$. An eigendecomposition is then performed on the Gram matrix $\boldsymbol{K}$ to obtain the eigenvalues $\{\lambda_i(\boldsymbol{K})\}_{i=1}^N$, whose plotted distribution forms the spectral curve for visual analysis. For quantitative analysis, we normalize these eigenvalues into a probability distribution $p_i = \lambda_i(\boldsymbol{K}) / \sum_j \lambda_j(\boldsymbol{K})$ and use it to compute the Von Neumann entropy $S = -\sum_i p_i \log p_i$, and finally, the effective rank $R_{\mathrm{eff}} = \exp(S)$ (Skean et al., 2025) ($N \gg d_p$).

#### 3.1.3 TANGENT SPACE ANALYSIS

To probe for nonlinearity beyond the low-dimensionality, we measure the local curvature of the manifold via tangent space analysis (Zhang & Zha, 2004). This technique provides evidence of a nonlinear structure (Lim et al., 2024) by quantifying how the local geometry changes across the feature space. We approximate the geodesic paths by constructing a $k$-nearest-neighbor graph (Tenenbaum et al., 2000) ($k = 12$) over the features $\boldsymbol{F}_n$ with cosine similarity and compute the local tangent space $\mathcal{T}_i$ at each point $\boldsymbol{f}_i$ using local PCA (Kambhatla & Leen, 1997). Given the orthogonal bases of the $d_s$-dimensional tangent spaces $\boldsymbol{V}_i$ and $\boldsymbol{V}_j$, the Frobenius norm $\|\cdot\|_F$, the drift between the tangent spaces of two points, $\mathcal{T}_i$ and $\mathcal{T}_j$, can then be expressed as,

$$\mathcal{D}(\mathcal{T}_i, \mathcal{T}_j) = 1 - \frac{1}{d_s} \left\| \boldsymbol{V}_i^\top \boldsymbol{V}_j \right\|_F^2. \tag{1}$$

We use these metrics as qualitative diagnostics rather than direct predictors of performance gains. In practice, a single geometric statistic cannot reliably forecast the exact improvement on each dataset and backbone, because the final performance also depends on various other factors, such as architecture design, which layers are replaced, label difficulty, and the number of shots. Therefore, we only use these metrics to describe how vanilla linear layers and our MR block transform the pretrained feature manifold, rather than as dataset-level performance predictors.

## 3.2 MANIFOLD RESIDUAL BLOCK

The design of the MR block is an architectural response to the geometric deficiencies of vanilla linear layers. We first detail its architecture, and then its empirical motivation and theoretical guarantees.

### 3.2.1 ARCHITECTURE

To mitigate the geometric degradation introduced by a vanilla linear layer, we introduce the MR block, a parameter-efficient, plug-and-play substitute that is explicitly geometry-aware. As illustrated in Fig. 2(c), the MR block reframes a linear map as the sum of two pathways: a fixed geometric anchor for manifold preservation and a trainable LRP for task-specific adaptation.

Given the input $X \in \mathbb{R}^{N \times d_0}$ with the input feature dimension being $d_0$, our MR block is given by,

$$f_{\mathrm{MR}}(X) = \mathrm{GELU}(XW_2)W_1 + XB, \tag{2}$$

with output dimension $d_1$. Here $\mathrm{GELU}(\cdot)$ is the element-wise GELU activation function (Hendrycks & Gimpel, 2016); the LRP uses $W_2 \in \mathbb{R}^{d_0 \times r}$ and $W_1 \in \mathbb{R}^{r \times d_1}$ with rank parameter $r \ll \min(d_0, d_1)$; the anchor uses a fixed random matrix $B \in \mathbb{R}^{d_0 \times d_1}$ initialized with Kaiming uniform distribution (He et al., 2015). For initialization, we set $W_1 = 0$ and draw $W_2$ from Kaiming uniform distribution, so the residual path initially contributes zero and $f_{\mathrm{MR}}(X) = XB$ at step zero. During training, the LRP is activated only if it improves the training objective. Therefore, even though Fig. 1(d) shows that the tangent space drift of the LRP alone can be significant, the anchor balances this effect and the overall MR mapping better preserves manifold geometry.

Beyond preserving geometry, the MR block also reduces the number of trainable parameters. For any $r < (d_0 d_1)/(d_0 + d_1)$, the trainable parameter count of LRP, $r(d_0 + d_1)$, is strictly fewer than that of a vanilla linear layer ($d_0 d_1$), theoretically reducing the risk of overfitting (see Sec. C for details).

### 3.2.2 EMPIRICAL EVIDENCE

First, *we show that features are inherently low-dimensional by spectral analysis* (Sec. 3.1.2). The eigenvalue spectrum decays rapidly, yielding an effective rank of 29.7 ($\ll 512$). This provides quantitative evidence that variance is concentrated within a compact, low-dimensional structure.

Second, *we provide evidence that the representation forms a curved manifold rather than a linear subspace*. To distinguish a curved manifold from a linear subspace, we probe nonlinearity via tangent space analysis (Sec. 3.1.3). As shown in Fig. 1(d), geometric drift increases with graph-hop distance. This nonzero curvature indicates a curved, nonlinear manifold geometry that a trained linear layer fails to preserve. This geometric fragility is the central problem the MR block is designed to address.

### 3.2.3 COMPONENT JUSTIFICATION AND THEORETICAL GUARANTEES

*The geometric anchor preserves the manifold structure that the vanilla linear layers distort.* The fixed random projection serves as a geometric anchor. Random projection theory shows that this transformation approximately preserves most of the essential geometric properties (see Sec. D). Figure 1 empirically corroborates that the anchor approximately preserves cluster and neighborhood structure (Fig. 1(c)), maintains the intrinsic curvature pattern in tangent space drift (Fig. 1(d)), and sharpens the eigenvalue spectrum, improving spectral concentration and separation (Fig. 1(a)).

*The LRP enables task-specific adaptation.* Although our LRP is similar to LoRA (Hu et al., 2022), they serve fundamentally different purposes. The LRP structure is motivated by the low effective rank of features extracted by pathology foundation models. Its low-rank bottleneck imposes a structured inductive bias, ensuring that task-specific adaptation remains consistent with the low-rank nature of the manifold, as evidenced in Fig. 1(a). This reduces learning to a simpler residual-fitting task.

Table 1: Experimental results comparing our method with baselines are presented under various shot settings. Performance metrics in which our method outperforms the baseline are highlighted in italic blue and the best metrics are in bold red. "#P" is the number of parameters (million).

| $k$ | Methods | #P | Camelyon16 | | | NSCLC | | | RCC | | |
|---|---|---|---|---|---|---|---|---|---|---|---|
| | | | AUC↑ | F1↑ | Acc.↑ | AUC↑ | F1↑ | Acc.↑ | AUC↑ | F1↑ | Acc.↑ |
| 4 | ViLaMIL | 40.7 | $53.6_{12.4}$ | $48.4_{9.1}$ | $62.3_{2.3}$ | $82.9_{2.1}$ | $73.9_{2.9}$ | $74.3_{2.6}$ | $96.0_{1.4}$ | $81.7_{3.8}$ | $85.5_{2.9}$ |
| | FOCUS | 86.9 | $63.3_{16.5}$ | $56.4_{18.2}$ | $68.7_{9.0}$ | $87.9_{2.6}$ | $79.4_{2.2}$ | $79.5_{2.1}$ | $97.7_{2.1}$ | $87.2_{2.8}$ | $89.5_{2.7}$ |
| | ABMIL | 0.26 | $45.5_{7.6}$ | $49.2_{7.2}$ | $56.1_{5.2}$ | $86.5_{5.1}$ | $80.1_{5.0}$ | $80.2_{4.9}$ | $89.3_{6.4}$ | $74.2_{11.9}$ | $76.7_{11.3}$ |
| | **MR**-ABMIL | 0.10 | $46.1_{13.7}$ | $43.8_{10.0}$ | $56.6_{5.2}$ | $86.8_{2.5}$ | $79.3_{2.3}$ | $79.4_{2.1}$ | $96.5_{3.6}$ | $85.5_{6.0}$ | $87.8_{5.5}$ |
| | TransMIL | 2.41 | $51.9_{10.3}$ | $46.7_{7.5}$ | $51.0_{9.2}$ | $79.9_{2.9}$ | $71.0_{2.8}$ | $71.6_{2.5}$ | $96.5_{1.3}$ | $83.5_{2.3}$ | $86.6_{1.7}$ |
| | **MR**-TransMIL | 2.01 | $55.2_{11.0}$ | $50.9_{6.3}$ | $58.6_{6.5}$ | $83.0_{3.5}$ | $75.4_{2.6}$ | $75.5_{2.5}$ | $97.0_{1.2}$ | $84.0_{3.3}$ | $87.4_{2.5}$ |
| | CATE | 1.93 | $59.7_{12.4}$ | $50.0_{9.8}$ | $58.3_{6.7}$ | $81.4_{3.4}$ | $72.3_{4.1}$ | $72.8_{3.6}$ | $96.9_{1.8}$ | $83.9_{2.9}$ | $86.5_{2.5}$ |
| | **MR**-CATE | 0.84 | $62.1_{16.9}$ | $58.4_{15.8}$ | $62.3_{13.5}$ | $87.0_{3.9}$ | $79.4_{3.2}$ | $79.4_{3.2}$ | $\mathbf{98.2_{1.2}}$ | $\mathbf{88.7_{2.0}}$ | $\mathbf{90.5_{1.8}}$ |
| | DGRMIL | 4.08 | $68.0_{24.6}$ | $59.4_{22.0}$ | $66.0_{18.4}$ | $71.5_{10.6}$ | $60.3_{9.5}$ | $62.4_{7.7}$ | $96.4_{0.9}$ | $85.3_{2.4}$ | $87.4_{1.7}$ |
| | **MR**-DGRMIL | 3.29 | $65.4_{19.1}$ | $55.7_{18.4}$ | $63.6_{15.2}$ | $80.1_{3.4}$ | $72.5_{3.0}$ | $73.1_{2.8}$ | $95.8_{2.0}$ | $83.2_{3.9}$ | $86.3_{2.7}$ |
| | RRTMIL | 2.44 | $57.8_{4.4}$ | $54.8_{5.2}$ | $57.1_{6.3}$ | $65.7_{4.8}$ | $60.8_{2.9}$ | $61.5_{3.3}$ | $96.0_{1.9}$ | $81.6_{2.7}$ | $84.7_{2.3}$ |
| | **MR**-RRTMIL | 2.04 | $46.5_{3.8}$ | $42.7_{3.6}$ | $62.9_{0.6}$ | $69.9_{6.3}$ | $63.9_{5.0}$ | $64.0_{5.0}$ | $96.7_{1.4}$ | $82.5_{2.7}$ | $86.1_{2.2}$ |
| 8 | ViLaMIL | 40.7 | $74.0_{17.2}$ | $65.9_{22.1}$ | $71.8_{16.5}$ | $88.4_{5.2}$ | $79.2_{5.7}$ | $79.2_{5.7}$ | $96.8_{0.9}$ | $86.2_{2.1}$ | $88.0_{1.7}$ |
| | FOCUS | 86.9 | $87.2_{7.0}$ | $79.2_{9.4}$ | $80.2_{9.4}$ | $93.7_{5.6}$ | $85.0_{5.8}$ | $85.1_{5.8}$ | $98.1_{0.6}$ | $86.7_{2.3}$ | $88.9_{2.0}$ |
| | ABMIL | 0.26 | $49.2_{5.6}$ | $48.9_{3.0}$ | $59.7_{4.8}$ | $87.5_{5.9}$ | $81.1_{5.4}$ | $81.1_{5.4}$ | $91.8_{6.3}$ | $76.6_{8.4}$ | $79.1_{7.9}$ |
| | **MR**-ABMIL | 0.10 | $56.5_{15.5}$ | $53.8_{13.9}$ | $61.9_{8.7}$ | $94.0_{3.3}$ | $85.9_{3.3}$ | $86.0_{3.3}$ | $98.1_{0.8}$ | $87.6_{2.9}$ | $89.6_{2.1}$ |
| | TransMIL | 2.41 | $57.5_{2.5}$ | $52.0_{8.9}$ | $53.6_{7.4}$ | $87.9_{6.1}$ | $80.3_{5.2}$ | $80.4_{5.2}$ | $97.9_{0.6}$ | $86.5_{3.9}$ | $88.4_{3.1}$ |
| | **MR**-TransMIL | 2.01 | $65.3_{8.5}$ | $54.2_{9.3}$ | $63.6_{3.8}$ | $91.6_{4.1}$ | $82.4_{4.1}$ | $82.6_{4.0}$ | $97.9_{0.4}$ | $86.7_{2.5}$ | $89.1_{1.9}$ |
| | CATE | 1.93 | $79.8_{14.7}$ | $69.1_{20.0}$ | $76.0_{12.6}$ | $87.7_{5.7}$ | $78.0_{4.9}$ | $78.2_{4.9}$ | $97.8_{0.8}$ | $86.4_{3.0}$ | $88.3_{2.4}$ |
| | **MR**-CATE | 0.84 | $88.0_{14.8}$ | $81.7_{16.0}$ | $82.9_{15.3}$ | $93.5_{5.3}$ | $84.6_{5.7}$ | $84.7_{5.6}$ | $\mathbf{98.6_{0.3}}$ | $89.1_{1.3}$ | $\mathbf{90.7_{1.0}}$ |
| | DGRMIL | 4.08 | $84.7_{10.0}$ | $72.5_{17.8}$ | $77.4_{12.6}$ | $85.5_{6.3}$ | $73.4_{8.6}$ | $74.2_{7.7}$ | $96.7_{1.1}$ | $86.8_{2.9}$ | $88.7_{2.3}$ |
| | **MR**-DGRMIL | 3.29 | $\mathbf{93.4_{0.5}}$ | $\mathbf{86.0_{3.5}}$ | $\mathbf{86.7_{3.8}}$ | $91.6_{7.3}$ | $82.8_{9.6}$ | $83.0_{9.4}$ | $97.6_{1.0}$ | $85.1_{3.6}$ | $87.9_{1.7}$ |
| | RRTMIL | 2.44 | $68.8_{17.8}$ | $62.6_{15.4}$ | $63.7_{15.2}$ | $82.5_{9.9}$ | $73.9_{6.7}$ | $74.2_{6.5}$ | $97.0_{1.1}$ | $85.2_{3.3}$ | $86.9_{3.2}$ |
| | **MR**-RRTMIL | 2.04 | $85.3_{16.3}$ | $79.2_{16.6}$ | $81.7_{13.3}$ | $87.3_{7.8}$ | $79.6_{7.9}$ | $79.8_{7.8}$ | $97.8_{0.4}$ | $86.3_{1.5}$ | $87.8_{1.4}$ |
| 16 | ViLaMIL | 40.7 | $81.4_{23.1}$ | $77.5_{22.0}$ | $82.0_{15.0}$ | $92.8_{2.9}$ | $84.9_{2.7}$ | $85.0_{2.7}$ | $97.4_{1.1}$ | $85.5_{3.0}$ | $87.9_{2.2}$ |
| | FOCUS | 86.9 | $93.2_{2.5}$ | $\mathbf{89.0_{3.0}}$ | $\mathbf{89.8_{3.0}}$ | $\mathbf{96.5_{1.5}}$ | $\mathbf{89.3_{2.7}}$ | $\mathbf{89.3_{2.7}}$ | $98.2_{0.6}$ | $89.0_{2.1}$ | $90.6_{1.6}$ |
| | ABMIL | 0.26 | $73.2_{17.1}$ | $71.2_{15.4}$ | $75.7_{10.2}$ | $93.6_{5.2}$ | $87.8_{3.4}$ | $87.8_{3.4}$ | $94.7_{5.2}$ | $81.0_{7.8}$ | $82.8_{8.5}$ |
| | **MR**-ABMIL | 0.10 | $79.0_{23.1}$ | $74.5_{22.4}$ | $78.0_{17.1}$ | $96.5_{1.2}$ | $89.1_{1.0}$ | $89.1_{1.0}$ | $98.3_{0.7}$ | $89.1_{2.8}$ | $91.1_{2.0}$ |
| | TransMIL | 2.41 | $69.7_{15.7}$ | $62.5_{19.2}$ | $69.3_{13.3}$ | $95.6_{1.7}$ | $88.2_{3.6}$ | $88.2_{3.6}$ | $98.4_{0.3}$ | $89.3_{1.6}$ | $90.9_{0.7}$ |
| | **MR**-TransMIL | 2.01 | $79.3_{17.4}$ | $75.1_{14.8}$ | $77.1_{13.2}$ | $93.2_{4.4}$ | $84.0_{6.1}$ | $84.1_{6.0}$ | $\mathbf{98.6_{0.3}}$ | $89.3_{1.6}$ | $91.2_{1.0}$ |
| | CATE | 1.93 | $92.2_{3.1}$ | $87.5_{3.2}$ | $88.5_{2.8}$ | $94.2_{2.1}$ | $85.1_{2.0}$ | $85.2_{2.0}$ | $97.9_{0.6}$ | $87.4_{2.7}$ | $89.3_{2.0}$ |
| | **MR**-CATE | 0.84 | $92.4_{3.2}$ | $86.7_{3.9}$ | $87.8_{3.3}$ | $96.1_{1.4}$ | $88.6_{3.3}$ | $88.7_{3.3}$ | $\mathbf{98.6_{0.6}}$ | $\mathbf{89.8_{1.1}}$ | $\mathbf{91.6_{2.3}}$ |
| | DGRMIL | 4.08 | $88.5_{5.0}$ | $79.4_{10.2}$ | $80.9_{10.3}$ | $92.4_{3.4}$ | $85.8_{4.0}$ | $85.8_{3.9}$ | $94.8_{1.4}$ | $86.3_{1.0}$ | $88.7_{1.2}$ |
| | **MR**-DGRMIL | 3.29 | $92.3_{5.8}$ | $87.6_{5.8}$ | $88.7_{5.2}$ | $93.9_{1.7}$ | $86.6_{2.7}$ | $86.7_{2.7}$ | $97.7_{0.8}$ | $88.0_{2.0}$ | $90.3_{1.5}$ |
| | RRTMIL | 2.44 | $84.0_{16.9}$ | $74.8_{12.9}$ | $75.5_{12.7}$ | $90.7_{3.2}$ | $82.6_{4.2}$ | $82.7_{4.1}$ | $98.3_{0.6}$ | $86.9_{2.6}$ | $88.8_{2.4}$ |
| | **MR**-RRTMIL | 2.04 | $\mathbf{93.7_{1.6}}$ | $89.0_{2.4}$ | $89.8_{2.3}$ | $91.0_{5.4}$ | $82.5_{7.6}$ | $82.8_{7.1}$ | $97.8_{0.8}$ | $86.5_{4.1}$ | $88.9_{2.9}$ |

*Approximation behavior of the MR block.* Theoretically, our MR block can approximate any linear layer to arbitrary precision, thereby providing a worst-case performance guarantee: if the MR block cannot improve over the linear layer then it can match behavior of the target optimal linear layer arbitrarily closely, ensuring no loss in worst-case performance. In addition, rather than directly proving our formulation, we establish a more fundamental and challenging result without activation functions. Subsequently, with the universal approximation theorem (Barron, 1993; Cybenko, 1989; Funahashi, 1989; Hornik et al., 1989; Lu & Lu, 2020), our architecture with nonlinear activation function can achieve even superior approximation precision. The proof is given in Sec. E.

## 4 EXPERIMENTS AND RESULTS

### 4.1 DATASET DESCRIPTIONS AND IMPLEMENTATION DETAILS

**Datasets.** We conduct extensive experiments on Camelyon16 (Litjens et al., 2018), TCGA-NSCLC and TCGA-RCC datasets. The details of the datasets used in this work are available in Sec. G.1.

**Evaluation Metrics.** We employ four complementary metrics for a comprehensive assessment of these data-imbalanced datasets: Area Under the Receiver Operating Characteristic Curve (AUC), Area Under the Precision-Recall Curve (AUPRC), macro F1-score, and Accuracy.

**Implementation Details.** To ensure a fair and robust comparison, we use a consistent set of hyperparameters and training and evaluation protocols for all methods, as detailed in Sec. G.2. The

Table 2: Ablation studies of our proposed method across varying shot settings on the Camelyon16, TCGA-NSCLC, and TCGA-RCC datasets. Metrics highlighted in bold red indicate the top result.

| $k$ | Methods | #P | Camelyon16 | | | NSCLC | | | RCC | | |
|---|---|---|---|---|---|---|---|---|---|---|---|
| | | | AUC↑ | F1↑ | Acc.↑ | AUC↑ | F1↑ | Acc.↑ | AUC↑ | F1↑ | Acc.↑ |
| **8** | Ours | 2.01 | **71.4**$_{14.9}$ | **65.5**$_{12.1}$ | **69.9**$_{7.3}$ | **95.1**$_{3.5}$ | **86.1**$_{4.7}$ | **86.2**$_{4.6}$ | 96.8$_{1.3}$ | 84.3$_{5.4}$ | 87.0$_{4.4}$ |
| | + $B$ FT | 2.54 | 55.3$_{1.1}$ | 55.8$_{2.7}$ | 60.6$_{2.3}$ | 88.7$_{2.1}$ | 81.7$_{3.3}$ | 81.8$_{3.3}$ | 96.4$_{1.9}$ | 84.3$_{3.6}$ | 86.4$_{3.2}$ |
| | - $B$ | 2.01 | 55.3$_{0.6}$ | 55.7$_{1.6}$ | 58.4$_{1.9}$ | 88.2$_{5.2}$ | 81.0$_{6.9}$ | 81.2$_{6.6}$ | 97.6$_{1.1}$ | 86.4$_{5.0}$ | 88.6$_{4.4}$ |
| | - $Bx$ | 2.01 | 56.3$_{2.5}$ | 55.8$_{2.9}$ | 59.1$_{2.7}$ | 92.7$_{2.7}$ | 84.7$_{3.4}$ | 84.9$_{3.4}$ | **98.1**$_{0.4}$ | **87.4**$_{3.7}$ | **89.3**$_{3.0}$ |
| | - LRP | 1.89 | 51.0$_{17.7}$ | 49.4$_{14.2}$ | 58.9$_{8.8}$ | 89.4$_{6.3}$ | 81.6$_{5.8}$ | 81.7$_{5.8}$ | 98.1$_{0.8}$ | 86.9$_{2.8}$ | 89.3$_{2.1}$ |
| **16** | Ours | 2.01 | **86.4**$_{11.7}$ | **80.9**$_{12.0}$ | **81.9**$_{11.8}$ | **96.0**$_{1.7}$ | **88.7**$_{1.4}$ | **88.7**$_{1.4}$ | 98.3$_{0.8}$ | 87.7$_{2.3}$ | 90.0$_{2.0}$ |
| | + $B$ FT | 2.54 | 51.3$_{9.4}$ | 47.2$_{7.3}$ | 61.1$_{4.5}$ | 93.1$_{2.4}$ | 87.2$_{1.2}$ | 87.2$_{1.2}$ | 97.4$_{1.1}$ | 85.4$_{2.8}$ | 87.9$_{2.4}$ |
| | - $B$ | 2.01 | 61.9$_{13.6}$ | 52.8$_{8.3}$ | 63.3$_{7.4}$ | 91.9$_{2.4}$ | 84.7$_{2.4}$ | 84.8$_{2.4}$ | 97.7$_{1.3}$ | 86.0$_{4.1}$ | 88.3$_{3.6}$ |
| | - $Bx$ | 2.01 | 72.2$_{19.8}$ | 67.6$_{19.9}$ | 73.5$_{13.0}$ | 95.5$_{0.7}$ | 87.4$_{1.0}$ | 87.4$_{1.0}$ | **98.5**$_{0.4}$ | **88.4**$_{2.1}$ | **90.6**$_{1.6}$ |
| | - LRP | 1.89 | 68.3$_{20.2}$ | 65.1$_{17.8}$ | 69.8$_{12.8}$ | 93.7$_{2.0}$ | 86.4$_{1.1}$ | 86.4$_{1.0}$ | 98.3$_{0.6}$ | 87.0$_{1.3}$ | 89.6$_{1.0}$ |

Table 3: Capacity-matched comparison between ABMIL and MR-ABMIL under various shot settings. MR-ABMIL uses an MR block with the same number of trainable parameters as the vanilla ABMIL.

| $k$ | Methods | #P | Camelyon16 | | | NSCLC | | | RCC | | |
|---|---|---|---|---|---|---|---|---|---|---|---|
| | | | AUC↑ | F1↑ | Acc.↑ | AUC↑ | F1↑ | Acc.↑ | AUC↑ | F1↑ | Acc.↑ |
| **4** | ABMIL | 0.26 | 45.5$_{7.6}$ | 49.2$_{7.2}$ | 56.1$_{5.2}$ | **86.5**$_{5.1}$ | **80.1**$_{5.0}$ | **80.2**$_{4.9}$ | 89.3$_{6.4}$ | 74.2$_{11.9}$ | 76.7$_{11.3}$ |
| | **MR**-ABMIL | 0.26 | **56.6**$_{16.3}$ | **54.0**$_{14.7}$ | **58.4**$_{13.2}$ | 78.7$_{4.0}$ | 70.8$_{3.7}$ | 71.1$_{3.5}$ | **97.7**$_{1.3}$ | **86.9**$_{2.5}$ | **89.3**$_{2.6}$ |
| **8** | ABMIL | 0.26 | 49.2$_{5.6}$ | 48.9$_{3.0}$ | 59.7$_{4.8}$ | 87.7$_{5.9}$ | **81.0**$_{5.4}$ | **81.1**$_{5.4}$ | 91.8$_{6.3}$ | 76.6$_{8.4}$ | 79.1$_{7.9}$ |
| | **MR**-ABMIL | 0.26 | **73.2**$_{15.0}$ | **70.7**$_{15.9}$ | **72.6**$_{15.8}$ | **88.9**$_{8.5}$ | 80.0$_{8.1}$ | 80.1$_{8.2}$ | **98.6**$_{0.6}$ | **88.7**$_{3.2}$ | **90.9**$_{2.9}$ |
| **16** | ABMIL | 0.26 | 73.2$_{17.1}$ | 71.2$_{15.4}$ | 75.7$_{10.2}$ | 93.6$_{2.4}$ | 86.9$_{2.8}$ | 87.0$_{2.8}$ | 94.7$_{5.2}$ | 81.0$_{7.8}$ | 82.8$_{8.5}$ |
| | **MR**-ABMIL | 0.26 | **84.3**$_{11.7}$ | **79.3**$_{11.2}$ | **80.3**$_{11.1}$ | **93.6**$_{1.9}$ | **87.3**$_{3.4}$ | **87.8**$_{3.4}$ | **98.8**$_{0.3}$ | **89.7**$_{2.0}$ | **91.7**$_{1.6}$ |

Table 4: Performance of MR-ABMIL with replaced linear layers under varying shot settings. "U" and "V" denote two matrices in MR-ABMIL.

| $k$ | Methods | #P | Camelyon16 | | | NSCLC | | | RCC | | |
|---|---|---|---|---|---|---|---|---|---|---|---|
| | | | AUC↑ | F1↑ | Acc.↑ | AUC↑ | F1↑ | Acc.↑ | AUC↑ | F1↑ | Acc.↑ |
| **4** | $V + U$ | 0.10 | 46.1$_{13.7}$ | 43.8$_{10.0}$ | 56.6$_{5.2}$ | **86.8**$_{2.5}$ | **79.3**$_{2.3}$ | **79.4**$_{2.1}$ | **96.5**$_{3.6}$ | **85.5**$_{6.0}$ | **87.8**$_{5.5}$ |
| | $V$ | 0.18 | 45.6$_{0.3}$ | 43.9$_{0.2}$ | **59.8**$_{1.4}$ | 86.1$_{3.5}$ | 78.2$_{3.2}$ | 78.4$_{3.1}$ | 94.0$_{4.8}$ | 79.0$_{7.5}$ | 81.7$_{7.3}$ |
| | $U$ | 0.18 | **52.8**$_{10.5}$ | **47.9**$_{6.6}$ | 59.4$_{2.5}$ | 85.5$_{6.8}$ | 77.2$_{9.5}$ | 77.6$_{8.9}$ | 90.4$_{8.4}$ | 73.4$_{11.5}$ | 77.4$_{10.3}$ |
| **8** | $V + U$ | 0.10 | 56.5$_{15.5}$ | 53.8$_{13.9}$ | 61.9$_{8.7}$ | 94.0$_{3.3}$ | 85.9$_{3.3}$ | 86.0$_{3.3}$ | **98.1**$_{0.8}$ | **87.6**$_{2.9}$ | **89.6**$_{2.1}$ |
| | $V$ | 0.18 | 71.4$_{14.9}$ | 65.5$_{12.1}$ | 69.9$_{7.3}$ | **95.1**$_{3.5}$ | **86.1**$_{4.7}$ | **86.2**$_{4.6}$ | 96.8$_{1.3}$ | 84.3$_{5.4}$ | 87.0$_{4.4}$ |
| | $U$ | 0.18 | **72.8**$_{19.1}$ | **67.8**$_{20.0}$ | **71.5**$_{16.6}$ | 94.6$_{4.7}$ | 85.8$_{6.8}$ | 85.9$_{6.7}$ | 95.1$_{3.8}$ | 79.8$_{11.7}$ | 83.6$_{7.9}$ |

specific locations where linear layers are replaced by MR blocks are described in Sec. G.4. The FLOP and the wall-clock inference time statistics are available in Sec. G.5.

## 4.2 COMPARISON WITH STATE-OF-THE-ART METHODS

Tables 1 and 7 to 9 present few-shot results (typically with $k \in \{2, 4, 8, 16\}$, where available) on multiple datasets, including tumor detection (Camelyon16), tumor subtyping (TCGA-NSCLC, TCGA-RCC), and treatment response (Boehmk, Trastuzumab; ABMIL variants only in Table 9). Three significant findings emerge from this analysis. First, across both artificially constructed few-shot datasets on large cohorts and inherently few-shot treatment response datasets, models augmented with the MR block consistently outperform their corresponding baselines over datasets and shot counts. On Camelyon16, TCGA-NSCLC, and TCGA-RCC, they match or surpass the state-of-the-art ViLaMIL (Shi et al., 2024) and FOCUS (Guo et al., 2025b) while using far fewer trainable parameters. Second, our MR block improves parameter efficiency: replacing the vanilla linear layer with our MR block yields smaller models that deliver better performance across multiple MIL backbones, indicating that MR provides a beneficial low-rank inductive bias rather than simply adding capacity. Third, performance increases steadily as $k$ grows for all methods, with MR variants showing the largest gains at moderate shots and remaining competitive at higher shots, where RCC results approach a ceiling. We also observe reduced variability across runs for several MR settings,

Table 5: Experimental results on different initialization methods in MR-CATE. "K." and "X." stands for Kaiming and Xavier initialization. "U." and "N." stands for uniform and normal distribution. The number following is the input parameter for initialization. The gray rows stand for our main setting.

| $k$ | Methods | Camelyon16 | | | NSCLC | | | RCC | | |
|---|---|---|---|---|---|---|---|---|---|---|
| | | AUC↑ | F1↑ | Acc.↑ | AUC↑ | F1↑ | Acc.↑ | AUC↑ | F1↑ | Acc.↑ |
| | K. N. 0 | $63.0_{16.1}$ | $49.5_{19.7}$ | $\mathbf{65.7}_{11.1}$ | $82.2_{6.2}$ | $75.0_{5.7}$ | $75.2_{5.5}$ | $98.0_{1.4}$ | $87.8_{1.8}$ | $90.1_{1.8}$ |
| | K. U. 0 | $62.4_{17.0}$ | $48.9_{16.9}$ | $63.7_{10.5}$ | $86.5_{4.4}$ | $79.4_{3.2}$ | $\mathbf{79.4}_{3.1}$ | $98.1_{1.4}$ | $88.2_{3.0}$ | $90.3_{2.5}$ |
| 4 | K. U. $\sqrt{5}$ | $62.1_{16.9}$ | $\mathbf{58.4}_{15.8}$ | $62.3_{13.5}$ | $\mathbf{87.0}_{3.9}$ | $79.4_{3.2}$ | $79.4_{3.2}$ | $\mathbf{98.2}_{1.2}$ | $\mathbf{88.7}_{2.0}$ | $\mathbf{90.5}_{1.8}$ |
| | X. N. 0 | $\mathbf{66.2}_{14.1}$ | $48.8_{20.1}$ | $64.7_{12.9}$ | $84.0_{5.3}$ | $76.3_{4.2}$ | $76.4_{4.2}$ | $98.1_{1.4}$ | $87.9_{3.4}$ | $90.1_{2.9}$ |
| | X. U. 0 | $61.1_{17.1}$ | $51.2_{17.8}$ | $63.6_{12.8}$ | $86.4_{4.6}$ | $79.1_{2.7}$ | $79.1_{2.6}$ | $98.1_{1.3}$ | $88.2_{2.9}$ | $90.2_{2.6}$ |
| | K. N. 0 | $93.8_{1.7}$ | $88.9_{4.6}$ | $89.6_{4.5}$ | $95.4_{2.2}$ | $86.9_{2.8}$ | $87.0_{2.8}$ | $98.6_{0.4}$ | $88.2_{2.2}$ | $89.6_{2.0}$ |
| | K. U. 0 | $92.2_{2.2}$ | $87.5_{1.9}$ | $88.4_{2.0}$ | $94.6_{2.3}$ | $86.3_{2.2}$ | $86.4_{2.1}$ | $98.6_{0.4}$ | $88.8_{2.3}$ | $90.9_{2.0}$ |
| 16 | K. U. $\sqrt{5}$ | $92.4_{3.2}$ | $86.7_{3.9}$ | $87.8_{3.4}$ | $\mathbf{96.1}_{1.4}$ | $88.6_{3.3}$ | $88.7_{3.3}$ | $98.6_{0.6}$ | $\mathbf{89.8}_{3.1}$ | $\mathbf{91.6}_{2.3}$ |
| | X. N. 0 | $\mathbf{94.5}_{1.5}$ | $\mathbf{90.1}_{1.5}$ | $\mathbf{90.7}_{1.6}$ | $95.6_{2.0}$ | $88.5_{1.4}$ | $88.5_{1.4}$ | $\mathbf{98.7}_{0.5}$ | $88.9_{2.2}$ | $90.9_{1.4}$ |
| | X. U. 0 | $92.7_{3.7}$ | $88.5_{3.5}$ | $89.1_{3.4}$ | $95.8_{1.3}$ | $\mathbf{88.7}_{3.1}$ | $\mathbf{88.8}_{3.1}$ | $98.7_{0.4}$ | $89.5_{3.1}$ | $91.2_{2.5}$ |

Table 6: Experimental results on different pretrained feature extractors under various shot settings. "R50" stands for ResNet50, and "MR-" stands for MIL model with our MR block.

| $k$ | Methods | ABMIL | | | | | | CATE | | | | | |
|---|---|---|---|---|---|---|---|---|---|---|---|---|---|
| | | NSCLC | | | RCC | | | NSCLC | | | RCC | | |
| | | AUC↑ | F1↑ | Acc.↑ | AUC↑ | F1↑ | Acc.↑ | AUC↑ | F1↑ | Acc.↑ | AUC↑ | F1↑ | Acc.↑ |
| | R50 | $59.0_{3.7}$ | $54.7_{8.1}$ | $57.0_{4.6}$ | $74.6_{6.9}$ | $56.6_{9.0}$ | $58.5_{10.7}$ | $64.9_{7.8}$ | $57.4_{10.6}$ | $60.0_{6.9}$ | $82.6_{11.0}$ | $64.2_{10.2}$ | $68.9_{7.0}$ |
| 8 | MR-R50 | $64.3_{4.2}$ | $59.7_{3.3}$ | $60.1_{3.2}$ | $80.8_{4.5}$ | $62.3_{6.7}$ | $63.9_{8.3}$ | $63.5_{6.4}$ | $60.7_{5.7}$ | $61.4_{5.0}$ | $89.5_{3.6}$ | $75.0_{7.2}$ | $76.6_{6.0}$ |
| | UNI | $80.8_{7.9}$ | $72.8_{6.2}$ | $73.1_{6.2}$ | $98.6_{0.4}$ | $88.7_{2.1}$ | $90.3_{1.8}$ | $82.7_{7.8}$ | $72.4_{5.1}$ | $73.0_{5.1}$ | $98.1_{0.6}$ | $87.8_{3.4}$ | $89.3_{3.0}$ |
| | MR-UNI | $\mathbf{91.3}_{5.0}$ | $\mathbf{81.7}_{4.4}$ | $\mathbf{81.8}_{4.3}$ | $\mathbf{98.6}_{0.6}$ | $\mathbf{91.2}_{1.8}$ | $\mathbf{92.3}_{1.4}$ | $\mathbf{88.8}_{6.6}$ | $\mathbf{80.6}_{6.8}$ | $\mathbf{80.8}_{6.7}$ | $\mathbf{98.5}_{0.4}$ | $\mathbf{90.8}_{1.4}$ | $\mathbf{91.8}_{1.1}$ |
| | R50 | $65.6_{9.9}$ | $62.3_{9.0}$ | $62.5_{8.8}$ | $84.0_{4.0}$ | $67.7_{3.8}$ | $70.9_{5.1}$ | $66.3_{10.1}$ | $56.2_{12.8}$ | $61.1_{7.4}$ | $90.0_{8.0}$ | $74.7_{13.2}$ | $78.0_{9.0}$ |
| 16 | MR-R50 | $67.2_{6.6}$ | $63.7_{5.0}$ | $63.9_{4.8}$ | $85.9_{2.1}$ | $70.8_{2.4}$ | $73.4_{1.8}$ | $66.3_{12.6}$ | $59.2_{15.1}$ | $63.0_{8.2}$ | $94.1_{4.2}$ | $81.4_{5.1}$ | $82.4_{5.6}$ |
| | UNI | $91.1_{2.7}$ | $83.6_{3.0}$ | $83.6_{3.0}$ | $98.9_{0.3}$ | $89.2_{1.9}$ | $91.0_{1.8}$ | $87.2_{7.5}$ | $75.8_{9.2}$ | $76.9_{7.7}$ | $98.0_{0.9}$ | $88.0_{2.5}$ | $90.3_{1.6}$ |
| | MR-UNI | $\mathbf{94.8}_{2.8}$ | $\mathbf{87.0}_{3.6}$ | $\mathbf{87.0}_{3.6}$ | $\mathbf{98.9}_{0.4}$ | $\mathbf{91.1}_{2.4}$ | $\mathbf{92.4}_{1.8}$ | $\mathbf{91.7}_{4.6}$ | $\mathbf{82.6}_{6.0}$ | $\mathbf{82.9}_{5.6}$ | $\mathbf{98.5}_{0.6}$ | $\mathbf{91.0}_{1.9}$ | $\mathbf{92.2}_{1.5}$ |

suggesting improved training stability. Overall, MR is a plug-in replacement for vanilla linear layers in MIL models that improves accuracy, data efficiency, and robustness without increasing model size.

## 4.3 Ablation Studies and Sensitivity Analysis

**Ablation Study of MR Block Components.** Our ablation studies in Table 2 validate our decoupled design. Removing the LRP term ($f_{-\mathrm{LRP}} = \boldsymbol{X}\boldsymbol{B}$) consistently degrades performance, confirming its necessity for task adaptation. Removing the geometric anchor, either while retaining the residual connection ($f_{-\boldsymbol{B}} = \mathrm{GELU}(\boldsymbol{X}\boldsymbol{W}_2)\boldsymbol{W}_1 + \boldsymbol{X}\boldsymbol{I} = \mathrm{GELU}(\boldsymbol{X}\boldsymbol{W}_2)\boldsymbol{W}_1 + \boldsymbol{X}$) or while removing it ($f_{-\boldsymbol{B}\boldsymbol{X}} = \mathrm{GELU}(\boldsymbol{X}\boldsymbol{W}_2)\boldsymbol{W}_1$), is detrimental in both cases, underscoring its dual role as a geometric anchor and a spectral shaper. More importantly, making the anchor trainable leads to a catastrophic performance collapse. These observations provide direct empirical support for our core hypothesis: an unconstrained linear layer tends to shatter the feature manifold, whereas our design preserves it.

**Capacity-Matched Analysis of MR Block Effects.** To disentangle parameter reduction from the geometric inductive bias introduced by our MR block, we construct a capacity-matched MR-ABMIL in which the MR block has exactly the same number of trainable parameters as the original gated attention layer in ABMIL; the results are reported in Table 3. For a gated attention layer with input dimension $d_0 = 512$ and output dimension $d_1 = 256$, we set the rank to $r = (d_0 d_1)/(d_0 + d_1) = 170$, so that MR and the vanilla linear layer are exactly matched in parameter count. Across Camelyon16 and RCC, this capacity-matched MR-ABMIL consistently achieves substantial improvements in all metrics over the ABMIL baseline for all shot settings. On NSCLC, performance remains comparable, with a slight drop at 4-shot and similar or better performance at 8- and 16-shot. Since these gains are obtained under exactly matched trainable parameter counts, they provide direct evidence that the geometry-aware design of MR as a structural inductive bias, rather than parameter reduction alone, plays a central role in improving few-shot performance.

**Layer-wise Performance Analysis.** We conducted a layer-wise ablation study to evaluate how our method affects performance when implemented at different layers within MIL. We selected ABMIL (Ilse et al., 2018) due to its structural simplicity, with results reported in Tables 4 and 10.

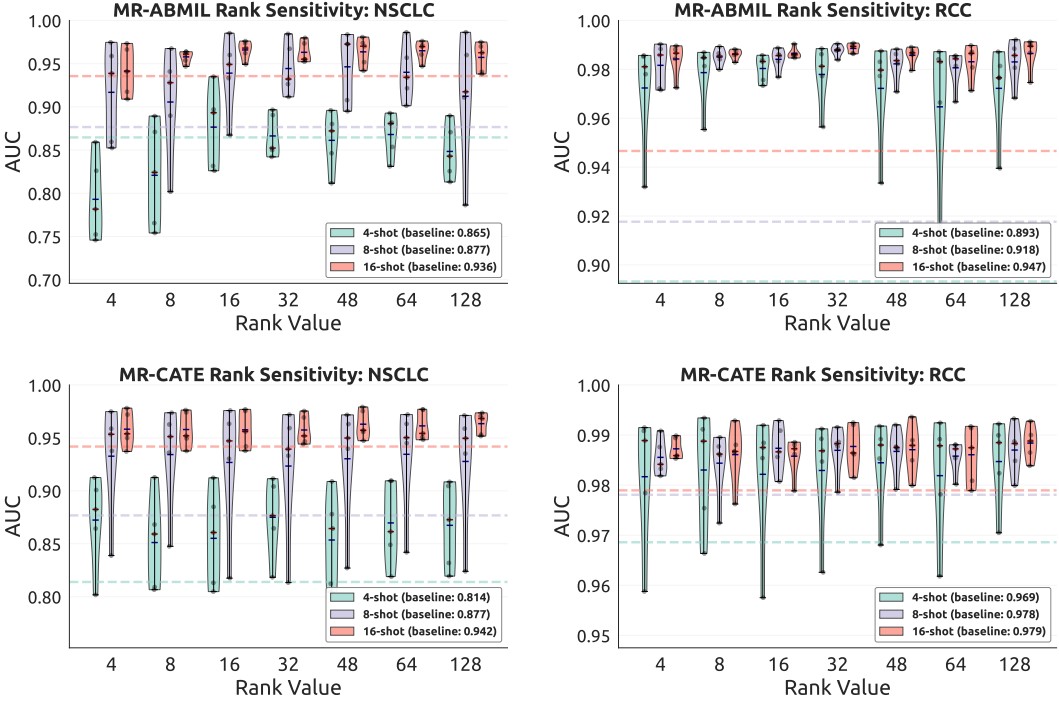

Figure 3: Sensitivity analysis of parameter $r$ across multiple datasets using MR-ABMIL and MR-CATE. Dashed lines indicate the performance of the baseline methods (*i.e.*, ABMIL and CATE).

This study investigates two components of ABMIL (Ilse et al., 2018), $U$ and $V$ (detailed in Sec. G.4), analyzing performances when selectively replacing these components with the MR block. The results demonstrate that optimal performance is achieved when both components are replaced with our MR block, and our method is robust to positions in MIL.

**Choice of Initialization.** We investigated the sensitivity of the MR block to the initialization scheme of $B$, by testing four schemes: Kaiming/Xavier Normal/Uniform. The results in Tables 5 and 11 demonstrate that our method is broadly insensitive to the specific choice of initialization. Across most settings, the performance differences between these methods are marginal, confirming that the benefits of the geometric anchor are not sensitive to initialization. Our chosen default, Kaiming Uniform with $\sqrt{5}$, consistently delivers superior performance, validating it as an effective choice without the need for additional hyper-parameter tuning.

**Foundation Model Selection.** We investigated whether our approach demonstrates compatibility across different foundation model feature extractors. Using CONCH, UNI (Chen et al., 2024), and ResNet50 (He et al., 2016) with ABMIL and CATE, we present results in Tables 6 and 12. Our method demonstrated consistent improvements over baselines across all model combinations, substantiating its robustness across diverse MIL and foundation models. CONCH embeddings consistently outperformed UNI and ResNet50, aligning with expectations as ResNet50 lacks pathology pretraining. The dimensional difference between UNI (1,024) and CONCH (512) likely contributes to lower performance of UNI due to inflated higher-dimensional spaces under few-shot settings.

**Sensitivity Analysis of the Rank.** The residual rank $r$ is a key design hyperparameter that directly controls how much information can flow through the low-rank path. We perform a sensitivity analysis over $r$, and the results are shown in Fig. 3. Across these datasets and shot settings, model performance saturates around $r = 32$. On the datasets and backbones evaluated, this empirically observed saturation point aligns closely with the theoretically predicted effective rank of the features, indicating that a low-rank path is sufficient to capture the essential task-specific information residing in the manifold structure. The simpler MR-ABMIL model peaks sharply at $r = 32$, while the more expressive MR-CATE exhibits only marginal gains at higher ranks, which we attribute to its additional capacity for modeling finer-grained feature interactions beyond the primary manifold structure.

**Analysis Beyond the Extreme Few-Shot Regime.** To assess whether MR continues to provide benefits when data volume increases, we conduct experiments on TCGA-NSCLC and TCGA-RCC with larger shots, $k \in \{32, 64\}$, and report the results in Table 13. Across both datasets and both shot settings, MR-ABMIL and MR-CATE consistently outperform their corresponding vanilla baselines on all metrics. These observations align with our interpretation of MR as a geometry-aware regularizer: the relative gains are typically stronger in the extreme few-shot regime, tend to decrease as supervision increases, yet remain positive and do not degrade performance even when more data is available.

## 4.4 MODEL INTERPRETABILITY ANALYSIS UNDER EXTREME LIMITED RESOURCE

Figure 6 illustrates the heatmaps generated by our MR-CATE, which demonstrates remarkable robustness in the extreme *2-shot setting* where the standard model (CATE) often fails to produce meaningful heatmaps. In the original images (upper left), blue curves delineate the approximate tumor boundaries, and the corresponding heatmap is shown in the lower left panel. Notably, our method captures additional fine-grained boundaries not presented in the original annotations, as shown in the right panel. Although all regions on the right fall within the blue-delineated tumor boundaries, our model precisely identifies the boundaries between distinct morphological patterns, demonstrating its sensitivity in capturing tumor and normal tissue heterogeneity.

## 5 CONCLUSION

In this work, we introduced a new geometric perspective on the problem of overfitting in few-shot WSI classification. We provided both quantitative and visual evidence that features from pathology foundation models exhibit a fragile, low-dimensional manifold geometry, and identified a common model failure mode of the current MIL models: the systematic degradation of this manifold structure by the geometry-agnostic linear layer, which is the indispensable component of modern MIL models. We proposed the Manifold Residual block, a plug-and-play module that preserves this geometry via a fixed random geometric anchor while enabling parameter-efficient, task-specific adaptation through a low-rank residual path. Our extensive experimental results not only achieve state-of-the-art results but also empirically support our geometric diagnosis, thereby establishing a new, geometry-aware paradigm for developing robust models with implications beyond computational pathology.

## REPRODUCIBILITY STATEMENT

Code is available in `https://github.com/BearCleverProud/MR-Block`.

In addition, we point to all components needed to replicate our results. The MIL setup and geometry tools are in Secs. 3.1, 3.1.2 and 3.1.3. The MR architecture is in Sec. 3.2.1; full experimental protocols (datasets, few-shot $k$, backbones, metrics) and implementation details are in Sec. G. Ablations and sensitivity studies (component/placement/initialization/rank) appear in Tables 2, 4 and 5 and Fig. 3 and their appendix counterparts. Assumptions and complete proofs are in Secs. D and E.

## ACKNOWLEDGEMENTS

The research presented in this paper was partially supported by the Research Grants Council of the Hong Kong Special Administrative Region, China (CUHK 2410072, RGC R1015-23) and (CUHK 2300246, RGC C1043-24G).

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

# Appendix

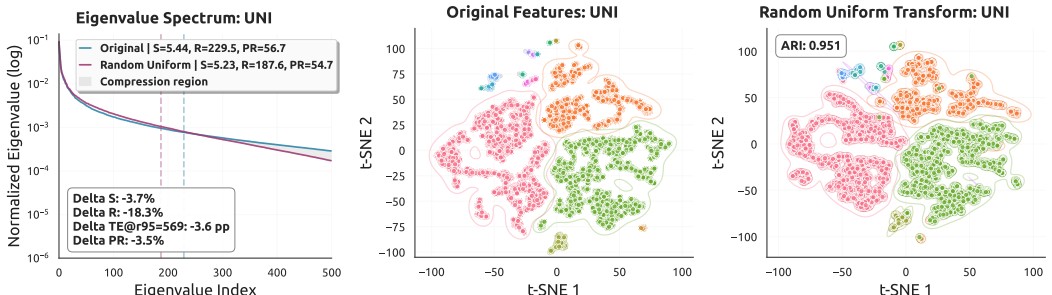

Figure 4: Random transformation preserves the intrinsic low-dimensional manifold of UNI features. (a) Spectral analysis confirms the low-rank structure. (b) t-SNE reveals the cluster topology of the manifold. (c) High ARI and consistent coloring visually and quantitatively prove this topology is robustly preserved.

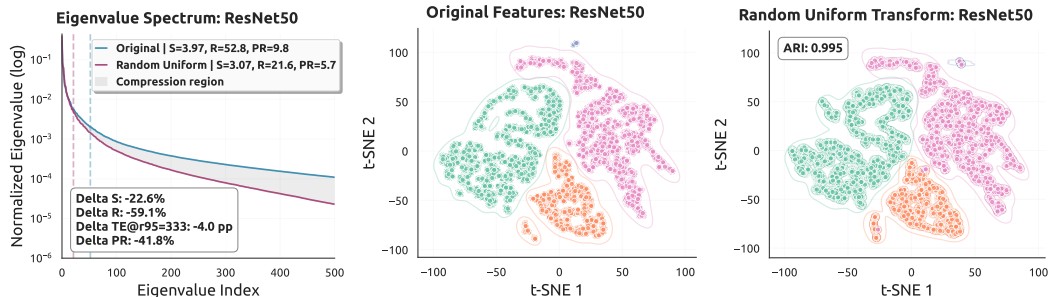

Figure 5: Random transformation preserves the intrinsic low-dimensional manifold of ResNet50 features. (a) Spectral analysis confirms the low-rank structure. (b) t-SNE reveals the cluster topology of the manifold. (c) High ARI and consistent coloring visually and quantitatively prove this topology is robustly preserved.

## A    LLM USAGE

The authors utilized LLMs as a writing assistant during the preparation of this manuscript. Its role was primarily focused on tasks related to language refinement and improving the clarity of the narrative. Its application included: (1) enhancing the conciseness and academic tone of sentences and paragraphs; (2) restructuring complex sentences for better readability; and (3) brainstorming alternative phrasings for key scientific arguments.

It is important to emphasize that all core scientific ideas, experimental design, results, and conclusions were exclusively developed by the human authors. The LLM served as a sophisticated editing and brainstorming tool. All text generated or modified by the LLM was critically reviewed, edited, and ultimately approved by the authors to ensure it accurately reflected our original intent and scientific findings.

## B    SPECTRAL ANALYSIS RESULTS

We apply the same procedures to UNI (Chen et al., 2024) and ResNet50 (He et al., 2016) features of Camelyon16 (Litjens et al., 2018) dataset, and present the results in Figs. 4 and 5. From these figures, we can see that features from these two feature extractors share similar characteristics as those from CONCH (Lu et al., 2024): rapid decaying spectrum and low effective rank. In addition, after applying a random uniform transformation on the features, the high ARIs and the cluster visualizations demonstrate that the random transformation can preserve the manifold structure well, both visually and quantitatively.

Interestingly, our analysis reveals a significant variance in the degree of this compression across models, which reflects their distinct, training-induced information encoding strategies. At one extreme,

ImageNet-pretrained ResNet50 exhibits a dramatic rank collapse (from 52.8 to 21.6). We argue this is not a sign of superior and compact structure discovery, but a domain-induced information collapse. This aligns with foundational studies on transfer learning, which show that ImageNet priors are often suboptimal for specialized domains like pathology (Raghu et al., 2019; Zoph et al., 2020), leading to a reliance on superficial "shortcut" features rather than genuine histological patterns (Geirhos et al., 2020). At the other extreme, CONCH, trained with text-alignment, is incentivized to learn a highly abstract, semantic-driven representation that captures key diagnostic concepts (Goh et al., 2021), a behavior consistent with findings in the vision-language models (Liang et al., 2022). This results in a highly compressed, specialized feature manifold. In contrast, UNI trained via self-supervision, is driven to capture a broader spectrum of morphological diversity. Its objective of instance discrimination forces the model to learn features sensitive to fine-grained visual details (Chen et al., 2020; Caron et al., 2021), resulting in a richer, higher-dimensional (yet still low-rank) feature manifold. Crucially, despite these differences, the features from both expert PFMs (CONCH and UNI) provide a far richer and more reliable structural prior than the collapsed representation of the generalist model.

## C COMPLEXITY ANALYSIS

### C.1 SPACE COMPLEXITY ANALYSIS

In the linear layer, the matrix $\boldsymbol{A}_*$ contains $d_0 d_1$ trainable parameters; however, in our approach, as $\boldsymbol{B}$ is frozen, the only trainable part is LRP, with $(d_0 + d_1)r$ trainable parameters. Defining the reduction ratio $r_r = (d_0 + d_1)r/(d_0 d_1)$, and enforcing $r_r < 1$, yields the constraint $r < (d_0 d_1)/(d_0 + d_1)$. If $r_r \geq 1$, the trainable parameter count in the MR block would match or exceed $d_0 d_1$, the trainable parameter count in $\boldsymbol{A}_*$, which makes our MR block meaningless.

### C.2 TIME COMPLEXITY ANALYSIS

We begin by analyzing a general case time complexity and then use the result to analyze our method.

In this section, we derive the computational cost of both the forward and backward passes through a single linear layer. We show that each pass scales linearly with the total number of weights $d_0 d_1$.

For convenience of the following derivations, we define the notations as follows,

$$M = [M_{ij}] \in \mathbb{R}^{d_0 \times d_1}, \tag{3}$$
$$x = [x_j] \in \mathbb{R}^{d_0}, \tag{4}$$
$$z = Mx = [z_i] \in \mathbb{R}^{d_1}, \tag{5}$$
$$\delta = \nabla_z L = [\delta_i] \in \mathbb{R}^{d_1}. \tag{6}$$

### C.3 FORWARD PASS

To compute the pre-activation vector $z$, each component $z_i$ performs a dot product between the $i$th row of $M$ and the input $x$, the $z_i$ is given by,

$$z_i = \sum_{j=1}^{d_0} M_{ij} x_j. \tag{7}$$

Since there are $d_1$ such outputs and each requires $d_0$ multiplications (and roughly the same number of additions), the total cost of the forward pass is $\mathcal{O}(d_0 d_1)$.

### C.4 BACKWARD PASS

The backward pass consists of three main steps: computing the gradient with respect to the weights, propagating the gradient to the inputs, and updating the weights.

**Weight Gradients**   The derivative of the loss $L$ with respect to each weight $M_{ij}$ follows by the chain rule, is given by,

$$\frac{\partial L}{\partial M_{ij}} = \frac{\partial L}{\partial z_i}\,\frac{\partial z_i}{\partial M_{ij}} = \delta_i \cdot x_j. \tag{8}$$

Collecting these into the full gradient matrix yields the outer product,

$$\nabla_M L = \delta\, x^\top, \tag{9}$$

which requires computing one scalar product for each of the $d_0 d_1$ entries. Hence, the cost of this step is $\mathcal{O}(d_0 d_1)$.

**Input Gradients**   To propagate the error back into the input space, we compute,

$$\nabla_x L = M^\top \delta. \tag{10}$$

This is a matrix–vector product of dimensions $d_0 \times d_1$, again costing $\mathcal{O}(d_0 d_1)$.

**Weight Update**   A typical gradient descent update modifies each entry of $M$ by

$$M_{ij} \;\leftarrow\; M_{ij} \;-\; \eta\,\frac{\partial L}{\partial M_{ij}}. \tag{11}$$

Updating all $d_0 d_1$ weights elementwise thus costs

$$\mathcal{O}(d_0\, d_1). \tag{12}$$

### C.5   Total Time Complexity

Summing the costs of the forward pass and all backward-pass components, we obtain $\mathcal{O}(d_0 d_1)$ complexity. Therefore, a single backpropagation step for one example in this layer has time complexity linear in the number of parameters $d_0 d_1$.

## D   Proof of Random Recalibration Matrix Preserving Input Characteristics

We categorize the geometric characteristics of the input features into variance and covariance; inner products and norms; cosine similarity; pairwise distances; condition numbers; the restricted isometry property; subspace embeddings; manifold geometry; cluster structure; nearest-neighbor relationships; and simplex volumes. We then demonstrate that each of these properties satisfies precise invariance conditions, ensuring that the recalibration matrix preserves the underlying geometric structure.

In this proof, we take the Kaiming uniform initialization with $a = \sqrt{5}$ as an example, which is the default initialization method for linear layers.

### D.1   Variance and Covariance Preservation

Intuitively, this result says that when you multiply your data covariance by a "random" matrix $M$, all of the original variance gets spread out evenly across the new $d_1$ dimensions, up to the constant factor $d_1/(3\,d_0)$. In other words, no particular direction in the original space is favored or suppressed on average.

To derive the scaling of total variance under projection, we start from

$$\mathrm{tr}(M^\top \Sigma\, M) = \sum_{p=1}^{d_1}\big(M^\top \Sigma\, M\big)_{pp} = \sum_{p=1}^{d_1}\sum_{q=1}^{d_0}\sum_{r=1}^{d_0} M_{q,p}\,\Sigma_{q,r}\,M_{r,p}. \tag{13}$$

Taking expectation and using linearity gives

$$E\big[\mathrm{tr}(M^\top \Sigma\, M)\big] = \sum_{p=1}^{d_1}\sum_{q=1}^{d_0}\sum_{r=1}^{d_0}\Sigma_{q,r}\,E\big[M_{q,p}\,M_{r,p}\big]. \tag{14}$$

Because the entries $M_{q,p}$ are independent with $E[M_{q,p}] = 0$, only terms with $q = r$ survive:

$$E\big[M_{q,p}\, M_{r,p}\big] = \begin{cases} \mathrm{Var}(M_{q,p}) = \frac{1}{3\,d_0}, & q = r, \\ 0, & q \neq r. \end{cases} \tag{15}$$

Substituting back, we have

$$E\big[\mathrm{tr}(M^\top \Sigma\, M)\big] = \sum_{p=1}^{d_1}\sum_{q=1}^{d_0} \Sigma_{q,q}\, \frac{1}{3\,d_0} = \frac{d_1}{3\,d_0}\, \sum_{q=1}^{d_0} \Sigma_{q,q} = \frac{d_1}{3\,d_0}\, \mathrm{tr}(\Sigma). \tag{16}$$

## D.2 INNER PRODUCTS AND NORMS

A random projection with independent, zero-mean entries treats every coordinate of your vectors in an unbiased, "democratic" way. On average, the dot product between any two vectors u and v simply gets scaled by the factor $d_1/(3\,d_0)$, but its sign and relative magnitude remain the same. Geometrically, this means angles and lengths are preserved in expectation, no particular direction in the original space is preferred, and so the overall similarity structure of the data survives the embedding up to a known scale.

To see how pairwise inner products scale, write

$$\langle M\, u,\ M\, v\rangle = \sum_{p=1}^{d_1} (M\, u)_p\, (M\, v)_p = \sum_{p=1}^{d_1}\sum_{i=1}^{d_0}\sum_{j=1}^{d_0} M_{p,i}\, u_i\, M_{p,j}\, v_j. \tag{17}$$

Taking expectation:

$$E\big[\langle M\, u,\ M\, v\rangle\big] = \sum_{p,i,j} u_i\, v_j\, E\big[M_{p,i}\, M_{p,j}\big]. \tag{18}$$

By independence and zero mean, only $i = j$ terms contribute:

$$E\big[M_{p,i}\, M_{p,j}\big] = \begin{cases} \frac{1}{3\,d_0}, & i = j, \\ 0, & i \neq j, \end{cases} \tag{19}$$

so

$$E\big[\langle M\, u,\ M\, v\rangle\big] = \sum_{p=1}^{d_1}\sum_{i=1}^{d_0} u_i\, v_i\, \frac{1}{3\,d_0} = \frac{d_1}{3\,d_0}\, \langle u, v\rangle. \tag{20}$$

Similarly, the norm squared follows as the special case $u = v$,

$$E\big[\|M\, u\|^2\big] = E\big[\langle M\, u,\ M\, u\rangle\big] = \frac{d_1}{3\,d_0}\, \|u\|^2. \tag{21}$$

## D.3 COSINE SIMILARITY

Cosine similarity measures only the angle between two vectors, not their lengths. Since the random projection scales all lengths and dot products by the same constant in expectation, it does not distort angles on average. In effect, no particular direction is stretched more than any other, so the typical cosine similarity between any two points remains exactly as it was before projection.

Since both inner products and norms incur the same factor $\frac{d_1}{3\,d_0}$ in expectation, the mean cosine similarity is exactly preserved:

$$E[\cos\theta'] = \frac{E[\langle M\, u,\ M\, v\rangle]}{\sqrt{E[\|M\, u\|^2]\, E[\|M\, v\|^2]}} = \frac{\frac{d_1}{3\,d_0} \langle u, v\rangle}{\sqrt{\frac{d_1}{3\,d_0}\|u\|^2\, \frac{d_1}{3\,d_0}\|v\|^2}} = \cos\theta. \tag{22}$$

## D.4 PAIRWISE DISTANCES

This property guarantees that a random projection will, with overwhelming likelihood, maintain the original distances between every pair of data points up to a small, controllable error. In effect, the

projection spreads each original distance across many independent components, and by concentration of measure those components collectively reproduce the true separation very closely. As a result, one can embed high-dimensional data into a much lower-dimensional space without significantly distorting the geometric relationships, ensuring that algorithms relying on interpoint distances, such as clustering or nearest-neighbor retrieval, continue to perform reliably.

Let $v = x_i - x_j$. Then

$$\|M\,v\|^2 = \sum_{p=1}^{d_1} \left( \sum_{q=1}^{d_0} M_{p,q}\,v_q \right)^2. \tag{23}$$

Define for each $p$,

$$Y_p = \sum_{q=1}^{d_0} M_{p,q}\,v_q, \quad E[Y_p] = 0, \quad \mathrm{Var}(Y_p) = \frac{\|v\|^2}{3\,d_0}. \tag{24}$$

By concentration of $\sum_p Y_p^2$, with probability $\geq 1 - \delta$,

$$(1 - \varepsilon)\,\|v\|^2 \leq \|M\,v\|^2 \leq (1 + \varepsilon)\,\|v\|^2, \tag{25}$$

provided

$$d_1 \geq C\,\varepsilon^{-2}\,\ln(N/\delta). \tag{26}$$

### D.5 CONDITION NUMBER

This property ensures that a random projection will not substantially worsen the numerical stability of the data. In other words, the projection maintains the balance between the directions in which the data stretches the most and the least. As a result, any algorithms that depend on solving linear systems or performing matrix decompositions will continue to behave reliably after projection, because the projected matrix remains nearly as well-conditioned as the original.

For data matrix $X$ and any unit $u$,

$$\|X\,M\,u\|^2 = u^\top M^\top X^\top X\,M\,u. \tag{27}$$

Applying matrix concentration on $XM$ shows that if

$$d_1 \geq C\,\varepsilon^{-2}\,d_0, \tag{28}$$

then with high probability

$$(1 - \varepsilon)\,u^\top X^\top X\,u \leq u^\top M^\top X^\top X\,M\,u \leq (1 + \varepsilon)\,u^\top X^\top X\,u, \tag{29}$$

so

$$\kappa(XM) \leq \frac{1 + \varepsilon}{1 - \varepsilon}\,\kappa(X) \approx (1 + \varepsilon)\,\kappa(X). \tag{30}$$

### D.6 RESTRICTED ISOMETRY PROPERTY

This property ensures that a random projection acts almost like a perfect length-preserver on all sparse vectors at once. In practice, it means that if the data has an underlying sparse structure, we can compress it drastically without losing the ability to tell sparse signals apart or to reconstruct them reliably. By making the projection dimension grow in line with how many nonzeros we expect, we guard against any sparse pattern being overly distorted.

For any $K$-sparse $x$,

$$\|M\,x\|^2 = \sum_{p=1}^{d_1} \left( \sum_{q \in \mathrm{supp}(x)} M_{p,q} x_q \right)^2. \tag{31}$$

A union bound over all supports yields that if

$$d_1 \geq C\,\delta^{-2}\,K\,\ln\frac{d_0}{K}, \tag{32}$$

then for all $K$-sparse $x$,

$$(1 - \delta)\,\|x\|^2 \leq \|M\,x\|^2 \leq (1 + \delta)\,\|x\|^2. \tag{33}$$

Table 7: Experimental results comparing our method with baselines are presented under various shot settings. Performance metrics in which our method outperforms the baseline are highlighted in italic blue and the best metrics are in bold red. "#P" is the number of parameters (million).

| $k$ | Methods | #P | Camelyon16 AUC↑ | F1↑ | Acc.↑ | NSCLC AUC↑ | F1↑ | Acc.↑ | RCC AUC↑ | F1↑ | Acc.↑ |
|---|---|---|---|---|---|---|---|---|---|---|---|
| | ViLaMIL | 40.7 | $53.8_{12.7}$ | $45.8_{11.5}$ | $61.1_{7.8}$ | $77.6_{5.0}$ | $69.4_{2.8}$ | $69.7_{2.8}$ | $90.3_{4.3}$ | $74.5_{6.8}$ | $79.7_{5.7}$ |
| | ABMIL | 0.26 | $41.2_{7.6}$ | $43.1_{6.5}$ | $46.5_{6.2}$ | $77.4_{14.1}$ | $68.4_{9.8}$ | $69.1_{9.4}$ | $84.4_{7.3}$ | $63.2_{8.1}$ | $64.7_{9.1}$ |
| | **MR**-ABMIL | 0.10 | $45.0_{11.6}$ | $43.2_{8.6}$ | $57.1_{6.2}$ | $85.5_{6.5}$ | $75.9_{7.2}$ | $76.3_{7.0}$ | $92.6_{6.3}$ | $78.1_{7.5}$ | $81.8_{7.2}$ |
| | TransMIL | 2.41 | $47.5_{9.4}$ | $45.0_{4.9}$ | $62.3_{1.7}$ | $74.5_{3.7}$ | $65.7_{6.0}$ | $66.6_{4.8}$ | $92.5_{4.2}$ | $75.1_{9.8}$ | $79.6_{7.4}$ |
| | **MR**-TransMIL | 2.01 | $51.2_{10.0}$ | $47.5_{8.9}$ | $64.0_{3.8}$ | $79.4_{7.1}$ | $71.4_{6.6}$ | $71.7_{6.4}$ | $93.5_{3.9}$ | $77.1_{5.4}$ | $81.7_{3.8}$ |
| 2 | CATE | 1.93 | $49.1_{10.3}$ | $45.7_{6.1}$ | $52.9_{7.8}$ | $78.9_{3.8}$ | $70.8_{3.2}$ | $71.0_{3.0}$ | $94.7_{2.1}$ | $81.7_{3.2}$ | $84.0_{2.5}$ |
| | **MR**-CATE | 0.84 | $54.9_{18.8}$ | $51.9_{16.1}$ | $54.3_{16.6}$ | $83.2_{7.9}$ | $75.1_{7.3}$ | $75.3_{7.2}$ | $96.0_{3.2}$ | $82.4_{4.1}$ | $85.3_{4.0}$ |
| | DGRMIL | 4.08 | $42.6_{6.4}$ | $42.7_{3.7}$ | $62.3_{2.2}$ | $68.1_{9.5}$ | $60.4_{8.7}$ | $61.8_{7.6}$ | $93.7_{2.6}$ | $77.8_{5.7}$ | $81.6_{4.2}$ |
| | **MR**-DGRMIL | 3.29 | $59.7_{2.5}$ | $52.0_{5.4}$ | $58.8_{3.7}$ | $74.1_{3.8}$ | $62.9_{7.8}$ | $64.8_{5.8}$ | $92.7_{3.5}$ | $75.8_{4.9}$ | $80.7_{3.5}$ |
| | RRTMIL | 2.44 | $56.0_{3.7}$ | $56.6_{6.0}$ | $62.9_{5.4}$ | $61.2_{6.7}$ | $55.5_{6.9}$ | $56.0_{6.5}$ | $91.6_{3.6}$ | $74.1_{8.7}$ | $78.0_{6.5}$ |
| | **MR**-RRTMIL | 2.04 | $47.6_{8.1}$ | $49.0_{5.8}$ | $61.7_{4.2}$ | $55.4_{13.1}$ | $53.3_{7.8}$ | $55.2_{6.5}$ | $91.6_{4.4}$ | $74.5_{6.0}$ | $78.7_{5.0}$ |
| | ViLaMIL | 40.7 | $53.6_{12.4}$ | $48.4_{9.1}$ | $62.3_{2.3}$ | $82.9_{2.1}$ | $73.9_{2.9}$ | $74.3_{2.6}$ | $96.0_{1.4}$ | $81.7_{3.8}$ | $85.5_{2.9}$ |
| | ABMIL | 0.26 | $45.5_{7.6}$ | $49.2_{7.2}$ | $56.1_{5.2}$ | $86.5_{5.1}$ | $80.1_{5.0}$ | $80.2_{4.9}$ | $89.3_{6.4}$ | $74.2_{11.9}$ | $76.7_{11.3}$ |
| | **MR**-ABMIL | 0.10 | $46.1_{13.7}$ | $43.8_{10.0}$ | $56.6_{5.2}$ | $86.8_{2.5}$ | $79.3_{2.3}$ | $79.4_{2.1}$ | $96.5_{3.6}$ | $85.5_{6.0}$ | $87.8_{5.5}$ |
| | TransMIL | 2.41 | $51.9_{10.3}$ | $46.7_{7.5}$ | $51.0_{9.2}$ | $79.9_{2.9}$ | $71.0_{2.8}$ | $71.6_{2.5}$ | $96.5_{1.3}$ | $83.5_{2.3}$ | $86.6_{1.7}$ |
| | **MR**-TransMIL | 2.01 | $55.2_{11.0}$ | $50.9_{6.3}$ | $58.6_{6.5}$ | $83.0_{3.5}$ | $75.4_{2.6}$ | $75.5_{2.5}$ | $97.0_{1.2}$ | $84.0_{3.3}$ | $87.4_{2.5}$ |
| 4 | CATE | 1.93 | $59.7_{12.4}$ | $50.0_{9.8}$ | $58.3_{6.7}$ | $81.4_{3.4}$ | $72.3_{4.1}$ | $72.8_{3.6}$ | $96.9_{1.8}$ | $83.9_{2.9}$ | $86.5_{2.5}$ |
| | **MR**-CATE | 0.84 | $62.1_{16.9}$ | $58.4_{15.8}$ | $62.3_{13.5}$ | $87.0_{3.9}$ | $79.4_{3.2}$ | $79.4_{3.2}$ | $98.2_{1.2}$ | $88.7_{2.0}$ | $90.5_{1.8}$ |
| | DGRMIL | 4.08 | $68.0_{24.6}$ | $59.4_{22.0}$ | $66.0_{18.4}$ | $71.5_{10.6}$ | $60.3_{9.5}$ | $62.4_{7.7}$ | $96.4_{0.9}$ | $85.3_{2.4}$ | $87.4_{1.7}$ |
| | **MR**-DGRMIL | 3.29 | $65.4_{19.1}$ | $55.7_{18.4}$ | $63.6_{15.2}$ | $80.1_{3.4}$ | $72.5_{3.0}$ | $73.1_{2.8}$ | $95.8_{2.0}$ | $83.2_{3.9}$ | $86.3_{2.7}$ |
| | RRTMIL | 2.44 | $57.8_{4.4}$ | $54.8_{5.2}$ | $57.1_{6.3}$ | $65.7_{4.8}$ | $60.8_{2.9}$ | $61.5_{3.3}$ | $96.0_{1.9}$ | $81.6_{2.7}$ | $84.7_{2.3}$ |
| | **MR**-RRTMIL | 2.04 | $46.5_{3.8}$ | $42.7_{3.6}$ | $62.9_{0.6}$ | $69.9_{6.3}$ | $63.9_{5.0}$ | $64.0_{5.0}$ | $96.7_{1.4}$ | $82.5_{2.7}$ | $86.1_{2.2}$ |
| | ViLaMIL | 40.7 | $74.0_{17.2}$ | $65.9_{22.1}$ | $71.8_{16.5}$ | $88.4_{5.2}$ | $79.2_{5.7}$ | $79.2_{5.7}$ | $96.8_{0.9}$ | $86.2_{2.1}$ | $88.0_{1.7}$ |
| | ABMIL | 0.26 | $49.2_{5.6}$ | $48.9_{3.0}$ | $59.7_{4.8}$ | $87.7_{5.9}$ | $81.0_{5.4}$ | $81.1_{5.4}$ | $91.8_{6.3}$ | $76.6_{8.4}$ | $79.1_{7.9}$ |
| | **MR**-ABMIL | 0.10 | $56.5_{15.5}$ | $53.8_{13.9}$ | $61.9_{8.7}$ | $94.0_{3.3}$ | $85.9_{3.3}$ | $86.0_{3.3}$ | $98.1_{0.8}$ | $87.6_{2.9}$ | $89.6_{2.1}$ |
| | TransMIL | 2.41 | $57.5_{2.5}$ | $52.0_{8.9}$ | $53.6_{7.4}$ | $87.9_{6.1}$ | $80.3_{5.2}$ | $80.4_{5.2}$ | $97.9_{0.6}$ | $86.5_{3.9}$ | $88.4_{3.1}$ |
| | **MR**-TransMIL | 2.01 | $65.3_{8.5}$ | $54.2_{9.3}$ | $63.6_{3.8}$ | $91.6_{4.1}$ | $82.4_{4.1}$ | $82.6_{4.0}$ | $97.9_{0.4}$ | $86.7_{2.5}$ | $89.1_{1.9}$ |
| 8 | CATE | 1.93 | $79.8_{14.7}$ | $69.1_{20.0}$ | $76.0_{12.6}$ | $87.7_{5.7}$ | $78.0_{4.9}$ | $78.2_{4.9}$ | $97.8_{0.8}$ | $86.4_{3.0}$ | $88.3_{2.4}$ |
| | **MR**-CATE | 0.84 | $88.0_{14.8}$ | $81.7_{16.0}$ | $82.9_{15.3}$ | $93.5_{5.3}$ | $84.6_{5.7}$ | $84.7_{5.6}$ | $98.6_{0.3}$ | $89.1_{1.3}$ | $90.7_{1.0}$ |
| | DGRMIL | 4.08 | $84.7_{10.0}$ | $72.5_{17.8}$ | $77.4_{12.6}$ | $85.5_{6.3}$ | $73.4_{8.6}$ | $74.2_{7.7}$ | $96.7_{1.1}$ | $86.8_{2.9}$ | $88.7_{2.3}$ |
| | **MR**-DGRMIL | 3.29 | $93.4_{0.5}$ | $86.0_{3.5}$ | $86.7_{3.8}$ | $91.6_{7.3}$ | $82.8_{9.6}$ | $83.0_{9.4}$ | $97.6_{1.0}$ | $85.1_{3.6}$ | $87.9_{1.7}$ |
| | RRTMIL | 2.44 | $68.8_{17.8}$ | $62.6_{15.4}$ | $63.7_{15.2}$ | $82.5_{9.9}$ | $73.9_{6.7}$ | $74.2_{6.5}$ | $97.0_{1.1}$ | $85.2_{3.3}$ | $86.9_{3.2}$ |
| | **MR**-RRTMIL | 2.04 | $85.3_{16.3}$ | $79.2_{16.6}$ | $81.7_{13.3}$ | $87.3_{7.8}$ | $79.6_{7.9}$ | $79.8_{7.8}$ | $97.8_{0.4}$ | $86.3_{1.5}$ | $87.8_{1.4}$ |
| | ViLaMIL | 40.7 | $81.4_{23.1}$ | $77.5_{22.0}$ | $82.0_{15.0}$ | $92.8_{2.9}$ | $84.9_{2.7}$ | $85.0_{2.7}$ | $97.4_{1.1}$ | $85.5_{3.0}$ | $87.9_{2.2}$ |
| | ABMIL | 0.26 | $73.2_{17.1}$ | $71.2_{15.4}$ | $75.7_{10.2}$ | $93.6_{2.4}$ | $87.8_{3.4}$ | $87.8_{3.4}$ | $94.7_{5.2}$ | $81.0_{7.8}$ | $82.8_{8.5}$ |
| | **MR**-ABMIL | 0.10 | $79.0_{23.1}$ | $74.5_{22.4}$ | $78.0_{17.1}$ | $96.5_{1.2}$ | $89.1_{1.0}$ | $89.1_{1.0}$ | $98.3_{0.7}$ | $89.1_{2.8}$ | $91.1_{2.0}$ |
| | TransMIL | 2.41 | $69.7_{15.7}$ | $62.5_{19.2}$ | $69.3_{13.3}$ | $95.6_{1.7}$ | $88.2_{3.6}$ | $88.2_{3.6}$ | $98.4_{0.3}$ | $89.3_{1.6}$ | $90.9_{0.7}$ |
| | **MR**-TransMIL | 2.01 | $79.3_{17.4}$ | $75.1_{14.8}$ | $77.1_{13.2}$ | $93.2_{4.4}$ | $84.0_{6.1}$ | $84.1_{6.0}$ | $98.6_{0.3}$ | $89.3_{1.6}$ | $91.2_{1.0}$ |
| 16 | CATE | 1.93 | $92.2_{3.1}$ | $87.5_{3.2}$ | $88.5_{2.8}$ | $94.2_{2.1}$ | $85.1_{2.0}$ | $85.2_{2.0}$ | $97.9_{0.6}$ | $87.4_{2.7}$ | $89.3_{2.0}$ |
| | **MR**-CATE | 0.84 | $92.4_{3.2}$ | $86.7_{3.9}$ | $87.8_{3.4}$ | $96.1_{1.4}$ | $88.6_{3.3}$ | $88.7_{3.3}$ | $98.6_{0.6}$ | $89.8_{3.1}$ | $91.6_{2.3}$ |
| | DGRMIL | 4.08 | $88.5_{5.0}$ | $79.4_{10.2}$ | $80.9_{10.3}$ | $92.4_{3.4}$ | $85.8_{4.0}$ | $85.8_{3.9}$ | $94.8_{1.4}$ | $86.3_{1.0}$ | $88.7_{1.2}$ |
| | **MR**-DGRMIL | 3.29 | $92.3_{5.8}$ | $87.6_{5.8}$ | $88.7_{5.2}$ | $93.9_{1.7}$ | $86.6_{2.7}$ | $86.7_{2.7}$ | $97.7_{0.8}$ | $88.0_{2.0}$ | $90.3_{1.5}$ |
| | RRTMIL | 2.44 | $84.0_{16.9}$ | $74.8_{12.9}$ | $75.5_{12.7}$ | $90.7_{3.2}$ | $82.6_{4.2}$ | $82.7_{4.1}$ | $98.3_{0.6}$ | $86.9_{2.6}$ | $88.8_{2.4}$ |
| | **MR**-RRTMIL | 2.04 | $93.7_{1.6}$ | $89.0_{2.4}$ | $89.8_{2.3}$ | $91.0_{5.4}$ | $82.5_{7.6}$ | $82.8_{7.1}$ | $97.8_{0.8}$ | $86.5_{4.1}$ | $88.9_{2.9}$ |

## D.7 SUBSPACE EMBEDDING

This property guarantees that a random projection acts as a near-perfect isometric embedding for any fixed low-dimensional subspace. It means that all geometric relationships, lengths and angles, within that subspace are preserved almost exactly after compression. As a result, any downstream algorithm that relies on the subspace structure (for example, solving linear least-squares problems or performing spectral decompositions) will perform almost identically in the reduced space, yet with greatly reduced computational and storage costs.

For a $d$-dimensional subspace with basis $A \in \mathbb{R}^{d_0 \times d}$, one shows with a net argument that if

$$d_1 = O(\varepsilon^{-2} d), \tag{34}$$

then with high probability

$$(1 - \varepsilon) A^\top A \preceq A^\top M^\top M A \preceq (1 + \varepsilon) A^\top A. \tag{35}$$

Table 8: Experimental results comparing our method with baselines are presented under various shot settings using data-imbalance metrics (AUPRC and F1-score). Performance metrics in which our method outperforms the baseline are highlighted in italic blue and the best metrics are in bold red.

| $k$ | Methods | #P | Camelyon16 | | NSCLC | | RCC | |
|---|---|---|---|---|---|---|---|---|
| | | | AUPRC↑ | F1↑ | AUPRC↑ | F1↑ | AUPRC↑ | F1↑ |
| 2 | ABMIL | 0.26 | $35.8_{6.6}$ | $43.1_{6.5}$ | $74.0_{12.0}$ | $68.4_{9.8}$ | $74.7_{11.5}$ | $63.2_{8.1}$ |
| | MR-ABMIL | 0.10 | $37.9_{12.1}$ | $43.2_{8.6}$ | $81.2_{9.3}$ | $75.9_{7.2}$ | $86.4_{11.3}$ | $78.1_{7.5}$ |
| | CATE | 1.93 | $40.7_{9.9}$ | $45.7_{6.1}$ | $74.6_{6.5}$ | $70.8_{3.2}$ | $89.6_{5.0}$ | $81.7_{3.2}$ |
| | MR-CATE | 0.84 | $48.2_{21.2}$ | $51.9_{16.1}$ | $80.1_{10.1}$ | $75.1_{7.3}$ | $91.0_{6.0}$ | $82.4_{4.1}$ |
| 4 | ABMIL | 0.26 | $40.7_{6.9}$ | $49.2_{7.2}$ | $84.7_{5.8}$ | $80.1_{5.0}$ | $82.7_{9.4}$ | $74.2_{11.9}$ |
| | MR-ABMIL | 0.10 | $39.0_{13.5}$ | $43.8_{10.0}$ | $81.9_{6.1}$ | $79.3_{2.3}$ | $93.6_{5.1}$ | $85.5_{6.0}$ |
| | CATE | 1.93 | $52.6_{12.1}$ | $50.0_{9.8}$ | $76.9_{4.9}$ | $72.3_{4.1}$ | $94.3_{2.5}$ | $83.9_{2.9}$ |
| | MR-CATE | 0.84 | $54.3_{19.6}$ | $58.4_{15.8}$ | $83.9_{7.9}$ | $79.4_{3.2}$ | $96.1_{2.4}$ | $88.7_{2.0}$ |
| 8 | ABMIL | 0.26 | $45.4_{4.1}$ | $48.9_{3.0}$ | $81.7_{7.2}$ | $81.0_{5.4}$ | $87.7_{8.2}$ | $76.6_{8.4}$ |
| | MR-ABMIL | 0.10 | $53.0_{19.1}$ | $53.8_{13.9}$ | $91.7_{5.2}$ | $85.9_{3.3}$ | $96.4_{1.2}$ | $87.6_{2.9}$ |
| | CATE | 1.93 | $76.8_{19.0}$ | $69.1_{20.0}$ | $84.9_{6.6}$ | $78.0_{4.9}$ | $96.0_{1.1}$ | $86.4_{3.0}$ |
| | MR-CATE | 0.84 | $84.9_{20.9}$ | $81.7_{16.0}$ | $92.6_{6.0}$ | $84.6_{5.7}$ | $97.2_{0.5}$ | $89.1_{1.3}$ |
| 16 | ABMIL | 0.26 | $72.5_{18.5}$ | $71.2_{15.4}$ | $91.0_{4.4}$ | $87.8_{3.4}$ | $90.8_{8.2}$ | $81.0_{7.8}$ |
| | MR-ABMIL | 0.10 | $77.9_{26.5}$ | $74.5_{22.4}$ | $96.3_{1.5}$ | $89.1_{1.0}$ | $96.4_{1.8}$ | $89.1_{2.8}$ |
| | CATE | 1.93 | $92.5_{2.7}$ | $87.5_{3.2}$ | $92.8_{3.7}$ | $85.1_{2.0}$ | $95.9_{1.3}$ | $87.4_{2.7}$ |
| | MR-CATE | 0.84 | $92.2_{3.3}$ | $86.7_{3.9}$ | $96.0_{1.8}$ | $88.6_{3.3}$ | $97.2_{1.1}$ | $89.8_{3.1}$ |

Table 9: Experimental results on two treatment response datasets.

| $k$ | Methods | #P | Boehmk | | | Trastuzumab | | |
|---|---|---|---|---|---|---|---|---|
| | | | AUC↑ | F1↑ | Acc.↑ | AUC↑ | F1↑ | Acc.↑ |
| 4 | ABMIL | 0.26 | $56.2_{6.7}$ | $45.9_{8.6}$ | $49.7_{10.8}$ | $47.7_{5.6}$ | $40.3_{3.9}$ | $42.4_{2.6}$ |
| | MR-ABMIL | 0.18 | $65.4_{11.3}$ | $56.9_{8.5}$ | $65.1_{3.7}$ | $56.9_{2.3}$ | $40.9_{8.7}$ | $58.8_{0.0}$ |
| 8 | ABMIL | 0.26 | $53.4_{5.1}$ | $46.1_{4.6}$ | $50.9_{8.2}$ | $49.4_{3.7}$ | $48.8_{10.8}$ | $54.1_{7.7}$ |
| | MR-ABMIL | 0.18 | $63.4_{9.5}$ | $57.6_{7.1}$ | $62.3_{9.8}$ | $55.1_{4.4}$ | $40.3_{4.5}$ | $54.1_{6.4}$ |
| 16 | ABMIL | 0.26 | $52.6_{5.5}$ | $46.9_{7.3}$ | $50.3_{8.0}$ | $46.3_{10.0}$ | $36.1_{1.3}$ | $48.2_{6.4}$ |
| | MR-ABMIL | 0.18 | $63.5_{4.1}$ | $55.1_{3.4}$ | $63.4_{6.5}$ | $57.7_{0.8}$ | $43.6_{3.3}$ | $56.5_{3.2}$ |

## D.8 MANIFOLD GEOMETRY

This property ensures that a random projection will faithfully reproduce the intrinsic geometry of any low-dimensional manifold embedded in high-dimensional space. Even though the data may lie on a curved, nonlinear surface, we can compress it down to far fewer dimensions and still retain nearly all of the true "geodesic" distances along that surface. The key idea is that the manifold can

Table 10: Performance of MR-ABMIL with replaced linear layers under varying shot settings on Camelyon16, TCGA-NSCLC, and TCGA-RCC datasets. "U" and "V" denote two matrices in MR-ABMIL.

| $k$ | Methods | #P | Camelyon16 | | | NSCLC | | | RCC | | |
|---|---|---|---|---|---|---|---|---|---|---|---|
| | | | AUC↑ | F1↑ | Acc.↑ | AUC↑ | F1↑ | Acc.↑ | AUC↑ | F1↑ | Acc.↑ |
| 4 | $V+U$ | 0.10 | $46.1_{13.7}$ | $43.8_{10.0}$ | $56.6_{5.2}$ | $86.8_{2.5}$ | $79.3_{2.3}$ | $79.4_{2.1}$ | $96.5_{3.6}$ | $85.5_{6.0}$ | $87.8_{5.5}$ |
| | $V$ | 0.18 | $45.6_{0.3}$ | $43.9_{0.2}$ | $59.8_{1.4}$ | $86.1_{3.5}$ | $78.2_{3.2}$ | $78.4_{3.1}$ | $94.0_{4.8}$ | $79.0_{7.5}$ | $81.7_{7.3}$ |
| | $U$ | 0.18 | $52.8_{10.5}$ | $47.9_{6.6}$ | $59.4_{2.5}$ | $85.5_{6.8}$ | $77.2_{9.5}$ | $77.6_{8.9}$ | $90.4_{8.4}$ | $73.4_{11.5}$ | $77.4_{10.3}$ |
| 8 | $V+U$ | 0.10 | $56.5_{15.5}$ | $53.8_{13.9}$ | $61.9_{8.7}$ | $94.0_{3.3}$ | $85.9_{3.3}$ | $86.0_{3.3}$ | $98.1_{0.8}$ | $87.6_{2.9}$ | $89.6_{2.1}$ |
| | $V$ | 0.18 | $71.4_{14.9}$ | $65.5_{12.1}$ | $69.9_{7.3}$ | $95.1_{3.5}$ | $86.1_{4.7}$ | $86.2_{4.6}$ | $96.8_{1.3}$ | $84.3_{5.4}$ | $87.0_{4.4}$ |
| | $U$ | 0.18 | $72.8_{19.1}$ | $67.8_{20.0}$ | $71.5_{16.6}$ | $94.6_{4.7}$ | $85.8_{6.8}$ | $85.9_{6.7}$ | $95.1_{3.8}$ | $79.8_{11.7}$ | $83.6_{7.9}$ |
| 16 | $V+U$ | 0.10 | $79.0_{23.1}$ | $74.5_{22.4}$ | $78.0_{17.1}$ | $96.5_{1.2}$ | $89.1_{1.0}$ | $89.1_{1.0}$ | $98.3_{0.7}$ | $89.1_{2.8}$ | $91.1_{2.0}$ |
| | $V$ | 0.18 | $86.4_{11.7}$ | $80.9_{12.0}$ | $81.9_{11.8}$ | $96.0_{1.7}$ | $88.7_{1.4}$ | $88.7_{1.4}$ | $98.3_{0.8}$ | $87.7_{2.3}$ | $90.0_{2.0}$ |
| | $U$ | 0.18 | $82.5_{18.4}$ | $77.9_{18.3}$ | $81.9_{12.0}$ | $96.9_{1.1}$ | $89.8_{1.5}$ | $89.8_{1.5}$ | $97.7_{1.1}$ | $86.3_{3.9}$ | $88.4_{3.5}$ |

be approximated by a finite set of local patches, and the projection preserves each patch's geometry so accurately that the entire shape remains intact. As a result, analyses that rely on the manifold's structure, such as nonlinear dimensionality reduction or manifold-based learning, remain valid and effective after compression.

Cover a $d$-dimensional manifold of volume $V$ by $N \approx (V/\tau)^d$ balls. If

$$d_1 \geq C\,\varepsilon^{-2}\,d\,\ln\frac{V}{\tau}, \tag{36}$$

then all geodesic distances are preserved within $1 \pm \varepsilon$.

### D.9 CLUSTER LABELS

This criterion ensures that the worst-case contraction of the gap between clusters still exceeds the worst-case expansion of each cluster's size. In practice it means that random projection will not cause any overlap between clusters that were originally well separated. As a result, any clustering or classification based on distance remains unchanged, and all original labels are preserved.

Clusters of diameter $D$ separated by $\Delta$ satisfy label preservation if

$$(1 - \varepsilon)\,\Delta > (1 + \varepsilon)\,D. \tag{37}$$

### D.10 NEAREST NEIGHBORS

The nearest-neighbor property refers to the guarantee that, after random projection, each point's set of $k$ closest points (its $k$-nearest-neighbor graph) remains exactly the same as in the original high-dimensional space.

Fast JL transforms $M = PHD$ satisfy the same distortion bounds and run in time $O(d_0 \log d_0 + d_1)$; the $k$-NN graph is unchanged if

$$\varepsilon < \frac{\gamma}{2D}. \tag{38}$$

### D.11 SIMPLEX VOLUME

A simplex in this context is the most elementary convex polytope determined by one "base" point together with a set of other points that do not all lie in a lower-dimensional subspace. In two dimensions it is a triangle, in three it is a tetrahedron, and in higher dimensions the generalization of those. Its volume quantifies the amount of space enclosed by those corner points. Because the projection acts almost as an exact length-preserver on each independent direction in the simplex, the total enclosed volume can only change by the small, precisely bounded factor implied by the near-isometry.

For $d + 1$ affinely independent points,

$$\mathrm{Vol}(\Delta) = \frac{1}{d!}\Big|\det[x_1 - x_0, \ldots, x_d - x_0]\Big|. \tag{39}$$

Since $M$ is an approximate isometry on this $d$-dimensional span, its singular values lie in $[\sqrt{1-\varepsilon}, \sqrt{1+\varepsilon}]$, giving

$$(1 - \varepsilon)^{d/2}\,\mathrm{Vol}(\Delta) \leq \mathrm{Vol}(\Delta') \leq (1 + \varepsilon)^{d/2}\,\mathrm{Vol}(\Delta), \tag{40}$$

hence squared volume changes by at most $(1 \pm \varepsilon)^d$.

## E    PROOF OF APPROXIMATION CAPABILITIES OF MANIFOLD RESIDUAL BLOCK

In this section, we prove that our MR block can approximate any linear layer to arbitrary precision, thereby providing a worst-case performance guarantee: if the MR block cannot improve over the linear layer then it can match its behavior arbitrarily closely, ensuring no loss in worst-case performance. In addition, rather than directly proving our formulation, we establish a more fundamental result

Table 11: Experimental results on different initialization methods in MR-CATE. "K." and "X." stands for Kaiming and Xavier initialization. "U." and "N." stands for uniform and normal distribution. The number following is the input parameter for initialization. The gray rows stand for our main setting.

| $k$ | Methods | Camelyon16 | | | NSCLC | | | RCC | | |
|---|---|---|---|---|---|---|---|---|---|---|
| | | AUC↑ | F1↑ | Acc.↑ | AUC↑ | F1↑ | Acc.↑ | AUC↑ | F1↑ | Acc.↑ |
| | K. N. 0 | $63.0_{16.1}$ | $49.5_{19.7}$ | $\mathbf{65.7}_{11.1}$ | $82.2_{6.2}$ | $75.0_{5.7}$ | $75.2_{5.5}$ | $98.0_{1.4}$ | $87.8_{1.8}$ | $90.1_{1.8}$ |
| | K. U. 0 | $62.4_{17.0}$ | $48.9_{16.9}$ | $63.7_{10.5}$ | $86.5_{4.4}$ | $79.4_{3.2}$ | $\mathbf{79.4}_{3.1}$ | $98.1_{1.4}$ | $88.2_{3.0}$ | $90.3_{2.5}$ |
| 4 | K. U. $\sqrt{5}$ | $62.1_{16.9}$ | $\mathbf{58.4}_{15.8}$ | $62.3_{13.5}$ | $\mathbf{87.0}_{3.9}$ | $79.4_{3.2}$ | $79.4_{3.2}$ | $\mathbf{98.2}_{1.2}$ | $\mathbf{88.7}_{2.0}$ | $\mathbf{90.5}_{1.8}$ |
| | X. N. 0 | $\mathbf{66.2}_{14.1}$ | $48.8_{20.1}$ | $64.7_{12.9}$ | $84.0_{5.3}$ | $76.3_{4.2}$ | $76.4_{4.2}$ | $98.1_{1.4}$ | $87.9_{3.4}$ | $90.1_{2.9}$ |
| | X. U. 0 | $61.1_{17.1}$ | $51.2_{17.8}$ | $63.6_{12.8}$ | $86.4_{4.6}$ | $79.1_{2.7}$ | $79.1_{2.6}$ | $98.1_{1.3}$ | $88.2_{2.9}$ | $90.2_{2.6}$ |
| | K. N. 0 | $\mathbf{89.5}_{13.8}$ | $78.1_{22.4}$ | $\mathbf{83.3}_{12.2}$ | $90.4_{5.9}$ | $82.1_{5.9}$ | $82.2_{5.9}$ | $98.5_{0.5}$ | $89.8_{2.1}$ | $91.1_{1.9}$ |
| | K. U. 0 | $87.4_{13.8}$ | $77.7_{20.9}$ | $82.8_{11.3}$ | $91.9_{4.8}$ | $82.9_{4.4}$ | $83.0_{4.3}$ | $98.5_{0.5}$ | $89.1_{2.0}$ | $90.7_{1.6}$ |
| 8 | K. U. $\sqrt{5}$ | $88.0_{14.8}$ | $\mathbf{81.7}_{16.0}$ | $82.9_{15.3}$ | $\mathbf{93.5}_{5.3}$ | $84.6_{5.7}$ | $84.7_{5.6}$ | $98.6_{0.3}$ | $89.1_{1.3}$ | $90.7_{1.0}$ |
| | X. N. 0 | $89.1_{13.2}$ | $75.9_{21.1}$ | $81.2_{11.0}$ | $92.1_{7.1}$ | $84.1_{6.7}$ | $84.2_{6.7}$ | $98.6_{0.4}$ | $\mathbf{90.3}_{1.1}$ | $\mathbf{91.5}_{1.0}$ |
| | X. U. 0 | $85.6_{14.7}$ | $76.2_{19.5}$ | $81.6_{11.1}$ | $\mathbf{93.5}_{4.7}$ | $\mathbf{85.0}_{5.3}$ | $\mathbf{85.0}_{5.2}$ | $\mathbf{98.7}_{0.4}$ | $90.0_{1.9}$ | $\mathbf{91.5}_{1.6}$ |
| | K. N. 0 | $93.8_{1.7}$ | $88.9_{4.6}$ | $89.6_{4.5}$ | $95.4_{2.2}$ | $86.9_{2.8}$ | $87.0_{2.8}$ | $98.6_{0.4}$ | $88.2_{2.2}$ | $89.6_{2.0}$ |
| | K. U. 0 | $92.2_{2.2}$ | $87.5_{1.9}$ | $88.4_{2.0}$ | $94.6_{2.3}$ | $86.3_{2.2}$ | $86.4_{2.1}$ | $98.6_{0.4}$ | $88.8_{2.3}$ | $90.9_{2.0}$ |
| 16 | K. U. $\sqrt{5}$ | $92.4_{3.2}$ | $86.7_{3.9}$ | $87.8_{3.4}$ | $\mathbf{96.1}_{1.4}$ | $88.6_{3.3}$ | $88.7_{3.3}$ | $98.6_{0.6}$ | $\mathbf{89.8}_{3.1}$ | $\mathbf{91.6}_{2.3}$ |
| | X. N. 0 | $\mathbf{94.5}_{1.5}$ | $\mathbf{90.1}_{1.5}$ | $\mathbf{90.7}_{1.6}$ | $95.6_{2.0}$ | $88.5_{1.4}$ | $88.5_{1.4}$ | $\mathbf{98.7}_{0.5}$ | $88.9_{2.2}$ | $90.9_{1.4}$ |
| | X. U. 0 | $92.7_{3.7}$ | $88.5_{3.5}$ | $89.1_{3.4}$ | $95.8_{1.3}$ | $\mathbf{88.7}_{3.1}$ | $\mathbf{88.8}_{3.1}$ | $\mathbf{98.7}_{0.4}$ | $89.5_{3.1}$ | $91.2_{2.5}$ |

Table 12: Experimental results on different pretrained feature extractors under various shot settings. "R50" stands for ResNet50, and "MR-" stands for MIL model with our MR block.

| $k$ | Methods | ABMIL | | | | | | CATE | | | | | |
|---|---|---|---|---|---|---|---|---|---|---|---|---|---|
| | | NSCLC | | | RCC | | | NSCLC | | | RCC | | |
| | | AUC↑ | F1↑ | Acc.↑ | AUC↑ | F1↑ | Acc.↑ | AUC↑ | F1↑ | Acc.↑ | AUC↑ | F1↑ | Acc.↑ |
| | R50 | $61.6_{2.6}$ | $53.5_{6.8}$ | $57.1_{3.6}$ | $74.7_{7.4}$ | $57.7_{7.5}$ | $61.1_{9.2}$ | $53.2_{8.2}$ | $40.3_{10.5}$ | $51.0_{3.3}$ | $77.8_{3.4}$ | $47.0_{5.3}$ | $61.7_{9.0}$ |
| 4 | MR-R50 | $62.3_{6.2}$ | $55.3_{11.7}$ | $59.0_{4.7}$ | $80.2_{5.9}$ | $63.1_{7.6}$ | $67.0_{6.6}$ | $51.4_{9.1}$ | $41.3_{11.4}$ | $51.4_{4.2}$ | $78.6_{2.7}$ | $48.2_{8.9}$ | $59.6_{14.0}$ |
| | UNI | $74.5_{8.5}$ | $68.5_{7.1}$ | $69.0_{7.0}$ | $96.6_{1.8}$ | $83.8_{4.3}$ | $86.1_{2.9}$ | $64.6_{8.6}$ | $46.6_{8.6}$ | $50.4_{4.5}$ | $88.5_{5.8}$ | $71.4_{9.0}$ | |
| | MR-UNI | $\mathbf{86.5}_{8.1}$ | $\mathbf{75.5}_{9.7}$ | $\mathbf{78.2}_{8.4}$ | $\mathbf{97.9}_{1.4}$ | $\mathbf{87.5}_{2.3}$ | $\mathbf{89.6}_{2.6}$ | $59.0_{11.5}$ | $49.7_{12.7}$ | $56.1_{9.7}$ | $92.1_{4.5}$ | $73.5_{9.7}$ | $76.9_{9.1}$ |
| | R50 | $59.0_{3.7}$ | $54.7_{8.1}$ | $57.0_{4.6}$ | $74.6_{6.9}$ | $56.6_{9.0}$ | $58.5_{10.7}$ | $64.9_{7.8}$ | $57.4_{10.6}$ | $60.0_{6.9}$ | $82.6_{11.0}$ | $64.2_{10.2}$ | $68.9_{7.0}$ |
| 8 | MR-R50 | $64.3_{4.2}$ | $59.7_{3.3}$ | $60.1_{3.2}$ | $80.8_{4.5}$ | $62.3_{6.7}$ | $63.9_{8.3}$ | $63.5_{6.4}$ | $60.7_{5.7}$ | $61.4_{5.0}$ | $89.5_{3.6}$ | $75.0_{7.2}$ | $76.6_{6.0}$ |
| | UNI | $80.8_{7.9}$ | $72.8_{6.2}$ | $73.1_{6.2}$ | $98.6_{0.4}$ | $88.7_{2.1}$ | $90.3_{1.8}$ | $82.7_{7.8}$ | $72.4_{5.1}$ | $73.0_{5.1}$ | $98.1_{0.6}$ | $87.8_{3.4}$ | $89.3_{3.0}$ |
| | MR-UNI | $\mathbf{91.3}_{5.0}$ | $\mathbf{81.7}_{4.4}$ | $\mathbf{81.8}_{4.3}$ | $\mathbf{98.6}_{0.6}$ | $\mathbf{91.2}_{1.8}$ | $\mathbf{92.3}_{1.4}$ | $\mathbf{88.8}_{6.6}$ | $\mathbf{80.6}_{6.8}$ | $\mathbf{80.8}_{6.7}$ | $\mathbf{98.5}_{0.4}$ | $\mathbf{90.8}_{1.4}$ | $\mathbf{91.8}_{1.1}$ |
| | R50 | $65.6_{9.9}$ | $62.3_{9.0}$ | $62.5_{8.8}$ | $84.0_{4.0}$ | $67.7_{3.8}$ | $70.9_{5.1}$ | $66.3_{10.1}$ | $56.2_{12.8}$ | $61.1_{7.4}$ | $90.0_{8.0}$ | $74.7_{13.2}$ | $78.0_{9.0}$ |
| 16 | MR-R50 | $67.2_{6.6}$ | $63.7_{5.0}$ | $63.9_{4.8}$ | $85.9_{2.1}$ | $70.8_{2.4}$ | $73.4_{1.8}$ | $66.3_{12.6}$ | $59.2_{15.1}$ | $63.0_{8.2}$ | $94.1_{4.2}$ | $81.4_{5.1}$ | $82.4_{5.6}$ |
| | UNI | $91.1_{2.7}$ | $83.6_{3.0}$ | $83.6_{3.0}$ | $98.9_{0.3}$ | $89.2_{1.9}$ | $91.0_{1.8}$ | $87.2_{7.5}$ | $75.8_{9.2}$ | $76.9_{7.7}$ | $98.0_{0.9}$ | $88.0_{2.5}$ | $90.3_{1.6}$ |
| | MR-UNI | $\mathbf{94.8}_{2.8}$ | $\mathbf{87.0}_{3.6}$ | $\mathbf{87.0}_{3.6}$ | $\mathbf{98.9}_{0.4}$ | $\mathbf{91.1}_{2.4}$ | $\mathbf{92.4}_{1.8}$ | $\mathbf{91.7}_{4.6}$ | $\mathbf{82.6}_{6.0}$ | $\mathbf{82.9}_{5.6}$ | $\mathbf{98.5}_{0.6}$ | $\mathbf{91.0}_{1.9}$ | $\mathbf{92.2}_{1.5}$ |

Table 13: Performance of ABMIL and CATE, with and without our MR block (MR-ABMIL and MR-CATE), under 32- and 64-shot settings on TCGA-NSCLC and TCGA-RCC datasets.

| $k$ | Methods | #P | NSCLC | | | RCC | | |
|---|---|---|---|---|---|---|---|---|
| | | | AUC↑ | F1↑ | Acc.↑ | AUC↑ | F1↑ | Acc.↑ |
| | ABMIL | 0.26 | $96.2_{1.9}$ | $89.7_{3.6}$ | $89.7_{3.6}$ | $97.6_{1.8}$ | $91.1_{2.0}$ | $93.1_{2.0}$ |
| 32 | **MR**-ABMIL | 0.18 | $\mathbf{98.1}_{0.4}$ | $\mathbf{93.1}_{1.7}$ | $\mathbf{93.1}_{1.7}$ | $\mathbf{99.5}_{0.3}$ | $\mathbf{93.4}_{0.6}$ | $\mathbf{94.9}_{0.6}$ |
| | CATE | 1.93 | $97.0_{0.4}$ | $91.1_{1.6}$ | $91.1_{1.6}$ | $99.3_{0.3}$ | $91.0_{2.0}$ | $93.1_{1.6}$ |
| | **MR**-CATE | 0.84 | $97.7_{0.5}$ | $92.1_{1.6}$ | $92.1_{1.6}$ | $\mathbf{99.6}_{0.1}$ | $92.9_{2.2}$ | $94.5_{1.9}$ |
| | ABMIL | 0.26 | $97.5_{0.9}$ | $91.5_{1.9}$ | $91.5_{1.9}$ | $99.0_{0.6}$ | $91.6_{2.9}$ | $93.2_{2.5}$ |
| 64 | **MR**-ABMIL | 0.18 | $\mathbf{98.2}_{0.3}$ | $\mathbf{92.8}_{1.2}$ | $\mathbf{92.8}_{1.2}$ | $99.6_{0.2}$ | $94.3_{1.1}$ | $95.6_{1.4}$ |
| | CATE | 1.93 | $97.7_{0.5}$ | $91.1_{1.4}$ | $91.1_{1.4}$ | $99.4_{0.2}$ | $93.6_{1.1}$ | $95.0_{0.8}$ |
| | **MR**-CATE | 0.84 | $97.9_{0.1}$ | $92.4_{1.1}$ | $92.4_{1.1}$ | $\mathbf{99.8}_{0.1}$ | $\mathbf{95.3}_{1.0}$ | $\mathbf{96.6}_{0.8}$ |

without activation functions. Subsequently, with the universal approximation theorem (Barron, 1993; Cybenko, 1989; Funahashi, 1989; Hornik et al., 1989; Lu & Lu, 2020), our architecture with nonlinear activation function can achieve even superior approximation precision.

**Theorem E.1** (Universal Approximation by Manifold Residual Block). *Let $\boldsymbol{A}_* \in \mathbb{R}^{d_0 \times d_1}$ have full rank $r_A = \min\{d_0, d_1\}$. Draw a frozen matrix $\boldsymbol{B} \in \mathbb{R}^{d_0 \times d_1}$ with sub-Gaussian entries. Then almost*

*surely: for every $\varepsilon > 0$ there exist integers $r \leq r_A$ and matrices $\boldsymbol{W}_2$ and $\boldsymbol{W}_1$ such that,*

$$\|\boldsymbol{A}_* - (\boldsymbol{B} + \boldsymbol{W}_2 \boldsymbol{W}_1)\|_F \leq \varepsilon, \text{ where } \boldsymbol{W}_2 \in \mathbb{R}^{d_0 \times r}, \text{ and } \boldsymbol{W}_1 \in \mathbb{R}^{r \times d_1}. \tag{41}$$

*Proof.* **Full-Rank Guarantees and Singular-Value Bounds.** We prove that $\boldsymbol{B}$ is full rank almost surely in Sec. F.1. Let $\delta, \eta \in (0, 1)$, according to Johnson–Lindenstrauss Lemma, we have,

$$(1-\delta)\|\boldsymbol{x}\|_2 \ \leq \ \tfrac{1}{\sqrt{d_0}}\|\boldsymbol{B}\,\boldsymbol{x}\|_2 \ \leq \ (1+\delta)\|\boldsymbol{x}\|_2 \implies \forall \sigma(\boldsymbol{B}) \in \ [\sqrt{d_0}\,(1-\delta),\ \sqrt{d_0}\,(1+\delta)], \tag{42}$$

with probability at least $1 - \eta$, if the input dimension satisfies $d_0 \ \geq \ C(d_1 + \ln(1/\eta))/\delta^2$.

**Truncated SVD of the Residual.** Let $\boldsymbol{E} = \boldsymbol{A}_* - \boldsymbol{B}$. Its singular value decomposition is given as $\boldsymbol{E} = \boldsymbol{U}\,\boldsymbol{\Sigma}\,\boldsymbol{V}^\top$. By the Eckart–Young–Mirsky theorem, we choose $r \leq r_A$ and we have,

$$\|\boldsymbol{E} - \boldsymbol{E}_r\|_F = \sqrt{\sum_{i=r+1}^{\text{rank}(\boldsymbol{E})} \sigma_i^2(\boldsymbol{E})} \leq \varepsilon, \text{ where } \boldsymbol{E}_r = \sum_{i=1}^{r} \sigma_i\,\boldsymbol{u}_i\,\boldsymbol{v}_i^\top. \tag{43}$$

**Low-rank Correction Construction.** From the last step, we can set $\boldsymbol{W}_1$ and $\boldsymbol{W}_2$ as,

$$\boldsymbol{W}_2 = [\,\boldsymbol{u}_1, \ldots, \boldsymbol{u}_r\,]\,\boldsymbol{\Sigma}_r^{1/2}, \quad \boldsymbol{W}_1 = \boldsymbol{\Sigma}_r^{1/2}\,[\,\boldsymbol{v}_1, \ldots, \boldsymbol{v}_r\,]^\top, \text{ where } \boldsymbol{\Sigma}_r = \text{diag}(\sigma_1, \ldots, \sigma_r), \tag{44}$$

which satisfies $\boldsymbol{W}_2\boldsymbol{W}_1 = \boldsymbol{E}_r$ and therefore, for sufficiently large $r$, we have

$$\|\boldsymbol{A}_* - (\boldsymbol{B} + \boldsymbol{W}_2\boldsymbol{W}_1)\|_F = \|\boldsymbol{E} - \boldsymbol{E}_r\|_F < \varepsilon. \tag{45}$$

$\square$

# F    ADDITIONAL PROOFS

## F.1    RANDOMLY INITIALIZED MATRIX HAS FULL RANK

**Theorem F.1.** *Let $\boldsymbol{W} \in \mathbb{R}^{d_0 \times d_1}$ have entries drawn i.i.d. from a continuous distribution (e.g. Kaiming–uniform with gain $\sqrt{5}$). Then with probability one,*

$$\text{rank}(\boldsymbol{W}) \ = \ \min\{d_0, d_1\}. \tag{46}$$

*Proof.* Set $r = \min\{d_0, d_1\}$. Define the singular set

$$S = \{\boldsymbol{W} : \text{rank}(\boldsymbol{W}) < r\}. \tag{47}$$

A matrix in $S$ must make every $r \times r$ minor vanish. Label those minors by index sets $I$, and write

$$Z_I = \{\boldsymbol{W} : \det_I(\boldsymbol{W}) = 0\}. \tag{48}$$

Each $\det_I$ is a nonzero polynomial in the entries of $\boldsymbol{W}$, so by basic measure theory its zero set $Z_I$ has Lebesgue measure zero. Since there are finitely many $I$,

$$S = \bigcup_I Z_I \tag{49}$$

is a finite union of null sets, hence also null. Finally, the distribution on entries is absolutely continuous with respect to Lebesgue measure, so

$$\Pr(\boldsymbol{W} \in S) = 0, \tag{50}$$

and therefore $\boldsymbol{W}$ has full rank $r$ almost surely. $\square$

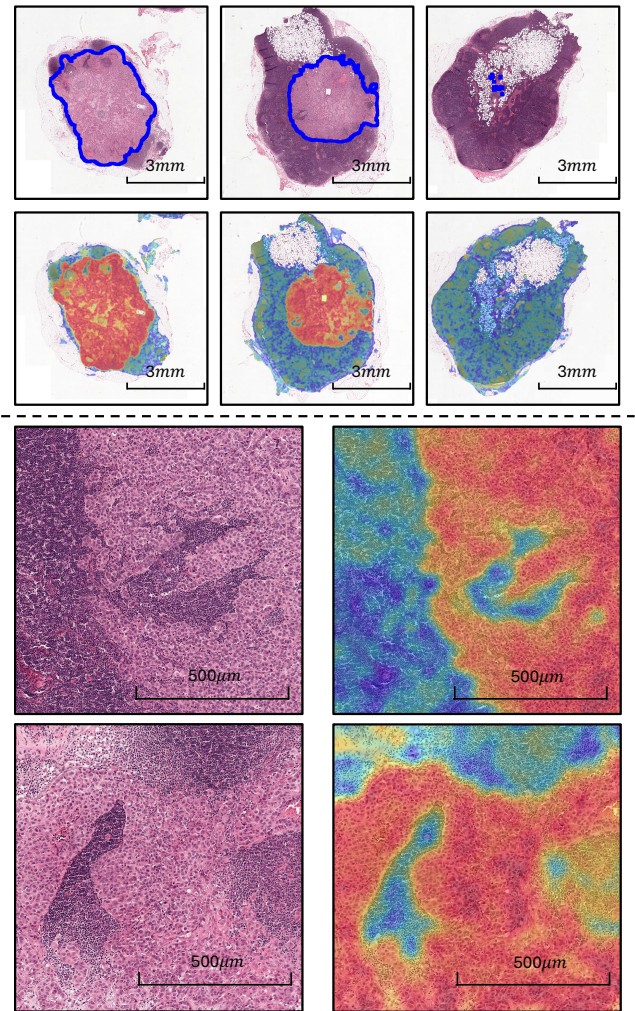

Figure 6: MR-CATE-generated heatmaps for tumor058 (Camelyon16). The left panel shows heatmaps for three biopsy regions, while the right panel displays fine-grained patches with their heatmaps. Red indicates high attention (tumor), whereas blue indicates low attention (normal tissue).

### F.2    MAXIMUM RANK OF THE LRP

**Theorem F.2** (Rank of a Low-Rank Product). *Let $W_2 \in \mathbb{R}^{m \times r}$ and $W_1 \in \mathbb{R}^{r \times n}$ with $r < \min(m, n)$. Then*

$$\mathrm{rank}\big(W_2 W_1\big) \leq r. \tag{51}$$

*Proof.* Observe that

$$\mathrm{rank}(W_2) \leq r, \qquad \mathrm{rank}(W_1) \leq r, \tag{52}$$

since $W_2$ has only $r$ columns and $W_1$ has only $r$ rows. A standard rank-inequality for matrix products states

$$\mathrm{rank}(AB) \leq \min\big(\mathrm{rank}(A), \mathrm{rank}(B)\big) \tag{53}$$

for any conformable $A, B$. Applying this with $A = W_2$ and $B = W_1$ gives

$$\mathrm{rank}\big(W_2 W_1\big) \leq \min\big(\mathrm{rank}(W_2), \mathrm{rank}(W_1)\big) \leq r. \tag{54}$$

$\square$

# G EXPERIMENT SETUPS

## G.1 DATASET DESCRIPTIONS

**Camelyon16** (Litjens et al., 2018) comprises 397 WSIs of lymph node sections from breast cancer patients, annotated for metastatic presence. The dataset is officially partitioned into 268 training specimens (157 normal, 111 tumor-containing) and 129 testing specimens.

**TCGA-RCC**[1] encompasses 940 WSIs from three distinct renal cell carcinoma subtypes: Kidney Chromophobe (TCGA-KICH, 121 WSIs from 109 cases), Kidney Renal Clear Cell Carcinoma (TCGA-KIRC, 519 WSIs from 513 cases), and Kidney Renal Papillary Cell Carcinoma (TCGA-KIRP, 300 WSIs from 276 cases).

**TCGA-NSCLC** consists of 1,053 WSIs representing two major lung cancer histological subtypes: Lung Squamous Cell Carcinoma (TCGA-LUSC, 512 WSIs from 478 cases) and Lung Adenocarcinoma (TCGA-LUAD, 541 WSIs from 478 cases).

**Boehmk** (Boehm et al., 2022) is a treatment-response cohort of high-grade serous ovarian cancer patients, with H&E-stained WSIs and associated progression-free survival (PFS) metadata. We frame this as a binary response prediction task using a 12-month landmark analysis: patients with PFS $\geq$ 12 months are labeled as responders, those with documented progression before 12 months as non-responders, and patients censored before 12 months are excluded.

**Trastuzumab** (Farahmand et al., 2022) is a HER2-positive breast cancer cohort comprising pre-treatment H&E tumor WSIs and clinical metadata describing response to trastuzumab-based therapy. We treat this as an intrinsically few-shot, slide-level treatment-response task by assigning each patient a binary label (responder vs. non-responder) based on the curated clinical "Responder" status.

## G.2 EXPERIMENTAL PROTOCOL AND HYPER-PARAMETER SETTING

**Experimental Protocol.** We benchmark our approach on established multiple instance learning frameworks: AB-MIL (Ilse et al., 2018) (attention-based method), TransMIL (Shao et al., 2021) (Transformer-based method), CATE (Huang et al., 2024) (information-theory-based method), DGR-MIL (Zhu et al., 2024) (density modeling method), and RRTMIL (Tang et al., 2024) (re-embedding method). Regularization techniques such as dropout, weight decay are used exactly as in the official implementations. In addition, we also compare our methods with ViLaMIL (Shi et al., 2024), the recent visual language models specifically proposed for few-shot WSI classification. All implementations utilize their official codes. We employ early stopping with a patience of 20 epochs on validation loss, while enforcing a minimum training duration of 50 epochs. For our experiments, we adhere to the official training and test partitions for the Camelyon16 dataset. We further split the official training set into new training and validation sets using a 70/30 ratio. For the rest of the datasets, we perform our own random split, partitioning the data into training, validation, and test sets with a 60/20/20 ratio. From these constructed training sets, we then create our few-shot scenarios by randomly sampling $k$ WSIs per class to create five different subsets for each fold of our cross-validation. The average performance and standard deviation across these five runs is reported.

**Hyper-parameters Details.** To rigorously assess the universal, plug-and-play value of our block, we adopt a strict paired-comparison protocol under a unified hyperparameter setting. For each baseline architecture, we evaluate its original and MR-enhanced versions while keeping all other conditions identical, ensuring the observed performance gain is solely attributable to our block. We use the AdamW (Loshchilov & Hutter, 2019) optimizer with learning rate and weight decay being $5 \times 10^{-4}$ and $10^{-5}$. The linear scheduler is utilized for our methods with starting factor and end factor being 0.01 and 0.1, respectively. Dropout is set to 0.25. We use CONCH (Lu et al., 2024), UNI (Chen et al., 2024) and ResNet-50 (He et al., 2016) to extract features from the non-overlapping $224 \times 224$ patches, obtained from $20\times$ magnification of the WSIs, and the resulting feature dimensions are 512, 1,024 and 1,024, respectively. The CONCH features are utilized throughout our experiments as they demonstrate superior performance to the other two features. The UNI and ResNet-50 features are only utilized for investigating the robustness of our method against features from different foundation

---

[1]The TCGA data used in our work is available in https://portal.gdc.cancer.gov.

models. The rank $r$ is set to $64$. GELU (Hendrycks & Gimpel, 2016) is chosen as it yields the best performance among the alternatives.

### G.3 COMPUTER RESOURCES

We conduct our experiments on an NVIDIA A100 GPU and Intel(R) Xeon(R) Silver 4410Y CPU, with Ubuntu system version 22.04.5 LTS (GNU/Linux 5.15.0-136-generic x86_64). More detailed Python environment would be released with the code.

### G.4 REPLACEMENT POSITION OF THE MODELS

**Preliminaries of ABMIL.** ABMIL can be formulated as follows,

$$a_k = \frac{\exp\left\{\boldsymbol{w}^\top\left(\tanh(\boldsymbol{V}\boldsymbol{h}_k^\top)\odot\mathrm{sigm}(\boldsymbol{U}\boldsymbol{h}_k^\top)\right)\right\}}{\sum_{j=1}^K \exp\left\{\boldsymbol{w}^\top\left(\tanh(\boldsymbol{V}\boldsymbol{h}_j^\top)\odot\mathrm{sigm}(\boldsymbol{U}\boldsymbol{h}_j^\top)\right)\right\}}, \qquad \boldsymbol{z} = \sum_{k=1}^K a_k\boldsymbol{h}_k, \qquad (55)$$

where $\boldsymbol{z}$ is the slide-level feature.

ABMIL: replace both the $\boldsymbol{V}$ and $\boldsymbol{U}$ matrix with our MR block.

TransMIL[2]: only replace the out matrix in Nyström attention implementation with our MR block.

CATE[3]: replace all linear layers in `x_linear`, `interv_linear`, feature output linear layer and the second linear layer in `encoder_IB` with our MR block.

DGRMIL[4]: replace linear layers in `self.fc1` of `optimizer_triple` and `self.crossffn` of `DGRMIL`.

RRTMIL[5]: replace linear layer in `self.proj` of `InnerAttention`.

More details will be available along with the code.

### G.5 COMPUTATION CONSUMPTION OF MR BLOCK

In this section, we present FLOPs and wall-clock inference time for the MR-enhanced backbones under different bag sizes and feature dimensions in Table 14.

## H ADDITIONAL EXPERIMENTAL RESULTS

The additional experimental results are presented in Tables 10 to 12. The discussions of these tables are already available in Sec. 4.

## I LIMITATIONS AND FUTURE WORK

This work introduces an MR block to mitigate the manifold degradation of the vanilla linear layer while enabling parameter-efficient task-specific adaptation. However, our empirical evaluation has several limitations. First, we fix the rank $r$ across all layers, a choice that is unlikely to be optimal in real-world scenarios. A more comprehensive exploration of different rank values within a single model would better characterize the trade-off between model size and accuracy. Second, our study of insertion positions is limited to the ABMIL architecture; to establish broader applicability, the MR block should also be tested in more sophisticated MIL frameworks such as TransMIL and CATE. Third, our geometric anchor now is frozen and does not contain any dataset-level information. An interesting extension is to replace the random anchor with a data-driven PCA projection, at the cost of dataset-specific precomputation for a more comprehensive utilization of dataset-level information. Fourth, we do not evaluate MR on top of slide-level pathology foundation models

---

[2]https://github.com/szc19990412/TransMIL
[3]https://github.com/HKU-MedAI/CATE
[4]https://github.com/ChongQingNoSubway/DGR-MIL
[5]https://github.com/DearCaat/RRT-MIL

| Model | Bag Size | 512-d features (CONCH) | | 1024-d features (UNI/ResNet50) | |
|---|---|---|---|---|---|
| | | FLOPs (G) | Time (ms) | FLOPs (G) | Time (ms) |
| ABMIL | 100 | 0.053 | 0.90 | 0.420 | 1.44 |
| | 1000 | 0.525 | 2.66 | 4.197 | 2.73 |
| | 10000 | 5.251 | 2.78 | 41.974 | 3.37 |
| CATE | 100 | 0.333 | 2.69 | 1.541 | 2.43 |
| | 1000 | 3.328 | 3.12 | 15.410 | 3.12 |
| | 10000 | 33.283 | 10.36 | 154.102 | 10.05 |
| TransMIL | 100 | 1.153 | 12.33 | 4.822 | 12.27 |
| | 1000 | 6.076 | 16.25 | 26.403 | 15.66 |
| | 10000 | 49.832 | 46.23 | 220.299 | 44.54 |
| DGRMIL | 100 | 1.332 | 12.19 | 5.560 | 12.18 |
| | 1000 | 4.862 | 12.75 | 21.566 | 12.77 |
| | 10000 | 43.431 | 31.36 | 194.718 | 31.59 |
| RRTMIL | 100 | 1.008 | 3.01 | 4.243 | 3.13 |
| | 1000 | 3.227 | 3.52 | 15.006 | 5.92 |
| | 10000 | 30.286 | 13.04 | 142.116 | 46.47 |

Table 14: FLOPs and inference time for different MIL models under varying bag sizes and feature dimensions.

such as TITAN (Ding et al., 2025), because this would require a non-trivial architectural redesign (MR cannot simply replace the tiny slide-level classifier, and TITAN is tightly coupled to CONCH v1.5 while our experiments span CONCH, UNI, and ResNet50). In addition, our geometric analysis relies on approximately 30,000 patch-level features per configuration, whereas TITAN only provides slide-level embeddings per WSI, so directly reusing the same analysis would yield much less stable statistics. We treat these limitations as future directions of our work.

We anticipate that the proposed MR block will broadly advance few-shot learning within computational pathology and beyond. Its minimal deployment requirements render it readily applicable in a wide range of settings. In computational pathology, it may enhance tasks such as few-shot report generation, survival outcome prediction, mutation inference, and treatment-response modeling, among others. More generally, the MR block could improve performance in computer vision applications, including few-shot segmentation and classification, and may even be adapted to natural-language-processing tasks. Further exploration across these domains is therefore warranted.

