# OpenReview forum: "Exploiting Low-Dimensional Manifold of Features for Few-Shot Whole Slide Image Classification"
_ICLR.cc/2026/Conference — ICLR 2026 Poster_

### Official Review · Reviewer_7Ghg · 2025-10-26

**Soundness:** 3
**Presentation:** 3
**Contribution:** 3
**Rating:** 6
**Confidence:** 4

**Summary:**

This paper identifies a novel cause for overfitting in few-shot WSI classification: the distortion of low-dimensional feature manifolds by standard linear layers . The authors propose a plug-and-play "Manifold Residual (MR) block" that replaces these layers, using a fixed random matrix as a "geometric anchor" to preserve manifold structure and a separate low-rank pathway for task adaptation

**Strengths:**

- The paper is built on a strong, clear, and insightful hypothesis. The diagnosis of overfitting as a geometric problem (i.e., manifold distortion by geometry-agnostic layers) rather than purely a data-scarcity problem is a novel and compelling contribution to the field.

- The core hypothesis is well-supported by quantitative analysis before the method is introduced. The use of spectral analysis to show low effective rank (Fig. 1) and tangent space analysis to demonstrate both the manifold's curvature and its distortion by standard linear layers (Fig. 1) provides a solid and convincing foundation for the proposed solution.

**Weaknesses:**

- Limited evaluation datasets. While many MIL methods are tested, the number of different types of tasks (classification only) and number of organs (limited to 3) is quite low for demonstrating a robust method improvement.
- The tasks are also artificial few shot tasks. These tasks (e.g., NSCLC subtyping) have 1000s of data points, but the few shot splits are artificially sampled. I recommend trying some real few shot tasks, such as treatment response prediction. This type of task will always be few shot in nature and helping improve performance in this domain will carry tremendous benefit for the field, which is not true for a rather solved task of RCC and NSCLC subtyping. 10+ treatment response prediction tasks can be found at: https://huggingface.co/datasets/MahmoodLab/Patho-Bench

**Questions:**

- Why was $r=64$ used for the main comparison tables, while it is shown that performance saturates at $r=32$ (Fig. 3)? Does this choice, which doubles the parameters of the LRP, potentially understate the true parameter efficiency and performance of the MR block at its optimal rank? It may be useful to report the $k=16$ results for MR-CATE with $r=32$ in the text to show that the SOTA performance holds at this more theoretically-motivated rank.
- Is a random matrix the optimal choice for preserving the specific, learned structure of a foundation model's feature manifold? For instance, would a fixed projection based on the principal components (PCA) of the training set features serve as a more "informed" (but still fixed) geometric anchor?

---

> ### Author Response · Authors · 2025-11-24
>
> We thank the reviewer for the positive evaluation, in particular for highlighting that our paper is built on a strong, clear, and insightful hypothesis and that the experimental evidence is convincing. We address the weaknesses and questions in detail below.
>
> ---
>
> ### W1 & W2: Treatment response datasets and artificial few-shot setting
>
> We fully agree that evaluating methods on inherently few-shot treatment response tasks is important for clinical impact. At the same time, using artificially sampled $k$-shot splits is a standard protocol in the few-shot learning literature and in recent few-shot WSI works (for example TOP, FAST, FOCUS). This protocol allows us to control the supervision level across datasets and backbones, so that changes in performance can be attributed to the model rather than to heterogeneous dataset sizes. Our use of artificial $k$-shot splits on Camelyon16, NSCLC, and RCC therefore follows this established practice and keeps the evaluation setting controlled across datasets.
>
> In response to the suggestion from the reviewer about using treatment response tasks from Patho-Bench, we additionally evaluated our method on two real treatment response datasets: Boehmk (ovarian cancer) and Trastuzumab (breast cancer). Both datasets are intrinsically small, and we still apply the same $k$-shot protocol to keep the setting consistent with our main experiments. These treatment response tasks are substantially more challenging than subtype classification due to noisier signals, so the absolute numbers are lower for all methods. Nevertheless, our MR block consistently improves over the vanilla ABMIL baseline in AUC and accuracy and typically also in F1:
>
> | Model | Shots |  | Boehmk |  |  | Trastuzumab |  |
> |----|-----|----|----|----|----|----|----|
> |  |  | AUC | F1 | ACC | AUC | F1 | ACC |
> | ABMIL | 4 | 56.2±6.7 | 45.9±8.6 | 49.7±10.8 | 47.7±5.6 | 40.3±3.9 | 42.4±2.6 |
> | MR-ABMIL |  | **65.4±11.3** | **56.9±8.5** | **65.1±3.7** | **56.9±2.3** | **40.9±8.7** | **58.8±0.0** |
> | ABMIL | 8 | 53.4±5.1 | 46.1±4.6 | 50.9±8.2 | 49.4±3.7 | 48.8±10.8 | 54.1±7.7 |
> | MR-ABMIL |  | **63.4±9.5** | **57.6±7.1** | **62.3±9.8** | **55.1±4.4** | 40.3±4.5 | **54.1±6.4** |
> | ABMIL | 16 | 52.6±5.5 | 46.9±7.3 | 50.3±8.0 | 46.3±10.0 | 36.1±1.3 | 48.2±6.4 |
> | MR-ABMIL |  | **63.5±4.1** | **55.1±3.4** | **63.4±6.5** | **57.7±0.8** | **43.6±3.3** | **56.5±3.2** |
>
> These additional results show that the geometric overfitting issue we identify is not restricted to subtype classification on large cohorts, but also appears in realistic inherently few-shot treatment response settings, and that the MR block remains beneficial in these more challenging tasks. In the revision, we will add this table to the appendix.

---

> ### Author Response · Authors · 2025-11-24
>
> ### Q1. Choice of rank: why use $r = 64$ in the main tables
>
> We thank the reviewer for pointing this out and for giving us the opportunity to clarify our choice of rank.
>
> Our analysis and experiments show that the performance of MR-ABMIL (Fig. 3) saturates around $r = 32$ and then varies only mildly for larger $r$, while the performance of MR-CATE (Fig. 6 in the Appendix, cited in Sec. 4.3) still slowly grows after $r = 32$. The statement in the main text that MR is stable once $r$ exceeds the effective rank is therefore based on these empirical results.
>
> Given these observations, we chose to use $r = 64$ in the main comparison tables for the following reasons:
>
> - **Fairness and simplicity**. Using one shared $r$ across datasets and backbones avoids per configuration tuning and makes the comparison cleaner and easier to reproduce.
> - **Parameter efficiency still holds**. $r < \frac{d_0 d_1}{d_0 + d_1}$ still holds for all replaced linear layers, which guarantees that MR always uses fewer trainable parameters than the corresponding full linear layer.
> - **Heavier models may benefit from a slightly larger rank**. For more expressive backbones such as CATE, the sensitivity curves show a small but monotonic improvement as $r$ increases beyond 32, so choosing $r = 64$ is a conservative global setting that avoids under tuning these models while remaining parameter efficient.
>
> The MR-CATE results with $r$ ranging from 4 to 128, including the $k = 16, r = 32$ setting mentioned by the reviewer, are already included in the appendix and cited in Sec. 4.3. The performance remains essentially SOTA, with only minor fluctuations compared to $r = 64$. In the revision, we will highlight this pointer more clearly in the main text so that readers can easily see the comparison at $r = 32$.
>
> In practice, when applying MR to a specific model, practitioners are free to tune $r$ and to choose which linear layers to replace. We will make this explicit in Sec. 4.3.
>
> ---
>
> ### Q2. Random matrix vs PCA-based anchor
>
> We appreciate the suggestion of using a PCA based projection as a more informed geometric anchor.
>
> We do not claim that a random matrix is the optimal anchor, but rather that it is a simple choice that is sufficient for the goals of this paper. In this work we focus on testing whether a very simple fixed anchor, combined with the MR block, is already enough to better exploit the feature manifold of a pathology foundation model. Random projection theory gives high probability guarantees that a random linear map approximately preserves pairwise distances and local tangent spaces of a smooth manifold, without any data-dependent precomputation. This makes a random matrix a convenient and intuitive geometry-preserving anchor in our setting and keeps the design entirely plug-and-play.
>
> A PCA-based anchor is indeed an interesting and potentially more powerful alternative, but it would require computing and freezing PCA for each dataset (and possibly each layer or each model architecture), and recomputing it when changing feature extractors, backbones, or label spaces, which ties the anchor closely to specific training setups, in a similar way as TITAN is tied to CONCH v1.5. In contrast, the same random anchor can be reused unchanged across datasets and backbones, and empirically Figs. 1, 4, and 5 already show that it preserves cluster structure and curvature well enough to yield consistent gains.
>
> In the revision we will add a short discussion in the appendix mentioning PCA anchors as a promising extension and a natural direction for future work, including a systematic comparison between random and data-driven anchors.

---

### Official Review · Reviewer_k14d · 2025-10-27

**Soundness:** 3
**Presentation:** 3
**Contribution:** 3
**Rating:** 10
**Confidence:** 5

**Summary:**

The study proposes **Manifold Residual block** to address the issue of overfitting in few-shot whole slide images (WSI) classification. It argues that overfitting not just from data scarcity but a fundamentally geometric problem. Features from pathology foundation models lie on a low-dimensional, nonlinear manifold that linear layers in MIL models distort.

**Strengths:**

1) The study provides quantitative proof that CONCH features exhibit a low-dimensional manifold with nonlinear curvature, which linear layers disrupt.

2) The study proposes **MR Block Innovation** with a fixed random geometric anchor and a trainable low-rank residual pathway, reducing overfitting and parameter count.

3) The study provides a extensive validation to demonstrates the generalization of the proposed method.

4) **MR Block Innovation** demonstrates SOTA performances on three datasets across 4, 8, and 16 shots settings.

**Weaknesses:**

1) The study does not provide comparison with SOTA methods for whole slide images classification in few-shot settings such as MGPATH [3], MSCPT [2], FOCUS [3].

2) The study does not report inference time and FLOPs for the proposed method.

3) The study does not fully explain the effective of rank on the model's performance. For example, the sensitivity analysis (Fig. 3) shows that performance saturates around a rank of **r=32**. The authors note this **aligns remarkably** with the features effective rank of 29.7. However, all main experiments in Table 1 and the ablation studies in Table 2 were run with **r=64**. In Fig. 3, NSCLC 8-shots, **r=64** performs worse than **r=32** or **r=48**, suggesting **r=64** may be a suboptimal.

4) The study lacks a clear description of how to apply **MR** block to complex methods such as TransMIL or CATE.

**Reference**:

1. Nguyen, A.-T., Nguyen, D. M. H., Diep, N. T., Nguyen, T. Q., Ho, N., Metsch, J. M., Maurer, M. C., Sonntag, D., Bohnenberger, H., & Hauschild, A.-C. (2025). MGPATH: A vision-language model with multi-granular prompt learning for few-shot whole-slide pathology classification. Transactions on Machine Learning Research (2025).

2. Han, Minghao, et al. "Mscpt: Few-shot whole slide image classification with multi-scale and context-focused prompt tuning." IEEE Transactions on Medical Imaging (2025).

3. Guo, Zhengrui, et al. "Focus: Knowledge-enhanced adaptive visual compression for few-shot whole slide image classification." Proceedings of the Computer Vision and Pattern Recognition Conference. 2025.

**Questions:**

1) how does the MR block perform if B is a non-random, fixed matrix, such as an identify matrix?

2) can you confirm if the same geometric distortion problem exists for pathology slide-level foundation models such as TITAN [1]?

3) Given this strong evidence for **r=32**, why were all main experiments (Table 1) and ablation studies (Table 2) run with **r=64**?

4) Could you elaborate on the methodology used to apply **MR** block to TransMIL and CATE?

**Reference**:

1. Ding, T., et al. "Multimodal whole slide foundation model for pathology (2024)." URL https://arxiv. org/abs/2411.19666 2411.

---

> ### Author Response · Authors · 2025-11-24
>
> We thank the reviewer for the very positive and detailed evaluation and for the strong accept recommendation. We are glad that you found both the geometric analysis and the MR block design valuable. Below we respond to each weakness and question point by point.
>
> ---
>
> ### W1. Comparison with SOTA few-shot WSI methods (MGPATH, MSCPT, FOCUS)
>
> We appreciate the reviewer highlighting MGPATH, MSCPT, and FOCUS, which are indeed important recent works on few-shot WSI classification.
>
> Our work is deliberately confined to a simple and widely-used setting: we use a frozen pathology foundation model as feature extractor, only slide-level labels, and we train only the MIL-style models on top of these features, without any extra image–text pretraining or patch-level supervision. Within this regime we study how standard linear layers in existing MIL backbones distort the low-dimensional manifold structure of PFM features, and we redesign these linear layers using MR blocks to be geometry-aware and parameter-efficient.
>
> In contrast, MGPATH builds a VLM by training visual and text adaptors on about 923k pathology image–text pairs on top of the Prov-GigaPath foundation model, and then learns multi-granular prompts and a graph aggregation module on this VLM, which leverages much more data than our current setting and goes beyond our frozen PFM plus MIL-head setting. MSCPT and FOCUS keep the underlying VLM or FM encoders frozen and also use only slide-level labels, so their supervision regime matches ours. Our contribution is orthogonal: under the stricter regime of frozen PFMs and standard MIL backbones, we show that simply replacing internal linear projections in these heads by MR blocks already yields consistent few-shot gains.
>
> Since MGPATH explicitly trains a new VLM using GigaPath, which requires substantial additional pretraining beyond our current setting, we do not include it as a baseline. Implementing MSCPT within our CONCH-based pipeline would require substantial engineering of the VLM and graph prompt modules, so in this rebuttal we focus on FOCUS as the closest method that can be directly evaluated under our setting. The results are given below.
>
> | Model | Shots |  | Camelyon16 |  |  | NSCLC |  |  | RCC |  |
> |----|-----|----|----|----|----|----|----|----|----|----|
> |  |  | AUC | F1 | ACC | AUC | F1 | ACC | AUC | F1 | ACC |
> | FOCUS | 4 | 63.3±16.5 | 56.4±18.2 | 68.7±9.0 | 87.9±2.6 | 79.4±2.2 | 79.5±2.1 | 97.7±2.1 | 87.2±2.8 | 89.5±2.7 |
> | FOCUS | 8 | 87.2±7.0 | 79.2±9.4 | 80.2±9.4 | 93.7±5.6 | 85.0±5.8 | 85.1±5.8 | 98.1±0.6 | 86.7±2.3 | 88.9±2.0 |
> | FOCUS | 16 | 93.2±2.5 | 89.0±3.0 | 89.8±3.0 | 96.5±1.5 | 89.3±2.7 | 89.3±2.7 | 98.2±0.6 | 89.0±2.1 | 90.6±1.6 |
>
> In comparison to our method, FOCUS can outperform MR enhanced MIL backbones in some settings, but mostly our MR enhanced MIL backbones have superior or comparable performance to FOCUS. Importantly, in our implementation FOCUS uses 86.9M trainable parameters, while our MR variants achieve competitive or superior performance with significantly fewer trainable parameters, highlighting the parameter efficiency of the proposed MR block. We will add FOCUS as a baseline in the main text.

---

> ### Author Response · Authors · 2025-11-24
>
> ### W2. Inference time and FLOPs of the MR block
>
> We thank the reviewer for pointing this out. The tables below report FLOPs and wall clock inference time for the MR-enhanced backbones under different bag sizes and feature dimensions.
>
> ##### Feature Dimension: 512 (CONCH)
>
> | Model    | Bag Size | FLOPs (G) | Time (ms) |
> |----------|----------|-----------|-----------|
> | ABMIL    | 100      |     0.053 |      0.90 |
> | ABMIL    | 1000     |     0.525 |      2.66 |
> | ABMIL    | 10000    |     5.251 |      2.78 |
> | CATE     | 100      |     0.333 |      2.69 |
> | CATE     | 1000     |     3.328 |      3.12 |
> | CATE     | 10000    |    33.283 |     10.36 |
> | TransMIL | 100      |     1.153 |     12.33 |
> | TransMIL | 1000     |     6.076 |     16.25 |
> | TransMIL | 10000    |    49.832 |     46.23 |
> | DGRMIL   | 100      |     1.332 |     12.19 |
> | DGRMIL   | 1000     |     4.862 |     12.75 |
> | DGRMIL   | 10000    |    43.431 |     31.36 |
> | RRTMIL   | 100      |     1.008 |      3.01 |
> | RRTMIL   | 1000     |     3.227 |      3.52 |
> | RRTMIL   | 10000    |    30.286 |     13.04 |
>
> ##### Feature Dimension: 1024 (UNI/ResNet50)
>
> | Model    | Bag Size | FLOPs (G) | Time (ms) |
> |----------|----------|-----------|-----------|
> | ABMIL    | 100      |     0.420 |      1.44 |
> | ABMIL    | 1000     |     4.197 |      2.73 |
> | ABMIL    | 10000    |    41.974 |      3.37 |
> | CATE     | 100      |     1.541 |      2.43 |
> | CATE     | 1000     |    15.410 |      3.12 |
> | CATE     | 10000    |   154.102 |     10.05 |
> | TransMIL | 100      |     4.822 |     12.27 |
> | TransMIL | 1000     |    26.403 |     15.66 |
> | TransMIL | 10000    |   220.299 |     44.54 |
> | DGRMIL   | 100      |     5.560 |     12.18 |
> | DGRMIL   | 1000     |    21.566 |     12.77 |
> | DGRMIL   | 10000    |   194.718 |     31.59 |
> | RRTMIL   | 100      |     4.243 |      3.13 |
> | RRTMIL   | 1000     |    15.006 |      5.92 |
> | RRTMIL   | 10000    |   142.116 |     46.47 |
>
> We will add these tables to the appendix and add an explicit link to them in the main text.
>
> ---
>
> ### W3 and Q3. Effect of rank and the choice $r = 64$
>
> We thank the reviewer for pointing this out and for giving us the opportunity to clarify our choice of rank.
>
> Our analysis and experiments show that the performance of MR-ABMIL (Fig. 3) saturates around $r = 32$ and then varies only mildly for larger $r$, while the performance of MR-CATE (Fig. 6 in the Appendix, cited in Sec. 4.3) still slowly grows after $r = 32$. The statement in the main text that MR is stable once $r$ exceeds the effective rank is therefore based on these empirical results.
>
> Given these observations, we chose to use $r = 64$ in the main comparison tables for the following reasons:
>
> - **Heavier models may benefit from a slightly larger rank**. For more expressive backbones such as CATE, the sensitivity curves show a small but monotonic improvement as $r$ increases beyond 32, so choosing $r = 64$ is a conservative global setting that avoids under tuning these models while remaining parameter efficient.
> - **Fairness and simplicity**. Using one shared $r$ across datasets and backbones avoids per configuration tuning and makes the comparison cleaner and easier to reproduce.
> - **Parameter efficiency still holds**. $r < \frac{d_0 d_1}{d_0 + d_1}$ still holds for all replaced linear layers, which guarantees that MR always uses fewer trainable parameters than the corresponding full linear layer.
>
> In practice, when applying MR to a specific model, practitioners are free to tune $r$ and to choose which linear layers to replace. We will make this explicit in Sec. 4.3.
>
> ---
>
> ### W4 and Q4. How MR is applied to TransMIL and CATE
>
> We thank the reviewer for this question. We clarify that we only replace a small set of internal linear layers in each backbone and Sec. G.4 of the appendix already lists the exact replacement positions for each backbone:
>
> - TransMIL: only replace the out matrix in Nystrom attention implementation with our MR block.
>
> - CATE: replace all linear layers in x\_linear, interv\_linear, feature output linear layer and the second linear layer in encoder\_IB with our MR block.
>
> - DGRMIL: replace linear layers in self.fc1 of optimizer\_triple and self.crossffn of DGRMIL.
>
> - RRTMIL: replace linear layer in self.proj of InnerAttention.
>
> In the revision, we will add a short paragraph in the main text that points explicitly to Sec. G.4.

---

> ### Author Response · Authors · 2025-11-24
>
> ### Q1. What if $B$ is a non-random fixed matrix such as the identity
>
> Thank you for this question. Our implementation of the MR block has three variants that we report in Table 2:
>
> - Full MR:
> $f_{\text{MR}}(X) = \operatorname{GELU}(X W_2) W_1 + XB$, where $B$ is a fixed random matrix drawn once at initialization and then frozen.
> - $-B$: we remove the random anchor term and keep the residual connection, so $f_{-B}(X) = \operatorname{GELU}(X W_2) W_1 + X$.
> - $-BX$: we remove both the random anchor and the residual, so only the low rank path remains:
> $f_{-BX}(X) = \operatorname{GELU}(X W_2) W_1$.
>
> The scenario suggested by the reviewer, $B = I$, is effectively realized by the $-B$ variant: if we replace the random anchor $B$ with the identity, the anchor term $X B$ becomes $X I = X$, and the block reduces to the $-B$ row that we already reported in Table 2.
>
> We will make this correspondence explicit in the revised manuscript so that readers can directly interpret $-B$ as the $B = I$ variant.
>
> ---
>
> ### Q2. Does the same geometric distortion appear for slide-level PFMs such as TITAN
>
> We appreciate this suggestion and agree that extending the geometric analysis to slide-level PFMs such as TITAN is an important next step. Our current analysis is implemented for patch-level features (CONCH, UNI, ResNet50), and we have not yet adapted this pipeline to slide-level embeddings. TITAN is built on CONCH v1.5 and outputs slide-level embeddings that are followed by a very small classification head (dim, numClass). In our design we never replace this final head with MR, since it mainly maps features to logits and an MR replacement would mostly increase parameters. Applying MR in combination with TITAN would therefore require a non-trivial redesign of where MR is inserted in the slide-level architecture.
>
> Moreover, our current manifold analysis relies on about 30,000 sampled patch features per configuration, whereas TITAN provides much less slide-level embeddings per dataset, which would make the same statistics substantially noisier and less stable. A careful adaptation of the analysis to this low sample regime is needed. We have not yet carried out this adaptation, and in the revision we will explicitly list this as a limitation and a natural direction for future work.

---

> > ### Comment · Reviewer_k14d · 2025-11-25
> >
> > Thank you very much for your high quality revision. It is addressed all my concerns. I will retain my scores.

---

### Official Review · Reviewer_tHAq · 2025-11-03

**Soundness:** 3
**Presentation:** 2
**Contribution:** 2
**Rating:** 2
**Confidence:** 4

**Summary:**

This paper tackles few-shot whole-slide image classification by arguing that the root cause of overfitting lies in a geometric mismatch between pretrained pathology features and downstream linear classifiers. The authors propose a Manifold Residual block, which introduces a random geometric anchor and a trainable low-rank residual path to preserve manifold structure while reducing model capacity. Experiments across several MIL backbones show accuracy improvements and parameter reductions. The paper positions itself as introducing a geometry-aware inductive bias for few-shot computational pathology.

**Strengths:**

1. The paper identifies a real and practically significant issue in computational pathology. The connection between feature geometry and data efficiency is conceptually interesting and relevant to current efforts in adapting large pretrained models for medical imaging.

2. The proposed MR block is lightweight, easy to implement, and compatible with a wide range of MIL backbones. It can be viewed as a structured parameter-efficient adapter.

3. The paper reports consistent accuracy gains across multiple models with substantial parameter reductions.

**Weaknesses:**

Major:

1. The paper attributes few-shot overfitting to the “destruction” of pretrained feature manifolds by downstream linear layers. This interpretation is not entirely convincing. Linear mappings are expected to reshape representations to achieve class separability, which is the very purpose of a classifier. The observed overfitting could instead result from limited data or excessive model capacity rather than geometric distortion. The causal link between ‘destruction’ and overfitting is not yet well established and could be further clarified with additional controlled experiments.

2. The proposed MR block introduces a fixed random matrix \(B\) as a geometric anchor. The t-SNE panel shows a non-trivial disagreement (~14%) in neighborhood structure. From an intuitive perspective, once the input features are multiplied by \(B\), much of the pretrained manifold structure and discriminative geometry are likely disrupted. Classifying on \(XB\) rather than on the original \(X\) would likely reduce performance. Even with sufficient data, a full-rank MIL training setup might not learn to counteract this direct perturbation of pretrained features, let alone a low-rank adaptation like LoRA. In contrast, linear layers transform pretrained features into a task-relevant space in a data-driven manner. Injecting random noise in this way fits more closely with the definition of “destruction” than “preservation.”

3. It is not entirely clear why extreme few-shot WSI classification is a key constraint here. The computational bottleneck typically lies in patch-level feature extraction and pretraining, not in the downstream classifier. Moreover, pretrained slide-level feature extractors such as TITAN already exist, which weakens the motivation for emphasizing few-shot adaptation at the classifier level.

Minor:

1. The finding that pretrained pathology features exhibit low-dimensional manifold structure is broadly consistent with prior work in vision and contrastive representation learning. Classification layers are expected to transform pretrained features into spaces that better align with downstream tasks, which naturally alters the geometry.

2. The reported performance gains may primarily arise from substantial parameter reductions. The current experiments do not separate this effect from the claimed geometric contribution, making it difficult to assess which factor drives the improvement.

**Questions:**

1. Could the authors disentangle the improvement due to parameter reduction from the claimed geometric preservation? A control experiment using an equally sized models would clarify this.

2. Does the MR block help when training data is not extremely limited? This would clarify whether the method primarily acts as a regularizer rather than a geometry-preserving transformation.

3. Please refer to the other weaknesses for additional concerns.

---

> ### Author Response · Authors · 2025-11-24
>
> We thank the reviewer for highlighting the importance of the problem, recognizing the universal plug in nature of our block, its consistent performance gains, and for the careful reading and constructive suggestions. We address the concerns below.
>
> ---
>
> ### 1. On the geometric interpretation and the cause of overfitting
>
> We thank the reviewer for the opportunity to clarify our interpretation. We do not claim that manifold distortion is the only cause of overfitting, but that it is one important and measurable contributing factor supported by our analyses.
>
> Figs. 1, 4, 5(a,d) show that pathology foundation model features lie on a low-dimensional, nonlinear manifold. We then observe that standard linear layers inside MIL backbones significantly alter tangent drift relative to this manifold, whereas the random projection and MR keep the drift curve much closer to that of the original features. We therefore use “manifold distortion” to denote this concrete, quantified effect, and view it as one mechanism that contributes to overfitting in these intermediate layers, not as an exclusive explanation.
>
> Importantly, the layers we replace are intermediate projections and attention scoring modules inside the MIL backbone, whose primary role is to transform and weight instance features, not to serve as the final classifier that enforces linear separability. The last linear layer that maps the aggregated slide representation to logits remains a standard full rank classifier, so the model retains full capacity to shape task-specific decision boundaries. MR is used only to control how pretrained features are transformed before they reach this classifier.
>
> In addition, the MR block does not remove the ability to learn task-specific representations. As shown in Eq. (2) and Sec. 3.2.3, MR can approximate any linear map arbitrarily well, so in the worst case it has the ability to reproduce a standard linear layer. Its effect is to introduce a geometry-aware inductive bias.
>
> In the revision we will make this intent explicit and clarify that our empirical evidence supports the view that reducing tangent drift in these intermediate layers improves generalization in the extreme few-shot regime.
>
> ---
>
> ### 2. On the random geometric anchor and the notion of "destruction"
>
> We appreciate the concerns regarding the structure of our block design. We address this in three parts: (i) the random projection is a controlled, near isometric transform that also shapes the spectrum, (ii) the model does not classify on $X B$ alone, and (iii) the low-rank residual is matched to the low effective rank of pathology foundation model features.
>
> First, the random anchor is a gentle reshaping rather than a destructive perturbation. Figs. 1, 4, and 5 show that for features from different pathology foundation models the random projection preserves cluster topology well: the ARI is about 0.86, so most neighborhood relations are retained. At the same time, the singular value spectrum becomes slightly more concentrated and the effective rank decreases, meaning that energy is centralized into fewer directions. This is the mild spectral shaping we want from the anchor: it regularizes the representation while keeping local geometry close to that of the original manifold, but not exactly identical. Consistently, the tangent drift curve of the random anchor closely follows that of the original features, whereas a trained linear layer deviates much more strongly (Figs. 1, 4, 5(d)). This is consistent with classical random projection theory (Sec. 3.2.3 and Appendix D), which states that for appropriate output dimension a random projection is a near isometry with high probability.
>
> Second, the model never classifies on $X B$ alone. The MR block always includes a low-rank residual path that learns task-specific adaptation on top of the anchored representation. Our ablations in Table 2 show that using only the random projection or only the low-rank residual performs worse than the full MR block, indicating that both components are necessary: the anchor provides a stable, gently reshaped geometry and the residual path adds task-specific flexibility.
>
> Third, the low-rank residual is not an arbitrary restriction but is aligned with the measured low effective rank of the CONCH features (Sec. 3.1.2). The signal of interest already lives in a low-dimensional manifold, and with this premise, a LoRA style low-rank adaptation with activation is expressive enough to capture task-specific directions while avoiding overfitting in directions where the data carry little energy.
>
> In the revision we will make this interpretation more explicit and clarify the distinct roles of the anchor and residual paths.

---

> ### Author Response · Authors · 2025-11-24
>
> ### 3. Why focus on extreme few-shot WSI classification at the classifier level
>
> We thank the reviewer for asking about the motivation of this work. We agree that the main computational cost lies in patch-level pretraining and feature extraction, and that strong slide-level models such as TITAN already exist. However, our focus is on a different constraint: label scarcity that arises from clinical limitations rather than computational burden. In many clinically important endpoints (for example treatment response, rare subtypes, and biomarker surrogates), the number of available WSIs is inherently few-shot, so the few-shot setting has real-world impact.
>
> In this scenario, the MIL layers become the main place of overfitting, not the patch encoder, since the latter is pretrained on abundant data and kept frozen. Our contribution is an architectural change at this stage that respects the observed low-dimensional manifold structure of pretrained features and improves data efficiency under extreme few-shot supervision. This setting is exactly the FSWSL paradigm described in Sec. 2.2.
>
> Slide-level foundation models like TITAN are indeed highly valuable, and complementary to our approach rather than a replacement. However, each slide-level foundation model is tied to a specific patch encoder. Our goal is to provide a plug-in block that can be used with any existing or future pathology foundation model (for example UNI2) even when a matching slide-level FM is not yet available.
>
> In the revision, we will clarify this complementary role more clearly in the introduction and discussion, and explicitly state that applying MR on top of slide-level PFMs and inherently few-shot clinical endpoints is a natural extension rather than a competing direction.
>
> ---
>
> ### 4. Parameter reduction versus geometric effects
>
> We thank the reviewer for pointing this out and for helping us strengthen the analysis. We agree that it is important to separate performance gains due to parameter reduction from those due to the structural inductive bias of MR. To isolate capacity effects, we construct a capacity matched MR variant on ABMIL: the gated attention layer has input and output dimensions 512 and 256, so we set $r=\frac{512\cdot 256}{512+256}=170$ such that the MR block has exactly the same number of trainable parameters as the vanilla linear layer.
>
> The results below show that this capacity matched MR achieves clear gains over the baseline on Camelyon16 and RCC across all shot settings, and maintains comparable performance on NSCLC (marginally lower at 4 shot, similar or better for 8 and 16 shots). This indicates that parameter reduction alone cannot explain the improvements, and that the geometry-aware design of MR plays an important role beyond capacity control.
>
>
> | Model | Shots |  | Camelyon16 |  |  | NSCLC |  |  | RCC |  |
> |----|-----|----|----|----|----|----|----|----|----|----|
> |  |  | AUC | F1 | ACC | AUC | F1 | ACC | AUC | F1 | ACC |
> | ABMIL | 4 | 45.5±7.6 | 49.2±7.2 | 56.1±5.2 | 86.5±5.1 | 80.1±5.0 | 80.2±4.9 | 89.3±6.4 | 74.2±11.9 | 76.7±11.3 |
> | MR-ABMIL | | **56.6±16.3** | **54.0±14.7** | **58.4±13.2** | 78.7±4.0 | 70.8±3.7 | 71.1±3.5 | **97.7±1.3** | **86.9±2.5** | **89.3±2.6** |
> | ABMIL | 8 | 49.2±5.6 | 48.9±3.0 | 59.7±4.8 | 87.7±5.9 | 81.0±5.4 | 81.1±5.4 | 91.8±6.3 | 76.6±8.4 | 79.1±7.9 |
> | MR-ABMIL | | **73.2±15.0** | **70.7±15.9** | **72.6±15.8** | **88.9±8.5** | 80.0±8.1 | 80.1±8.2 | **98.6±0.6** | **88.7±3.2** | **90.9±2.9** |
> | ABMIL | 16 | 73.2±17.1 | 71.2±15.4 | 75.7±10.2 | 93.6±2.4 | 86.9±2.8 | 87.0±2.8 | 94.7±5.2 | 81.0±7.8 | 82.8±8.5 |
> | MR-ABMIL | | **84.3±11.7** | **79.3±11.2** | **80.3±11.1** | **93.6±1.9** | **87.8±3.4** | **87.8±3.4** | **98.8±0.3** | **89.7±2.0** | **91.7±1.6** |

---

> ### Author Response · Authors · 2025-11-24
>
> ### 5. Behavior when data is not extremely limited
>
> We agree with the reviewer that it is important to test whether MR continues to help when data is not extremely limited. Therefore, we conducted additional experiments on TCGA-NSCLC and TCGA-RCC with larger numbers of shots, $k = 32, 64$. The results in the table below show that MR-ABMIL and MR-CATE consistently outperform the corresponding vanilla baselines in macro F1 and AUC at both $k = 32$ and $k = 64$.
>
> | Model | Shots | | NSCLC |  |  | RCC |  |
> |----|-----|----|----|----|----|----|----|
> |  |  | AUC | F1 | ACC | AUC | F1 | ACC |
> | ABMIL | 32 | 96.2±1.9 | 89.7±3.6 | 89.7±3.6 | 97.6±1.8 | 91.1±2.0 | 93.1±2.0 |
> | MR-ABMIL |  | **98.1±0.4** | **93.1±1.7** | **93.1±1.7** | **99.5±0.3** | **93.4±0.6** | **94.9±0.6** |
> | CATE | | 97.0±0.4 | 91.1±1.6 | 91.1±1.6 | 99.3±0.3 | 91.0±2.0 | 93.1±1.6 |
> | MR-CATE |  | **97.7±0.5** | **92.1±1.6** | **92.1±1.6** | **99.6±0.1** | **92.9±2.2** | **94.5±1.9** |
> | ABMIL | 64 | 97.5±0.9 | 91.5±1.9 | 91.5±1.9 | 99.0±0.6 | 91.6±2.9 | 93.2±2.5 |
> | MR-ABMIL | | **98.2±0.3** | **92.8±1.2** | **92.8±1.2** | **99.6±0.2** | **94.3±1.1** | **95.6±1.4** |
> | CATE | | 97.7±0.5 | 91.1±1.4 | 91.1±1.4 | 99.4±0.2 | 93.6±1.1 | 95.0±0.8 |
> | MR-CATE |  | **97.9±0.1** | **92.4±1.1** | **92.4±1.1** | **99.8±0.1** | **95.3±1.0** | **96.6±0.8** |
>
> These results are consistent with our interpretation of MR as a geometry-aware regularizer. The relative gain is largest in the extreme few-shot regime, gradually decreases as supervision increases, and remains positive without degrading performance when more data is available.

---

### Official Review · Reviewer_jYL1 · 2025-11-05

**Soundness:** 3
**Presentation:** 2
**Contribution:** 3
**Rating:** 4
**Confidence:** 4

**Summary:**

The authors introduce a novel layer to preserve low-dimensional manifold geometry within modern Multiple Instance Learning (MIL) frameworks for few-shot classification of whole slide pathology images. They first show that embedding spaces from well-known pathology foundation models manifest low-dimensional manifold geometries and that these are not well-preserved in popular attention-based MIL framework. Authors show that this tends to be due to linear layers such as those present within gated-attention mechanism used in ABMIL and they propose a novel layer, called the Manifold Residual (MR) block, to better preserve geometry. The latter is decomposed in 2 parts operating on a feature matrix $X$: (1) a fixed random matrix transforming linearly $X$ useful to preserve topology; (2) a trainable low-rank residual pathway (LRP). Authors study the theoretical properties of their method, demonstrate its relevance to improve many MIL models for few-shot classification tasks and perform a range of ablation studies.

**Strengths:**

- The authors propose a relevant analysis to emphasise low-dimensional manifold properties of a range of foundation models for pathology.
- Propose a novel layer for few-shot MIL, the MR block with a custom training strategy.
- They provide theoretical results on a range of geometric/statistical properties preserved by perturbations by random matrices.
- Demonstrate a universality approximation theorem for the MR block.
- Show on 3 datasets that the MR blocks, instead of linear layers, within 5 MIL frameworks improve few-shot WSI classification while leveraging 3 different types of pretrained models (CONCH, UNI, ResNet50)
- Perform ablations on the 2 parts included within the MR blocks, which tend to show that coupling these 2 parts brings the best performance.
- Conduct several sensitivity analyses, to question where to replace linear layers with MT blocks within ABMIL, how to initialize the MR blocks and whether the MR blocks are robust to their rank hyperparameter.

**Weaknesses:**

- **W1 : clarity** There are several points in the paper that would benefit from clarification and/or further detail:
   - a) L63: "linear layer". For people knowing the MIL literature it is not clear at this stage, about which linear layers you are referring to, e.g those included in the gated-attention layer of ABMIL or actually the linear classifier at the end of the architecture, which can also have an effect. This should be clarified.
   - b) The dataset used for the geometric studies reported in Fig 1 and 5 is never mentioned.
   - c) Figure 2: The box "MIL model"  explaining how are supposed to intervene the MR blocks is really not clear.
   - d) Section 2.2: it is not clear to me why more generic few-shot learning paradigms/literature (see e.g [A]), applicable to any bag representations in MIL is omitted in the related work. For instance well-known prototypical neural networks [B] could be applied as a readout within ABMIL instead of a linear layer and their inherent dependence to distances could be a good proxy to preserve geometric properties. This observation underlines that it is not clear in the paper why different readout strategies are not discussed. I invite authors to do so during the rebuttal.
   - e) Section 3.1: While both instance-level MIL and embedding/bag-level MIL are mentioned in Section 2.1, Section 3.1 only formalizes bag-based approaches. It can be sufficient to mention that in Section 3.1 with a disclaimer that only bag-based methods are benchmarked in the paper.
   - f) To improve readibility of most tables, I suggest authors to express everything in % instead of 0.x.


[A] Song, Y., Wang, T., Cai, P., Mondal, S. K., & Sahoo, J. P. (2023). A comprehensive survey of few-shot learning: Evolution, applications, challenges, and opportunities. ACM Computing Surveys, 55(13s), 1-40.

[B] Snell, J., Swersky, K., & Zemel, R. (2017). Prototypical networks for few-shot learning. Advances in neural information processing systems, 30.

- **W2: benchmarks and ablations**:
   - a) Most tested datasets are significantly imbalanced hence I don't think that the choice of metrics such as the accuracy and AUC are the most appropriate. I believe that simply presenting macro F1 scores in the main paper could be sufficient to share the main messages, and potentially include AUPRC [C] for completeness in the main paper or supplementary.
   - b) In most ablation studies and sensivity analyses, many strong claims are made by authors when the results hold for at most 2 out of 3 datasets. Therefore I strongly encourage authors to include at least 2 other WSI datasets in their experiments to better support these claims.
   - c) Authors argue that a central issue of MIL methods for few-shot classification is overfitting. Nonetheless, there is a pletora of implicit or explicit regularizations (e.g dropout, attention dropout, norm constraints etc) that could be envisioned. Hence the scope of the baselines chosen by authors is not clear and should be further justified by authors or completed by including different regularization techniques.
   - d) Hyperparameters of benchmarked MIL models are not present in the paper and should be added.

- **W3: asymptotic analysis**: While authors mention that their method brings less improvements in 16-shots WSI classification than with less supervision, it could be interesting to stress test their methods with higher ranges of shots on the larger datasets like TCGA-NSCLC.

[C] McDermott, M., Zhang, H., Hansen, L., Angelotti, G., & Gallifant, J. (2024). A closer look at auroc and auprc under class imbalance. Advances in Neural Information Processing Systems, 37, 44102-44163.

**Questions:**

I invite authors to discuss the weaknesses mentioned above, knowing that I am really inclined to increase my initial grade. A last question:

Q1. Could authors clarify whether there are correlations between geometric properties of the different datasets with the results observed in the ablation studies reported in Table 2, Table 3 and Figure 3?

---

> ### Author Response · Authors · 2025-11-24
>
> We thank the reviewer for the positive assessment of the proposed novel layer, for recognizing the strength of the evaluation, and for the helpful comments on clarity, benchmarks, and analysis. Below we address each point in W1–W3 and Q1.
>
> ---
>
> ### W1 - Clarity
>
> **(a) Which linear layers we refer to**
>
> We thank the reviewer for this question. We clarify that we only replace a subset of internal linear layers in each backbone and Sec. G.4 of the appendix already lists the exact replacement positions for each backbone in the experiments:
>
> - ABMIL: replace both the $V$ and $U$ matrix with our MR block.
>
> - TransMIL: only replace the out matrix in Nystrom attention implementation with our MR block.
>
> - CATE: replace all linear layers in x\_linear, interv\_linear, feature output linear layer and the second linear layer in encoder\_IB with our MR block.
>
> - DGRMIL: replace linear layers in self.fc1 of optimizer\_triple and self.crossffn of DGRMIL.
>
> - RRTMIL: replace linear layer in self.proj of InnerAttention.
>
> In the revision, we will add a short paragraph in the main text that points explicitly to Sec. G.4.
>
> **(b) Dataset used for Fig. 1 and Fig. 5**
>
> We thank the reviewer for pointing this out. The dataset used to generate Figs. 1, 4, 5 is Camelyon16, which is stated in Section B in the appendix, where we describe the setup of the geometric analyses.
>
> To avoid any ambiguity, we will explicitly mention Camelyon16 in the main paper and add a pointer to the appendix.
>
> **(c) Clarifying Fig. 2**
>
> We agree that the "MIL model" box in Fig. 2 can be made clearer. Fig. 2 is intended as a backbone-agnostic schematic: MR can replace any linear layer in a MIL model. In the revision we will state this explicitly in the caption and highlight example insertion points to make the role of MR visually clearer.
>
> **(d) Missing generic few-shot literature and readout strategies**
>
> We thank the reviewer for highlighting the broader few-shot learning literature [A] and metric-based methods such as Prototypical Networks [B]. These methods mainly modify the readout: given an embedding, they replace the final linear classifier with a distance based decision rule (for example a softmax over negative distances to class prototypes).
>
> In this paper we deliberately keep the standard linear readout used in prior WSI MIL work, so that we can isolate the effect of geometry-aware intermediate layers. In addition, we would like to clarify that we do not apply MR to the final classifier head, which is already very small and only maps the slide embedding to logits, so an MR replacement there would mostly add parameters without clear geometric benefit. Instead, we insert MR into internal linear layers of the MIL backbone (for example the $U$ and $V$ matrices in ABMIL), where our tangent space analysis shows substantial distortion of the low-dimensional manifold of pretrained pathology features. Prototypical Networks rely on Euclidean distances, which provide a reasonable local metric but do not explicitly control the multi-hop tangent drift that we measure. MR directly regularizes these internal transformations and is in principle compatible with either linear or prototype based readouts.
>
> Therefore, we view these methods as a complementary stream to our method. In the revision we will cite [A] and [B], clarify this complementary scope, and mention combining MR with prototype based heads as an interesting direction for future work.
>
> **(e) Bag level vs instance level MIL**
>
> We thank the reviewer for bringing this up to make our paper more rigorous. We will add a short disclaimer at the start of Section 3.1 stating that our theoretical analysis and benchmarks are restricted to bag level MIL.
>
> **(f) Reporting metrics in percent**
>
> We agree with the suggestion from the reviewer that percent formatting improves readability. In the revision, we will present the results in percentage, not decimal.

---

> ### Author Response · Authors · 2025-11-24
>
> ### W2 - Benchmarks and ablations
>
> **(a) Metrics for imbalanced datasets**
>
> We thank the reviewer for suggesting more comprehensive metrics.
>
> We intended to follow the evaluation protocol of prior WSI works such as TransMIL, RRTMIL, and ViLaMIL. We therefore adopt the same widely-used datasets and metrics, so that our results are directly comparable to existing methods. As stated in Sec. 4.1, we report three complementary metrics: AUC, F1, and accuracy. The F1 reported in all tables is macro F1, so class imbalance is already taken into account in the main results.
>
> We agree that AUPRC is also informative under strong imbalance. During the rebuttal period we have computed AUPRC for a representative subset of settings, and we will include these values in an additional table in the appendix. We will also clarify in the text that macro F1 is our primary metric for handling imbalance, and that AUPRC is provided for completeness.
>
> | Model | Shots |  |  | Camelyon16 |  |  |  | NSCLC |  |  |  | RCC |  |
> |----|-----|----|----|----|----|----|----|----|----|----|----|----|----|
> |  |  | AUC | F1 | ACC | AUPRC | AUC | F1 | ACC | AUPRC | AUC | F1 | ACC | AUPRC |
> | ABMIL | 2 | 41.2±7.6 | 43.1±6.5 | 46.5±6.2 | 35.8±6.6 | 77.4±14.1 | 68.4±9.8 | 69.1±9.4 | 74.0±12.0 | 84.4±7.3 | 63.2±8.1 | 64.7±9.1 | 74.7±11.5 |
> | MR-ABMIL |  | **45.0±11.6** | **43.2±8.6** | **57.1±6.2** | **37.9±12.1** | **85.5±6.5** | **75.9±7.2** | **76.3±7.0** | **81.2±9.3** | **92.6±6.3** | **78.1±7.5** | **81.8±7.2** | **86.4±11.3** |
> | CATE | | 49.1±10.3 | 45.7±6.1 | 52.9±7.8 | 40.7±9.9 | 78.9±3.8 | 70.8±3.2 | 71.0±3.0 | 74.6±6.5 | 94.7±2.1 | 81.7±3.2 | 84.0±2.5 | 89.6±5.0 |
> | MR-CATE |  | **54.9±18.8** | **51.9±16.1** | **54.3±16.6** | **48.2±21.2** | **83.2±7.9** | **75.1±7.3** | **75.3±7.2** | **80.1±10.1** | **96.0±3.2** | **82.4±4.1** | **85.3±4.0** | **91.0±6.0** |
> | ABMIL | 4 | 45.5±7.6 | 49.2±7.2 | 56.1±5.2 | 40.7±6.9 | 86.5±5.1 | 80.1±5.0 | 80.2±4.9 | 84.7±5.8 | 89.3±6.4 | 74.2±11.9 | 76.7±11.3 | 82.7±9.4 |
> | MR-ABMIL |  | **46.1±13.7** | 43.8±10.0 | **56.6±5.2** | 39.0±13.5 | **86.8±2.5** | 79.3±2.3 | 79.4±2.1 | 81.9±6.1 | **96.5±3.6** | **85.5±6.0** | **87.8±5.5** | **93.6±5.1** |
> | CATE | | 59.7±12.4 | 50.0±9.8 | 58.3±6.7 | 52.6±12.1 | 81.4±3.4 | 72.3±4.1 | 72.8±3.6 | 76.9±4.9 | 96.9±1.8 | 83.9±2.9 | 86.5±2.5 | 94.3±2.5 |
> | MR-CATE |  | **62.1±16.9** | **58.4±15.8** | **62.3±13.5** | **54.3±19.6** | **87.0±3.9** | **79.4±3.2** | **79.4±3.2** | **83.9±7.9** | **98.2±1.2** | **88.7±2.0** | **90.5±1.8** | **96.1±2.4** |
> | ABMIL | 8 | 49.2±5.6 | 48.9±3.0 | 59.7±4.8 | 45.4±4.1 | 87.7±5.9 | 81.0±5.4 | 81.1±5.4 | 81.7±7.2 | 91.8±6.3 | 76.6±8.4 | 79.1±7.9 | 87.7±8.2 |
> | MR-ABMIL |  | **56.5±15.5** | **53.8±13.9** | **61.9±8.7** | **53.0±19.1** | **94.0±3.3** | **85.9±3.3** | **86.0±3.3** | **91.7±5.2** | **98.1±0.8** | **87.6±2.9** | **89.6±2.1** | **96.4±1.2** |
> | CATE | | 79.8±14.7 | 69.1±20.0 | 76.0±12.6 | 76.8±19.0 | 87.7±5.7 | 78.0±4.9 | 78.2±4.9 | 84.9±6.6 | 97.8±0.8 | 86.4±3.0 | 88.3±2.4 | 96.0±1.1 |
> | MR-CATE |  | **88.0±14.8** | **81.7±16.0** | **82.9±15.3** | **84.9±20.9** | **93.5±5.3** | **84.6±5.7** | **84.7±5.6** | **92.6±6.0** | **98.6±0.3** | **89.1±1.3** | **90.7±1.0** | **97.2±0.5** |
> | ABMIL | 16 | 73.2±17.1 | 71.2±15.4 | 75.7±10.2 | 72.5±18.5 | 93.6±2.4 | 87.8±3.4 | 87.8±3.4 | 91.0±4.4 | 94.7±5.2 | 81.0±7.8 | 82.8±8.5 | 90.8±8.2 |
> | MR-ABMIL |  | **79.0±23.1** | **74.5±22.4** | **78.0±17.1** | **77.9±26.5** | **96.5±1.2** | **89.1±1.0** | **89.1±1.0** | **96.3±1.5** | **98.3±0.7** | **89.1±2.8** | **91.1±2.0** | **96.4±1.8** |
> | CATE | | 92.2±3.1 | 87.5±3.2 | 88.5±2.8 | 92.5±2.7 | 94.2±2.1 | 85.1±2.0 | 85.2±2.0 | 92.8±3.7 | 97.9±0.6 | 87.4±2.7 | 89.3±2.0 | 95.9±1.3 |
> | MR-CATE |  | **92.4±3.2** | 86.7±3.9 | 87.8±3.4 | 92.2±3.3 | **96.1±1.4** | **88.6±3.3** | **88.7±3.3** | **96.0±1.8** | **98.6±0.6** | **89.8±3.1** | **91.6±2.3** | **97.2±1.1** |

---

> ### Author Response · Authors · 2025-11-24
>
> **(b) Limited number of datasets**
>
> We thank the reviewer for highlighting the generalization issue that ablation and sensitivity studies should be interpreted with care when trends hold on only a subset of datasets. In our work, the main mechanism and rank sensitivity analyses are already run on multiple WSI cohorts. Component ablations and rank sweeps are conducted on two TCGA datasets, NSCLC and RCC, and some variants are also evaluated on Camelyon16 (three datasets in total for the ablation studies). These datasets differ in organ, class structure, source, and difficulty. On these larger cohorts the few-shot averages are relatively stable, and the key qualitative patterns, such as MR improving over the baseline in the few-shot regime and becoming relatively insensitive to the rank once it exceeds the effective rank, are consistent across datasets. In the revision we will slightly refine our wording to make explicit that these conclusions are grounded in this set of datasets and backbones, rather than claimed as universal for all possible WSI settings.
>
> During the rebuttal period we also ran additional rank sensitivity experiments on two treatment response cohorts (Boehmk and Trastuzumab). These tasks are inherently few-shot and clinically important, but the total number of slides is much smaller than in Camelyon16, NSCLC, and RCC, which limits statistical power. On both Boehmk and Trastuzumab, MR consistently improves over ABMIL in AUC and accuracy across shots, confirming that the block remains helpful on realistic treatment response tasks. However, due to the very small sample sizes, rank-sweep curves on these cohorts are noisier, so we do not draw very fine grained geometric conclusions (such as a single optimal rank) from them. We use them as external validation of the method, while our main geometric interpretation is grounded in the larger TCGA cohorts.
>
> | Model | Shots |  | Boehmk |  |  | Trastuzumab |  |
> |----|-----|----|----|----|----|----|----|
> |  |  | AUC | F1 | ACC | AUC | F1 | ACC |
> | ABMIL | 4 | 56.2±6.7 | 45.9±8.6 | 49.7±10.8 | 47.7±5.6 | 40.3±3.9 | 42.4±2.6 |
> | MR-ABMIL |  | **65.4±11.3** | **56.9±8.5** | **65.1±3.7** | **56.9±2.3** | **40.9±8.7** | **58.8±0.0** |
> | ABMIL | 8 | 53.4±5.1 | 46.1±4.6 | 50.9±8.2 | 49.4±3.7 | 48.8±10.8 | 54.1±7.7 |
> | MR-ABMIL |  | **63.4±9.5** | **57.6±7.1** | **62.3±9.8** | **55.1±4.4** | 40.3±4.5 | **54.1±6.4** |
> | ABMIL | 16 | 52.6±5.5 | 46.9±7.3 | 50.3±8.0 | 46.3±10.0 | 36.1±1.3 | 48.2±6.4 |
> | MR-ABMIL |  | **63.5±4.1** | **55.1±3.4** | **63.4±6.5** | **57.7±0.8** | **43.6±3.3** | **56.5±3.2** |
>
> **(c) Scope of regularization baselines**
>
> We thank the reviewer for raising this point about regularization. We would like to clarify that standard regularization is active in all our baselines:
>
> - For ABMIL, dropout is enabled.
> - For TransMIL, RRTMIL, DGRMIL, and CATE, we keep the normalization layers, dropout, and other regularization components as specified in their original papers and codebases.
> - MR is applied on top of these already regularized backbones, and the gains reported in Table 1 and the ablation tables are clear improvements beyond what is achieved with dropout and normalization.
>
> Therefore, the baseline results in Table 1 and Table 6 already include standard regularization, and our method further improves performance because it introduces a geometry-aware inductive bias that is complementary to these geometry-agnostic techniques.
>
> We will make this explicit in Sec. 4.1 and Sec. G.2 by listing the regularization settings for each backbone.
>
> **(d) Hyperparameters of benchmarked MIL models**
>
> We thank the reviewer for this question. The hyperparameters and the experimental setups of the benchmarked MIL models are already included in Section G.2 of the appendix. This section provides:
>
> - learning rates, optimizers, batch sizes,
> - number of epochs and early stopping criteria,
> - dropout and weight decay values,
> - the rank $r$ and initialization details for MR blocks,
> - and any backbone specific settings.
>
> To make this easier to find, we will add an explicit pointer in Section 4.1 that directs readers to Section G.2 for full hyperparameter details.

---

> ### Author Response · Authors · 2025-11-24
>
> ### W3 - Asymptotic analysis
>
> We thank the reviewer for asking this question. In response to this question, we have conducted additional experiments on the two TCGA datasets for larger numbers of shots $k = 32, 64$. The results show that MR continues to outperform the corresponding vanilla linear layer baselines at $k = 32$ and $k = 64$ when using ABMIL and CATE.
>
> | Model | Shots | | NSCLC |  |  | RCC |  |
> |----|-----|----|----|----|----|----|----|
> |  |  | AUC | F1 | ACC | AUC | F1 | ACC |
> | ABMIL | 32 | 96.2±1.9 | 89.7±3.6 | 89.7±3.6 | 97.6±1.8 | 91.1±2.0 | 93.1±2.0 |
> | MR-ABMIL |  | **98.1±0.4** | **93.1±1.7** | **93.1±1.7** | **99.5±0.3** | **93.4±0.6** | **94.9±0.6** |
> | CATE | | 97.0±0.4 | 91.1±1.6 | 91.1±1.6 | 99.3±0.3 | 91.0±2.0 | 93.1±1.6 |
> | MR-CATE |  | **97.7±0.5** | **92.1±1.6** | **92.1±1.6** | **99.6±0.1** | **92.9±2.2** | **94.5±1.9** |
> | ABMIL | 64 | 97.5±0.9 | 91.5±1.9 | 91.5±1.9 | 99.0±0.6 | 91.6±2.9 | 93.2±2.5 |
> | MR-ABMIL | | **98.2±0.3** | **92.8±1.2** | **92.8±1.2** | **99.6±0.2** | **94.3±1.1** | **95.6±1.4** |
> | CATE | | 97.7±0.5 | 91.1±1.4 | 91.1±1.4 | 99.4±0.2 | 93.6±1.1 | 95.0±0.8 |
> | MR-CATE |  | **97.9±0.1** | **92.4±1.1** | **92.4±1.1** | **99.8±0.1** | **95.3±1.0** | **96.6±0.8** |
>
> ---
>
> ### Q1 - Correlations between geometry and ablation results
>
> We thank the reviewer for this thoughtful question. In the current paper, the geometric analyses in Sec. 3.1 are primarily used to characterize the pretrained pathology feature manifolds and to motivate the design of the MR block, rather than as per dataset predictors of the exact performance gains in Tables 2 and 3 and Fig. 3.
>
> Across datasets, we observe consistent qualitative trends: pathology foundation model features lie on a low-dimensional curved manifold, standard linear sublayers in MIL backbones can induce substantial tangent drift away from this manifold, and replacing these layers with MR reduces the drift. In the few-shot regime, this reduction in drift is usually accompanied by improved generalization and a smaller train test gap for MR compared to the baseline. However, when we examined simple one dimensional geometric statistics that quantitatively predict the magnitude of the performance gain for each dataset and backbone, we did not observe a single robust dataset wide correlation. The final outcomes depend on several interacting factors, including architecture, which linear layers are replaced, label difficulty, and the number of shots, making them hard to predict from a single geometric statistic.
>
> For this reason, we do not treat the geometric metrics as direct quantitative predictors of performance. Instead, we view them as providing a systematic, measurable characterization of how standard linear layers and MR transform the pretrained feature manifold, and as evidence that MR better preserves this geometry in settings where labels are scarce. In the revision, we will clarify this scope while leaving a more exhaustive correlation study across datasets and architectures as future work.

---

> > ### Comment · Reviewer_jYL1 · 2025-11-27
> > **Answer to reviewers**
> >
> > Thank you for your detailed responses. Taking these into account, along with the responses to all reviewers, I find the rebuttal very convincing and will increase my score accordingly.

---

### Author Response · Authors · 2025-11-24

We sincerely thank all reviewers for the time and care devoted to evaluating our paper. Your thoughtful comments and suggestions are very valuable to us and have guided several improvements to the work. Our point-by-point responses are provided below, and we will update the PDF to reflect these changes.

---

### Author Response · Authors · 2025-12-04

Dear Program Committee,

Thank you very much for taking over our submission at this stage. We understand that there have been changes in AC assignments, and we sincerely appreciate your time and effort in handling this paper. To help reduce your workload, we briefly summarize our work, the review situation, and the main changes after rebuttal.

**Summary**: Our work analyzes a key geometric cause of overfitting in few-shot WSI classification. Using a suite of geometric tools, we show that pathology foundation model features lie on a low-dimensional manifold that is distorted by standard linear layers. To address this, we introduce the Manifold Residual (MR) block, a plug-in, geometry-aware replacement for linear layers that combines a fixed geometric anchor with a low-rank residual pathway. Extensive experiments across multiple MIL backbones and datasets demonstrate its effectiveness and parameter efficiency.

**Scores**. The initial scores are 2, 4, 6, and 10. As reflected in the public discussion, Reviewer jYL1 (initially 4) stated that the rebuttal addressed the main concerns and indicated an intention to raise the score, and Reviewer k14d (10) explicitly confirmed that all concerns had been resolved and retained the strong-accept rating. We have had no further interaction with the other two reviewers, whose scores remained unchanged.

**Strengths**: Important and timely problem; strong, clear, and insightful geometric diagnosis of few-shot WSI overfitting; well-motivated, lightweight novel MR block for MIL backbones; theoretical support via random projection properties and a universality result; comprehensive experiments with baselines, ablations, and sensitivity analyses; consistent few-shot gains with substantially fewer trainable parameters.

**Weaknesses**. Reviewers raised concerns about scope and clarity (which linear layers are replaced, which datasets are used for geometry, how MR is inserted into complex MIL backbones, and how the rank is chosen), about limitations (for example PCA as a replacement for the random anchor, and the lack of explicit experiments on slide-level PFMs such as TITAN), about experiments (requests for AUPRC under imbalance, inherently few-shot treatment response cohorts, higher-shot settings, capacity-matched controls, and comparison to recent few-shot WSI baselines such as FOCUS), and about training details and regularization (initially under-specified hyperparameters and regularization settings).

---

> ### Author Response · Authors · 2025-12-04
>
> **Summary of Our Responses**. Below we briefly summarize how we addressed the main weaknesses highlighted in the reviews (not exhaustive). For full details, please refer to the rebuttal and the revised PDF.
>
>
> - **Scope and Clarity**.
> 	- We clarified that MR replaces only a subset of internal linear layers in MIL backbones, while the final slide-level classifier remains a vanilla linear layer. Sec. G.4 now lists the exact linear layers that are replaced for each backbone.
> 	- We state explicitly that our analysis and experiments focus on bag-level MIL, and that the geometric metrics are used as qualitative diagnostics of how different layers transform the pretrained feature manifold, rather than as single-dimensional quantitative predictors of performance on each dataset.
> 	- We clarify the rationale for setting the rank $r$ to 64, rather than the theoretically analyzed value 32.
> 	- We added a comparison with general few-shot learning techniques in the related work section.
>
> - **Limitations of This Work**.
> 	- Using PCA as a replacement for our geometric anchor is an interesting idea. We discuss this in the rebuttal and leave it as a promising direction for future work.
> 	- We highlight that extending both MR and the geometric analysis pipeline to slide-level pathology foundation models such as TITAN is non-trivial and is an important direction for future work that is not claimed in the current submission.
>
> - **Additional Experiments**.
> 	- For imbalanced settings, we now report AUPRC in addition to AUC, macro F1, and accuracy.
> 	- We added experiments on two inherently few-shot treatment response cohorts. On both cohorts MR-ABMIL consistently improves over ABMIL.
> 	- We added higher-shot results on TCGA-NSCLC and TCGA-RCC at $k = 32$ and $k = 64$ ($k = 2,4,8,16$ in the original text), which show that MR continues to provide small positive gains when more supervision is available.
> 	- To disentangle parameter count from geometric effects, we constructed a capacity-matched MR-ABMIL variant whose number of trainable parameters exactly matches the vanilla linear layer. This control still yields clear gains over ABMIL on Camelyon16 and RCC and comparable or better performance on NSCLC, indicating that the improvements are not due only to reduced capacity.
> 	- We added FOCUS as a strong few-shot baseline in the main comparison table. Our MR-enhanced MIL backbones achieve comparable or better few-shot performance with much fewer parameters.
> 	- We reported FLOPs and wall-clock latency for the MR-enhanced backbones under different bag sizes (100, 1,000, 10,000) and feature dimensions (512 for CONCH, 1,024 for UNI and ResNet50).
>
> - **Training Details and Regularization**.
> 	- We added a pointer to Sec. G.2, which contains all hyperparameters and training protocols, including optimizers, learning rates, batch sizes, early stopping, dropout, weight decay, etc.
> 	- We make explicit that all baseline MIL backbones already use standard regularization (normalization, dropout, weight decay, and the regularization components from the original papers), and that MR is applied on top of these geometry-agnostic techniques as a geometry-aware inductive bias.
>
> Thank you very much again for your time and for stepping in to handle this submission.

---

### Meta-Review · Area_Chair_1mzF · 2026-01-06

**Summary:**

This is an interesting paper that addresses the feature space transformation of frozen foundation models as an encoder. Specifically, they propose the Manifold Residual (MR) block, which would preserve the "GEOMETRIC shape" of the original embeddings. This design is based on their findings that naive learnable linear layers distort the low-dimensional manifold of features significantly. Therefore, the MR block replaces these linear layers with a dual-pathway structure: a fixed random projection (geometric anchor) and a trainable low-rank residual. Extensive experiments across multiple datasets (Camelyon16, TCGA-NSCLC, TCGA-RCC) and MIL backbones demonstrate that the method improves few-shot performance while significantly reducing trainable parameters.

**Reviewer Concerns:**

Addressed Concerns:
1. benchmarking against SOTA (Reviewer k14d): The authors added a comparison to FOCUS (CVPR 2025) and excluded MGPATH and MSCPT, arguing that they are not fair counterparts. The results show MR achieves comparable or better performance with significantly fewer parameters.
2. Reviewer 7Ghg suggested validating on more real-world few-shot tasks. In response,  the authors included experiments on two treatment response datasets (Boehmk and Trastuzumab), demonstrating that the gains hold on clinically realistic tasks beyond artificially subsampled cohorts. I think this experiment expands the clinical utility of this paper.
3.  Reviewer tHAq pointed out that parameter reduction could be another factor, rather than geometry. In rebuttal, the authors conducted a crucial capacity-matched control experiment to address this question directly. This demonstrated that MR outperforms a standard linear layer even when the linear layer is restricted to the same number of parameters, validating the geometric hypothesis over a pure "parameter efficiency" explanation.
4. Reviewer jYL1 suggested there are some presentation clarity issues and more metrics: The authors provided AUPRC for imbalanced settings and clarified the exact insertion points of the MR block in various backbones.

One outstanding concern might be from Reviewer tHAq, who maintained a philosophical disagreement regarding whether a random anchor "destroys" or "preserves" geometry. However, the authors effectively countered this with Random Projection theory references and the empirical capacity-matched results.

**Reviewer Scores:**

Reviewer k14d is highly positive and would keep the rating.

Reviewer jYL1 turned positive from 4 to 6 or 8 with more results.

Reviewer 7Ghg might keep 6 or increase a little bit since concerns about datasets were fully addressed.

Reviewer tHAq gave 2 initially, and might likely remain skeptical despite strong evidence.

---

### Decision · Program_Chairs · 2026-01-26

Accept (Poster)